# Fair Clustering via Alignment

## Abstract

Algorithmic fairness in clustering aims to balance the proportions of instances assigned to each cluster with respect to a given sensitive attribute. Recently, numerous algorithms have been developed for Fair Clustering (FC), most of which optimize a clustering objective under specifically designed fairness constraints. However, the inherent complexity or approximation of constrained optimization problems makes it challenging to achieve the optimal trade-off between fairness level and clustering utility in practice. For example, the obtained clustering utility by an existing FC algorithm might be suboptimal, or achieving a certain fairness level could be numerically unstable. To resolve these limitations, we propose a new FC algorithm based on a novel decomposition of the fair $K$-means clustering objective function. The proposed algorithm, called Fair Clustering via Alignment (FCA), operates by (i) finding a joint probability distribution to align the data from different protected groups, and (ii) optimizing cluster centers in the aligned space. A key advantage of FCA is that it guarantees (local) optimal clustering utility for any given fairness level while avoiding the need to solve complex constrained optimization problems, thereby obtaining (local) optimal fair clustering in practice. Experiments show that FCA offers several empirical benefits over existing methods such as (i) attaining the optimal trade-off between fairness level and clustering utility, and (ii) achieving near-perfect fairness level without numerical instability.

## 1 Introduction

As artificial intelligence (AI) technology has advanced and been successfully applied to diverse domains and tasks, the requirement for AI systems to make fair decisions (i.e., algorithmic fairness) has emerged as an important issue. This requirement is particularly necessary when observed data possess historical biases with respect to specific sensitive attributes, leading to unfair outcomes of learned models based on such biased data (Angwin et al., 2016; Ingold & Soper, 2016; Damodaran et al., 2018; Mehrabi et al., 2019). Specifically, group fairness is a category within algorithmic fairness that ensures models do not discriminate against certain protected groups, which are defined by specific sensitive attributes (e.g., men vs. women, white vs. black). In response, a large amount of researches have been conducted to develop algorithms for mitigating such biases in various supervised learning tasks such as classification (Zafar et al., 2017; Donini et al., 2018; Agarwal et al., 2018; Quadrianto et al., 2019; Jiang et al., 2020) and regression (Agarwal et al., 2019; Chzhen et al., 2020).

Along with supervised learning, algorithmic fairness for unsupervised learning tasks including clustering has also received much attention. Clustering algorithms have long been employed as fundamental unsupervised learning methods in various fields of machine learning, such as recommendation systems (Ahuja et al., 2019; Widiyaningtyas et al., 2021), image processing (Le, 2013; Guo et al., 2020; Mittal et al., 2022), and language modeling (Butnaru & Ionescu, 2017; Zhang et al., 2023).

**Related works for Fair Clustering (FC)** Combining algorithmic fairness and clustering, the notion of Fair Clustering (FC) has been initially introduced in Chierichetti et al. (2017). FC operates under the setting that each data point is assigned a color (i.e., a sensitive attribute), with the goal that the proportion of each color within each cluster should be similar to that in the population. To achieve this goal, various algorithms have been developed to minimize a given clustering objective under pre-specified fairness constraints (Bera et al., 2019; Kleindessner et al., 2019; Backurs et al., 2019; Li et al., 2020; Esmaeili et al., 2021; Ziko et al., 2021; Zeng et al., 2023), to name a few. Section A.5 of Appendix covers other fairness notions, including proportional fairness and individual fairness.

We can roughly categorize the existing FC algorithms into three: (i) pre-processing, (ii) in-processing, and (iii) post-processing. Pre-processing methods (Chierichetti et al., 2017; Backurs et al., 2019) involve transforming instances into a fair space based on the concept of fairlets. Fairlets are small subsets that satisfy (perfect) fairness, and thus performing standard clustering on the fairlet space yields a fair clustering (see Section A.1 of Appendix for details). In-processing methods (Kleindessner et al., 2019; Ziko et al., 2021; Li et al., 2020; Zeng et al., 2023) aim to simultaneously find both the cluster centers and assignments of the optimal fair clustering by solving certain constrained optimization problems. Post-processing methods (Bera et al., 2019; Harb & Lam, 2020) focus on finding fair assignments given fixed cluster centers. The fixed cluster centers are typically determined by a standard clustering algorithm.

**Our contributions**  In this paper, we focus on the *optimal trade-off* between fairness level and clustering utility, which aims to maximize clustering utility under a given fairness level. While the trade-off between fairness level and clustering utility is inevitable (Bertsimas et al., 2011; Chhabra et al., 2021), achieving the optimal trade-off with existing FC algorithms remains challenging. For instance, pre- or post-processing algorithms usually result in suboptimal clustering utility due to indirect maximization of clustering utility (e.g., Backurs et al. (2019); Esmaeili et al. (2021)). Even when designed for achieving the optimal trade-off, in-processing algorithms may have trouble due to numerical instability, particularly when a given fairness level is high (e.g., Ziko et al. (2021)).

This paper aims to address these challenges by developing a new in-processing algorithm that can achieve the optimal trade-off between fairness level and clustering utility without numerical instability. The primary idea of our proposed algorithm is to align data from two protected groups by transforming them into a common space (called the aligned space) and then applying a standard clustering algorithm in the aligned space. We prove that the optimal fair clustering, i.e., the clustering with minimal clustering cost under a given fairness constraint, is equivalent to the optimal clustering in the aligned space.

Based on the theoretical result, we devise a new FC algorithm, called *Fair Clustering via Alignment (FCA)*. FCA alternately finds the aligned space and the optimal clustering in the aligned space until convergence. To find the aligned space, we develop a modified version of an algorithm for finding the optimal transport map (Kantorovich, 2006), while a standard clustering algorithm can be used to find the optimal clustering in the aligned space.

It is worth noting that our proposed algorithm, FCA, can be compared with the fairlet-based methods (e.g., Backurs et al. (2019)). While existing fairlet-based methods find fairlets and perform clustering sequentially (separately), FCA finds the aligned space and performs clustering simultaneously to obtain the (local) optimal fair clustering. A detailed comparison is provided in Remark 3.2.

The main contributions of this paper can be summarized as follows.

- We provide a novel decomposition of the fair clustering cost into two components: (i) the transport cost with respect to a joint distribution between two protected groups, and (ii) the clustering cost with respect to cluster centers in the aligned space.
- Based on the decomposition, we develop a novel FC algorithm called FCA (Fair Clustering via Alignment), which is transparent, stable, and guarantees convergence.
- Theoretically, we prove that FCA achieves the optimal trade-off between fairness level and clustering utility.
- Experimentally, we show that FCA (i) outperforms existing baseline FC algorithms in terms of both the trade-off and numerical stability, and (ii) optimally controls the trade-off across various fairness levels.

## 2 PRELIMINARIES

**Notations**  Let $\mathcal{D} = \{(\mathbf{x}_i, s_i)\}_{i=1}^n$ be a given dataset, where $\mathbf{x}_i \in \mathbb{R}^d$ and $s_i \in \{0, 1\}$ are $d$-dimensional vector of data and binary variable for the sensitive attribute, respectively. Denote $(\mathbf{X}, S)$ as the random vector whose joint distribution denoted as $\mathbb{P}$ is the empirical distribution on $\mathcal{D}$. Let $\mathbb{P}_s$ represents the conditional distribution of $\mathbf{X}$ given $S = s$. We specifically define these distributions to discuss the (probabilistic) matching between two protected groups of different sizes, and subsequent probabilistic assignments for fair clustering. We denote $\mathbb{E}$ and $\mathbb{E}_s$ as the expectation operators of $\mathbb{P}$

and $\mathbb{P}_s$, respectively. Let $\mathcal{X} = \{\mathbf{x}_i : i = 1, \ldots, n\}$, $\mathcal{X}_s = \{\mathbf{x}_i : s_i = s\}$, and $n_s := |\mathcal{X}_s|$. Denote $\|\cdot\|^2$ as the $L_2$ norm.

We assume that the number of clusters, represented by $K \in \mathbb{N}$, is given a priori. The $K$-many cluster centers are denoted as $\boldsymbol{\mu} := \{\mu_1, \ldots, \mu_K\}$ where $\mu_k \in \mathbb{R}^d, \forall k \in [K] = \{1, \ldots, K\}$. Let $\mathcal{A} : \mathcal{X} \times \{0, 1\} \to \mathcal{S}^K$ be an assignment function that receives $(\mathbf{x}, s) \in \mathcal{X} \times \{0, 1\}$ as input and returns the probabilities of the data point belonging to each cluster, where $\mathcal{S}^K$ is the $K$-dimensional simplex (i.e., for all $\mathbf{v} = [v_1, \ldots, v_K]^\top \in \mathcal{S}^K$, $v_k \geq 0$ for all $k \in [K]$ and $\sum_{k=1}^K v_k = 1$). We consider this probabilistic assignment function to guarantee the existence of the optimal clustering.

**Problem setting** We present the mathematical formulation of the fair clustering objective. The original objective of the standard (i.e., fair-unaware) $K$-means clustering is to minimize the clustering cost $C(\boldsymbol{\mu}, \mathcal{A}) := \mathbb{E} \sum_{k=1}^K \mathcal{A}(\mathbf{X}, S)_k \|\mathbf{X} - \mu_k\|^2$ with respect to $\boldsymbol{\mu}$ and $\mathcal{A}$. Note that the optimal assignment function is deterministic, i.e., $\mathcal{A}(\mathbf{x}, s)_k = \mathbb{1}(\arg\min_{k' \in [K]} \|\mathbf{x} - \mu_{k'}^\circ\|^2 = k)$ for given $(\mathbf{x}, s) \in \mathcal{X}_s \times \{0, 1\}$, where $\mu_{k'}^\circ$ are the centers of the optimal clustering. Thus, $C(\boldsymbol{\mu}, \mathcal{A})$ becomes $\mathbb{E} \min_k \|\mathbf{X} - \mu_k\|^2$, and the optimal clustering is found by finding $\boldsymbol{\mu}$ minimizing $\mathbb{E} \min_k \|\mathbf{X} - \mu_k\|^2$.

An assignment function $\mathcal{A}$ is said to be *fair* if it satisfies $\mathbb{E}(\mathcal{A}(\mathbf{X}, s)_k | S = s) \approx \mathbb{E}(\mathcal{A}(\mathbf{X}, S)_k), \forall k \in [K], \forall s \in \{0, 1\}$. This constraint ensures the probability of data belonging to a cluster be (nearly) independent with the sensitive attribute, resulting in fair clustering. That is, we find the cluster center $\boldsymbol{\mu}$ and the assignment function $\mathcal{A}$ that minimize $C(\boldsymbol{\mu}, \mathcal{A})$ among $\boldsymbol{\mu} \in \mathbb{R}^K$ and all fair assignment functions $\mathcal{A}$ satisfying $\mathbb{E}(\mathcal{A}(\mathbf{X}, s)_k | S = s) \approx \mathbb{E}(\mathcal{A}(\mathbf{X}, S)_k), \forall k \in [K], \forall s \in \{0, 1\}$.

**Fairness measure** To assess the fairness level (i.e., how similar $\mathbb{E}(\mathcal{A}(\mathbf{X}, s)_k | S = s)$ are to $\mathbb{E}(\mathcal{A}(\mathbf{X}, S)_k)$) in clustering, *Balance* is a popularly used measure (Chierichetti et al., 2017; Bera et al., 2019; Backurs et al., 2019; Esmaeili et al., 2021; Ziko et al., 2021; Zeng et al., 2023), which is defined in Definition 2.1 below. Note that we slightly modify the original definition from Chierichetti et al. (2017) by normalizing the value to lie within $[0, 1]$.

**Definition 2.1** (Balance). *For a given assignment function $\mathcal{A}$, Balance (of $\mathcal{A}$) is defined by*

$$\text{Balance}(\mathcal{A}) := \min_{k \in [K]} \min_{s \in \{0,1\}} \frac{\mathbb{E}(\mathcal{A}(\mathbf{X}, s)_k | S = s)}{\mathbb{E}(\mathcal{A}(\mathbf{X}, S)_k)} = \min_{k \in [K]} \min_{s \in \{0,1\}} \frac{\frac{1}{n_s} \sum_{\mathbf{x}_i \in \mathcal{X}_s} \mathcal{A}(\mathbf{x}_i, s)_k}{\frac{1}{n} \sum_{(\mathbf{x}_i, s_i) \in \mathcal{D}} \mathcal{A}(\mathbf{x}_i, s_i)_k} \in [0, 1]. \quad (1)$$

For a small $\alpha > 0$, the objective of FC is to minimize $C(\boldsymbol{\mu}, \mathcal{A})$ with respect to $\boldsymbol{\mu} \in \mathbb{R}^K$ and $\mathcal{A}$, under the constraint that $\text{Balance}(\mathcal{A}) > 1 - \alpha$. The lower Balance is, the less fair the clustering is. Furthermore, any perfectly fair clustering satisfies $\text{Balance}(\mathcal{A}) = 1$. In Section 3, we prove that this FC objective can be decomposed into the sum of (i) the cost of transporting data from different protected groups to a common space (called the aligned space), and (ii) the clustering cost in the aligned space built by the transported data. In what follows, we write $\mathcal{A}(\cdot, s) = \mathcal{A}_s(\cdot)$ and $C(\boldsymbol{\mu}, \mathcal{A}) = C(\boldsymbol{\mu}, \mathcal{A}_0, \mathcal{A}_1)$ whenever their meanings are clear.

## 3 REFORMULATION OF THE FAIR CLUSTERING OBJECTIVE

This section presents our main theoretical contribution: a novel decomposition of the perfectly fair (i.e., Balance = 1) clustering objective, which motivates our proposed FCA algorithm. The case of Balance < 1 is considered in Section 4 after introducing FCA algorithm for perfect fairness.

In Section 3.1, we introduce our idea through discussing the simplest case where the two protected groups are of equal size ($n_0 = n_1$). Then, in Section 3.2, we generalize to the unequal case ($n_0 \neq n_1$) by introducing the notions of the alignment map and aligned space, constructed by a given joint distribution on $\mathcal{X}_0 \times \mathcal{X}_1$. We prove that there exists a joint distribution such that the objective function of perfectly fair clustering can be decomposed into the sum of the transport cost with respect to the joint distribution and the clustering cost on the aligned space. Full proofs of all the theoretical findings in this section are given in Section B of Appendix.

### 3.1 CASE OF EQUAL SAMPLE SIZES: $n_0 = n_1$

Assume that the sizes of two protected groups are equal (i.e., $n_0 = n_1$). For simplicity, we only consider deterministic assignment functions, meaning $\mathcal{A}_s(\mathbf{x})_k \in \{0, 1\}$. The case of probabilistic

assignment functions with $n_0 \neq n_1$ is discussed in the next subsection, where it is proven in Remark 3.4 that the optimal assignment function is deterministic when $n_0 = n_1$.

The core idea of FCA is to **match two instances from different protected groups and assign them to a same cluster**. By doing so – matching all instances from $\mathcal{X}_0$ to $\mathcal{X}_1$ in a one-to-one fashion and assigning each pair to a same cluster – the resulting clustering becomes perfectly fair.

Conversely, suppose we are given a perfectly fair clustering constructed by a deterministic assignment function $\mathcal{A}$. Since $n_0 = n_1$ and $\mathcal{A}$ is deterministic, there exists a one-to-one matching between $\mathcal{X}_0$ and $\mathcal{X}_1$ such that two matched instances belong to a same cluster. Thus, we can decompose the clustering cost in terms of the one-to-one matching, as presented in Theorem 3.1.

**Theorem 3.1.** *For any given perfectly fair deterministic assignment function $\mathcal{A}$ and cluster centers $\boldsymbol{\mu}$, there exists a one-to-one matching map $\mathbf{T} : \mathcal{X}_s \to \mathcal{X}_{s'}$ such that, for any $s \in \{0, 1\}$,*

$$C(\boldsymbol{\mu}, \mathcal{A}_0, \mathcal{A}_1) = \mathbb{E}_s \sum_{k=1}^{K} \mathcal{A}_s(\mathbf{X})_k \bigg( \underbrace{\frac{\|\mathbf{X} - \mathbf{T}(\mathbf{X})\|^2}{4}}_{\text{Transport cost w.r.t } \mathbf{T}} + \underbrace{\left\| \frac{\mathbf{X} + \mathbf{T}(\mathbf{X})}{2} - \mu_k \right\|^2}_{\text{Clustering cost w.r.t. } \boldsymbol{\mu} \text{ and } \mathbf{T}} \bigg). \qquad (2)$$

The assignment function $\mathcal{A}$ that minimizes eq. (2) given $\boldsymbol{\mu}$ and $\mathbf{T}$ assigns both $\mathbf{x}$ and $\mathbf{T}(\mathbf{x})$ to cluster $k$, where $k = \arg\min_{k' \in [K]} \|(\mathbf{x} + \mathbf{T}(\mathbf{x}))/2 - \mu_{k'}\|^2$. Hence, the optimal perfectly fair clustering can by found by minimizing $\mathbb{E}_s \left( \|\mathbf{X} - \mathbf{T}(\mathbf{X})\|^2/4 + \min_k \|(\mathbf{X} + \mathbf{T}(\mathbf{X}))/2 - \mu_k\|^2 \right)$ with respect to $\boldsymbol{\mu}$ and $\mathbf{T}$, instead of finding the optimal $\boldsymbol{\mu}$ and $\mathcal{A}$ minimizing $C(\boldsymbol{\mu}, \mathcal{A}_0, \mathcal{A}_1)$. We update $\boldsymbol{\mu}$ for a given $\mathbf{T}$ by applying a standard clustering algorithm to $\{\frac{\mathbf{x}+\mathbf{T}(\mathbf{x})}{2}, \mathbf{x} \in \mathcal{X}_s\}$, which is called the *aligned space*. To update $\mathbf{T}$, any algorithm for finding the optimal matching can be used, where the cost between two instances $\mathbf{x}_0 \in \mathcal{X}_0$ and $\mathbf{x}_1 \in \mathcal{X}_1$ is given by $\|\mathbf{x}_0 - \mathbf{x}_1\|^2/4 + \min_k \|(\mathbf{x}_0 + \mathbf{x}_1)/2 - \mu_k\|^2$ (see Section 4.1 for the specific algorithm we use). Note that there are no complex constraints in updating $\boldsymbol{\mu}$ and $\mathbf{T}$. Finally, we define $\mathcal{A}$ corresponding to $\mathbf{T}$, which assigns both $\mathbf{x}$ and $\mathbf{T}(\mathbf{x})$ to a same cluster on the aligned space $\{\frac{\mathbf{x}+\mathbf{T}(\mathbf{x})}{2}, \mathbf{x} \in \mathcal{X}_s\}$. See Section A.4 of Appendix for a similar decomposition result to any $L_p$ norm ($p \geq 1$), e.g., $L_1$ norm for $K$-median clustering.

**Remark 3.2** (Comparison to the fairlet-based methods). *Although our idea of matching data from different protected groups may seem similar to the fairlet-based methods (e.g., Chierichetti et al. (2017), see Section A.1 of Appendix for details about fairlets), it fundamentally differs in a key way:*

*Our method is an in-processing approach that directly minimizes the clustering cost with respect to both the matching map and cluster centers simultaneously. In contrast, the fairlet-based method is a two-step pre-processing approach, which does not directly minimize the clustering cost; instead, it first searches for fairlets and then finds the optimal cluster centers on the set of representatives for each fairlet. As a result, in the fairlet-based method, the matchings and cluster centers are not jointly optimized, which may lead to a suboptimal clustering utility.*

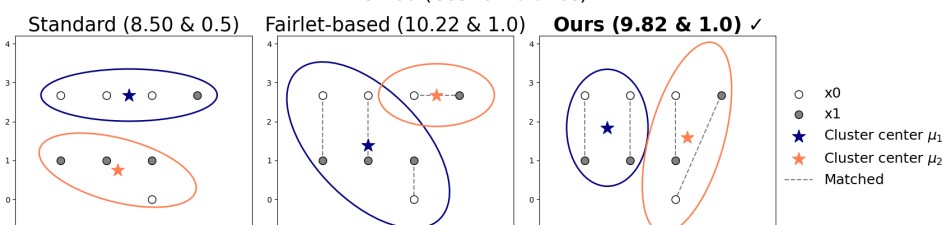

Figure 1: Comparison between the fairlet-based method and our approach with $n_0 = n_1 = 4$ and $K = 2$. For a fair comparison, the representative of each fairlet is set as the mean vector of the data within that fairlet. The standard $K$-means algorithm is then applied to this set of representatives. The resulting clusterings are visualized using contours. While both methods achieve Balance $= 1$, our approach yields a lower cost ($9.82 < 10.22$), due to different matchings.

*Figure 1 illustrates how the fairlet-based method can produce suboptimal clustering, when compared to our approach. Specifically, more efficient matchings exist that yield higher clustering utility than the matchings of fairlets, and our approach is designed to find these efficient matchings. We confirm this claim more comprehensively using real benchmark datasets in Section 5.2.*

## 3.2 CASE OF UNEQUAL SAMPLE SIZES: $n_0 \neq n_1$

The main approach for $n_0 \neq n_1$ is similar to $n_0 = n_1$, but instead of the matching map $\mathbf{T}$, we find the optimal joint distribution $\mathbb{Q}$ of $(\mathbf{X}_0, \mathbf{X}_1)$ minimizing the clustering cost. That is, we reformulate the perfectly fair clustering cost in terms of cluster centers $\boldsymbol{\mu}$ and joint distribution $\mathbb{Q}$ whose marginal distributions are $\mathbb{P}_s, s \in \{0, 1\}$. Note that $\mathbb{Q}$ serves as a smooth and stochastic version of $\mathbf{T}$.

Let $\mathcal{Q} = \{$all joint distributions $\mathbb{Q}$ on $\mathcal{X}_0 \times \mathcal{X}_1$ with $\mathbb{Q}_s = \mathbb{P}_s, s \in \{0, 1\}\}$. Theorem 3.3 below, which is the main result of this paper and motivation of our proposed algorithm, proves that the optimal perfectly fair clustering can be found by finding the joint distribution $\mathbb{Q}$ and cluster centers $\boldsymbol{\mu}$ optimally. Let $\pi_s = n_s/(n_s + n_{s'})$ for $s \neq s' \in \{0, 1\}$.

**Theorem 3.3.** *Let $\boldsymbol{\mu}^*$ and $\mathbb{Q}^*$ be the cluster centers and joint distribution solving*

$$\min_{\boldsymbol{\mu}, \mathbb{Q} \in \mathcal{Q}} \mathbb{E}_{(\mathbf{X}_0, \mathbf{X}_1) \sim \mathbb{Q}} \left( 2\pi_0 \pi_1 \|\mathbf{X}_0 - \mathbf{X}_1\|^2 + \min_k \|\mathbf{T}^{\mathrm{A}}(\mathbf{X}_0, \mathbf{X}_1) - \mu_k\|^2 \right), \quad (3)$$

*where $\mathbf{T}^{\mathrm{A}}(\mathbf{x}_0, \mathbf{x}_1) = \pi_0 \mathbf{x}_0 + \pi_1 \mathbf{x}_1$, which is called the alignment map.*
*Then, $(\boldsymbol{\mu}^*, \mathcal{A}_0^*, \mathcal{A}_1^*)$ is the solution of the perfectly fair $K$-means clustering, where $\mathcal{A}_0^*(\mathbf{x})_k :=$ $\mathbb{Q}^* \left( \arg\min_{k'} \|\mathbf{T}^{\mathrm{A}}(\mathbf{x}, \mathbf{X}_1) - \mu_{k'}\|^2 = k | \mathbf{X}_0 = \mathbf{x} \right)$ and $\mathcal{A}_1^*(\mathbf{x})_k$ is defined similarly.*

This result implies that, by simultaneously minimizing the transport cost $\|\mathbf{X}_0 - \mathbf{X}_1\|^2$ and finding the cluster centers in the aligned space $\{\mathbf{T}^{\mathrm{A}}(\mathbf{X}_0, \mathbf{X}_1) : (\mathbf{X}_0, \mathbf{X}_1) \sim \mathbb{Q}\}$, we obtain the optimal perfectly fair clustering. A notable observation is that there is no explicit constraint in eq. (3). In fact, the constraint for perfect fairness (i.e., $\mathbb{E}_s \mathcal{A}_s(\mathbf{X})_k = \mathbb{E} \mathcal{A}_S(\mathbf{X})_k$ for all $k, s$) is implicitly satisfied through the use of the alignment map $\mathbf{T}^{\mathrm{A}}(\mathbf{x}_0, \mathbf{x}_1)$. In conclusion, solving (3) with respect to $\boldsymbol{\mu}$ and $\mathbb{Q}$ yields the optimal perfectly fair clustering. The algorithm for solving (3) is detailed in Section 4.

**Remark 3.4** ($\mathcal{A}^*$ is deterministic when $n_0 = n_1$). *When $n_0 = n_1$, the optimal assignment function $\mathcal{A}^*$ in Theorem 3.3 becomes deterministic. This is because $\mathbb{Q}$ corresponds to the one-to-one matching map $\mathbf{T}$ in eq. (2). The detailed proof is provided at the end of the proof of Theorem 3.3.*

## 4 PROPOSED ALGORITHMS

### 4.1 FCA: FAIR CLUSTERING VIA ALIGNMENT

Based on Theorem 3.3, we propose an algorithm for finding the optimal perfectly fair clustering, called ***Fair Clustering via Alignment (FCA)***. FCA consists of two phases. Phase 1 finds the joint distribution $\mathbb{Q}$ with the cluster centers $\boldsymbol{\mu}$ being fixed. Phase 2 updates the cluster centers $\boldsymbol{\mu}$ with the joint distribution $\mathbb{Q}$ being fixed. Then, FCA iterates these two phases until cluster centers converge.

**Phase 1: Finding the joint distribution** Phase 1 finds the optimal joint distribution $\mathbb{Q}$ that minimizes the cost in eq. (3) given $\boldsymbol{\mu}$. For this goal, we recall the Kantorovich problem (Kantorovich, 2006; Villani, 2008), which finds the optimal coupling between two measures for a given cost matrix. For two given empirical distributions on $\mathcal{X}_0 = \{\mathbf{x}_i^{(0)}\}_{i=1}^{n_0}$ and $\mathcal{X}_1 = \{\mathbf{x}_j^{(1)}\}_{j=1}^{n_1}$, the cost matrix between the two is given by $\mathbf{C} := [c_{i,j}] \in \mathbb{R}_+^{n_0 \times n_1}$ where $c_{i,j} = \|\mathbf{x}_i^{(0)} - \mathbf{x}_j^{(1)}\|^2$. Then, the Kantorovich problem is to find the optimal joint distribution defined by the matrix $\Gamma = [\gamma_{i,j}] \in \mathbb{R}_+^{n_0 \times n_1}$ that solves the following objective: $\min_\Gamma \|\mathbf{C} \odot \Gamma\|_1 = \min_{\gamma_{i,j}} c_{i,j} \gamma_{i,j}$ s.t. $\sum_{i=1}^{n_0} \gamma_{i,j} = \frac{1}{n_1}, \sum_{j=1}^{n_1} \gamma_{i,j} = \frac{1}{n_0}, \gamma_{i,j} \geq 0$. See Section A.2 of Appendix for more details regarding the Kantorovich problem.

We modify this Kantorovich problem to solve eq. (3). The transport cost matrix between the two (the first term of eq. (3)) is defined by $\mathbf{C} := [c_{i,j}] \in \mathbb{R}_+^{n_0 \times n_1}$ where $c_{i,j} = 2\pi_0 \pi_1 \|\mathbf{x}_i^{(0)} - \mathbf{x}_j^{(1)}\|^2$. The clustering cost matrix between the aligned data and their assigned centers (the second term of eq. (3)) is defined by $\mathbf{D} := [d_{i,j}] \in \mathbb{R}_+^{n_0 \times n_1}$ where $d_{i,j} = \min_{k \in [K]} \|\pi_0 \mathbf{x}_i^{(0)} + \pi_1 \mathbf{x}_j^{(1)} - \mu_k\|^2$. Then, we find the optimal coupling $\Gamma = [\gamma_{i,j}] \in \mathbb{R}_+^{n_0 \times n_1}$ that solves the following objective:

$$\min_\Gamma \|(\mathbf{C} + \mathbf{D}) \odot \Gamma\|_1 = \min_{\gamma_{i,j}}(c_{i,j} + d_{i,j})\gamma_{i,j} \text{ s.t. } \sum_{i=1}^{n_0} \gamma_{i,j} = \frac{1}{n_1}, \sum_{j=1}^{n_1} \gamma_{i,j} = \frac{1}{n_0}, \gamma_{i,j} \geq 0. \quad (4)$$

Clearly, this problem becomes the original Kantorovich problem when $\mathbf{D} = \mathbf{0}$. In other words, the difference between our problem and the original Kantorovich problem is the existence of the

matrix $\mathbf{D}$, which represents the clustering cost on the aligned space. Hence, this objective can be also efficiently solved using linear programming, similar to the Kantorovich problem (Villani, 2008).

Based on the optimal coupling $\Gamma$ that solves eq. (4), we define the joint distribution as $\mathbb{Q}(\{\mathbf{x}_i^{(0)}, \mathbf{x}_j^{(1)}\}) = \gamma_{i,j}$. That is, we have the measures (weights) of $\{\gamma_{i,j}\}_{i\in[n_0], j\in[n_1]}$ for the aligned points in $\{\mathbf{T}^{\mathrm{A}}(\mathbf{x}_i^{(0)}, \mathbf{x}_j^{(1)})\}_{i\in[n_0], j\in[n_1]}$. Finally, we define the corresponding aligned space as the $n_0 \times n_1$ many pairs of the weight and the aligned instances: $\{(\gamma_{i,j}, \mathbf{T}^{\mathrm{A}}(\mathbf{x}_i^{(0)}, \mathbf{x}_j^{(1)}))\}_{i\in[n_0], j\in[n_1]}$.

**Phase 2: Optimizing cluster centers** Then, we optimize the cluster centers $\boldsymbol{\mu}$ on the aligned space obtained in Phase 1, by solving $\min_{\boldsymbol{\mu}} \sum_{i=1}^{n_0} \sum_{j=1}^{n_1} \gamma_{i,j} \min_k \|\mathbf{T}^{\mathrm{A}}(\mathbf{x}_i^{(0)}, \mathbf{x}_j^{(1)}) - \mu_k\|^2$. Standard clustering algorithms, such as the $K$-means++ algorithm (Arthur & Vassilvitskii, 2007) or a gradient descent-based algorithm, can be used to update $\boldsymbol{\mu}$. Note that in Section 5.4, we empirically compare these two algorithms to show that FCA is stable regardless of the algorithm used to optimize $\boldsymbol{\mu}$.

---

**FCA algorithm** We alternately update the joint distribution (Phase 1) and the cluster centers (Phase 2). At each iteration, we (i) find $\mathbb{Q}$ that minimizes the sum of the transport and clustering costs to build the aligned space, and then (ii) update the cluster centers by minimizing the clustering cost on the aligned space. This procedure continues until the cluster centers converge.

Then, define the assignment function as $\mathcal{A}_s(\mathbf{x}_i^{(s)})_k := \sum_{\mathbf{x}_j^{(s')} \in \mathcal{X}_{s'}} n_s \gamma_{i,j} \mathbb{1}(\arg\min_{k'} \|\pi_s \mathbf{x}_i^{(s)} + \pi_{s'} \mathbf{x}_j^{(s')} - \mu_{k'}\|^2 = k), k \in [K]$ for $\mathbf{x}_i^{(s)} \in \mathcal{X}_s$. Whenever needed, we can deterministically assign $\mathbf{x}_i^{(s)}$ to the $k^{\mathrm{th}}$ cluster where $k$ is the maximum argument of $\mathcal{A}_s(\mathbf{x}_i^{(s)})$.

---

### 4.2 FCA-C: CONTROL OF BALANCE

In addition to perfect fairness, it is also important to find optimal fair clusterings with Balance $< 1$. For this purpose, we introduce a feasible relaxation of FCA called *FCA-Control (FCA-C)*, a variant of FCA specifically designed for controlling Balance.

Let $\mathcal{W} \subset \mathcal{X}_0 \times \mathcal{X}_1$ be a given subset. The idea of our relaxation is to *apply FCA algorithm only to instances in $\mathcal{W}^c = \mathcal{X}_0 \times \mathcal{X}_1 \setminus \mathcal{W}$ and to apply the standard $K$-means algorithm to those in $\mathcal{W}$*. Note that $\mathcal{W} = \emptyset$ becomes FCA, while $\mathcal{W} = \mathcal{X}_0 \times \mathcal{X}_1$ results in standard (fair-unaware) clustering. Denote $\epsilon > 0$ as a hyper-parameter that controls Balance. For a given $\epsilon$, FCA-C algorithm minimizes

$$\mathbb{E}_{\mathbf{X}_0, \mathbf{X}_1 \sim \mathbb{Q}}\left( \underbrace{\left(2\pi_0\pi_1\|\mathbf{X}_0 - \mathbf{X}_1\|^2 + \min_k \|\mathbf{T}^{\mathrm{A}}(\mathbf{X}_0, \mathbf{X}_1) - \mu_k\|^2\right)}_{\text{FCA cost}} \mathbb{1}\left((\mathbf{X}_0, \mathbf{X}_1) \in \mathcal{W}^c\right) \right)$$

$$+ \mathbb{E}_{\mathbf{X}_0, \mathbf{X}_1 \sim \mathbb{Q}}\left( \underbrace{\min_k \left(\pi_0\|\mathbf{X}_0 - \mu_k\|^2\right) + \min_k \left(\pi_1\|\mathbf{X}_1 - \mu_k\|^2\right)}_{K\text{-means cost}} \mathbb{1}\left((\mathbf{X}_0, \mathbf{X}_1) \in \mathcal{W}\right) \right) \tag{5}$$

with respect to $\boldsymbol{\mu}, \mathbb{Q}$ and $\mathcal{W}$ satisfying $\mathbb{Q}((\mathbf{X}_0, \mathbf{X}_1) \in \mathcal{W}) \leq \epsilon$. See Figure 2 for an example visualizing $\mathcal{W}$ and $\mathbb{Q}$. Then, we construct the assignment function $\mathcal{A}_0$ by

$$\mathcal{A}_0(\mathbf{x}_0)_k = \mathbb{P}_1(\arg\min_{k'} \|\mathbf{T}^{\mathrm{A}}(\mathbf{x}_0, \mathbf{X}_1) - \mu_{k'}\|^2 = k, (\mathbf{x}_0, \mathbf{X}_1) \in \mathcal{W}^c)$$

$$+ \mathbb{1}(\arg\min_{k'} \|\mathbf{x}_0 - \mu_{k'}\|^2 = k) \cdot \mathbb{P}_1((\mathbf{x}_0, \mathbf{X}_1) \in \mathcal{W}). \tag{6}$$

The assignment function $\mathcal{A}_1$ is defined similarly. A practical algorithm for FCA-C is described below.

---

**FCA-C algorithm** We alternately update $\mathbb{Q}, \boldsymbol{\mu}$ and $\mathcal{W}$ as the following.
- (Update $\mathbb{Q}$) First, calculate the costs: $c_{i,j}$ for $(\mathbf{x}_i^{(0)}, \mathbf{x}_j^{(1)}) \in \mathcal{W}$ and $c_{i,j} + d_{i,j}$ for $(\mathbf{x}_i^{(0)}, \mathbf{x}_j^{(1)}) \in \mathcal{W}^c$. Then, find $\Gamma = [\gamma_{i,j}]_{i,j}$ (i.e., $\mathbb{Q}(\{\mathbf{x}_i^{(0)}, \mathbf{x}_j^{(1)}\} = \gamma_{i,j})$) minimizing (5).
- (Update $\boldsymbol{\mu}$) Minimize (5) with respect to $\boldsymbol{\mu}$: $\min_{\boldsymbol{\mu}} \sum_i \sum_j \gamma_{i,j} \big[ \min_k \|\mathbf{T}^{\mathrm{A}}(\mathbf{x}_i^{(0)}, \mathbf{x}_j^{(1)}) - \mu_k\|^2 \mathbb{1}(\mathbf{x}_i^{(0)}, \mathbf{x}_j^{(1)} \in \mathcal{W}^c) + (\min_k \|\mathbf{x}_i^{(0)} - \mu_k\|^2 + \min_k \|\mathbf{x}_j^{(1)} - \mu_k\|^2) \mathbb{1}(\mathbf{x}_i^{(0)}, \mathbf{x}_j^{(1)} \in \mathcal{W})\big]$.
- (Update $\mathcal{W}$) Let $\eta(\mathbf{x}_i^{(0)}, \mathbf{x}_j^{(1)}) := 2\pi_0\pi_1\|\mathbf{x}_i^{(0)} - \mathbf{x}_j^{(1)}\|^2 + \min_k \|\mathbf{T}^{\mathrm{A}}(\mathbf{x}_i^{(0)}, \mathbf{x}_j^{(1)}) - \mu_k\|^2$ and $\eta_\epsilon$ be the $\epsilon$th upper quantile. Update $\mathcal{W} = \{(\mathbf{x}_i^{(0)}, \mathbf{x}_j^{(1)}) \in \mathcal{X}_0 \times \mathcal{X}_1 : \eta(\mathbf{x}_i^{(0)}, \mathbf{x}_j^{(1)}) > \eta_\epsilon\}$.

---

Figure 2: Example illustration of $\mathcal{W}$ and $\Gamma$ when $(n_0, n_1) = (2, 5)$ and $\epsilon = \gamma_{1,4} + \gamma_{2,2} + \gamma_{2,5}$.

Notably, this FCA-C algorithm has an interesting theoretical property in terms of achieving the optimal trade-off between fairness level and clustering utility, for any given fairness level to be satisfied. First, in Theorem 4.1, we show that FCA-C provides the optimal clustering among all clusterings that satisfy a given fairness level. Then, in Corollary 4.2, we further show that Balance can be controlled by controlling $\epsilon$ in FCA-C. The proofs are deferred to Section B of Appendix.

For technical simplicity, we assume that the densities of $\mathbb{P}_0$ and $\mathbb{P}_1$ exist. That is, we consider the population version of FCA-C. See Remark B.1 in Section B of Appendix when the densities do not exist. Let $\mathbf{A}_\epsilon := \{(\mathcal{A}_0, \mathcal{A}_1) : \sum_{k=1}^{K} |\mathbb{E}_0(\mathcal{A}_0(\mathbf{X})_k) - \mathbb{E}_1(\mathcal{A}_1(\mathbf{X})_k)| \leq \epsilon\}$ for a given $\epsilon \geq 0$, represent a set of fair assignment functions. Let $\tilde{C}(\mathbb{Q}, \mathcal{W}, \boldsymbol{\mu})$ denotes the objective of FCA-C in eq. (5).

**Theorem 4.1** (FCA-C achieves the optimal trade-off). *Minimizing the FCA-C objective $\tilde{C}(\mathbb{Q}, \mathcal{W}, \boldsymbol{\mu})$ with the corresponding assignment function defined in eq. (6) is equivalent to minimizing $C(\boldsymbol{\mu}, \mathcal{A}_0, \mathcal{A}_1)$ subject to $(\mathcal{A}_0, \mathcal{A}_1) \in \mathbf{A}_\epsilon$.*

**Corollary 4.2.** *The assignment functions in $\mathbf{A}_\epsilon$ satisfies $|\text{Balance} - 1| \leq C\epsilon$, where $C = \max_{s,k} \frac{\pi_s}{\mathbb{E}(\mathcal{A}_s(\mathbf{X})_k)}$.*

We further establish the approximation guarantee (i.e., worst-case error) of FCA-C in Section B.5 of Appendix. To empirically validate Theorem 4.1, we conduct supporting experiments for FCA-C in Section 5, showing that the trade-off is effectively controlled by controlling $\epsilon$.

### 4.3 REDUCING COMPUTATIONAL COMPLEXITY

As discussed in Section 4.1, finding the joint distribution $\mathbb{Q}$ (i.e., the optimal coupling $\Gamma$) is technically equivalent to solving the Kantorovich problem, which involves linear programming (Villani, 2008). Its computational complexity is approximately $\mathcal{O}(n^3)$ when $n_0 = n_1 = n$ (Bonneel et al., 2011), indicating a high computational cost when $n$ is large. To address this issue in practice, we randomly split each group into $L$ partitions of (approximately) same sizes. That is, we have $L$ partitions for each group: $\{\mathcal{X}_0^{(l)}\}_{l=1}^{L}$ of $\mathcal{X}_0$ and $\{\mathcal{X}_1^{(l)}\}_{l=1}^{L}$ of $\mathcal{X}_1$. We then calculate the optimal coupling $\Gamma^{(l)}$ between $\mathcal{X}_0^{(l)}$ and $\mathcal{X}_1^{(l)}$, using eq. (4). The (full) optimal coupling between $\mathcal{X}_0$ and $\mathcal{X}_1$ is estimated by $\hat{\Gamma} := \text{diag}(\frac{1}{L}\Gamma^{(1)}, \ldots, \frac{1}{L}\Gamma^{(L)})$. Let $m = |\mathcal{X}_0^{(l)}| + |\mathcal{X}_1^{(l)}|$ be the partition size. This technique can theoretically reduce computational complexity by $\mathcal{O}(n/m) \times \mathcal{O}(m^3) = \mathcal{O}(nm^2)$. In our experiments, we apply this technique with $m = 1024$ for all datasets, which we find to be significantly faster than using full datasets, while yielding similar performance (see Section 5.4 for the results).

## 5 EXPERIMENTS

### 5.1 SETTINGS

**Datasets and performance measures** We use three benchmark tabular datasets, ADULT (Becker & Kohavi, 1996), BANK (Moro et al., 2012), and CENSUS (Meek et al.), from the UCI Machine Learning Repository[1] (Dua & Graff, 2017). The sensitive attribute is defined by gender (male/female), marital status (married/not married), and gender (male/female) for ADULT, BANK, and CENSUS, respectively. The number of clusters $K$ is set to 10 for ADULT and BANK, and 20 for CENSUS, following Ziko et al. (2021). The data (input features) are scaled to have zero mean and unit variance. We then optionally apply $L_2$ normalization, as used in Ziko et al. (2021), and results are provided both with and without $L_2$ normalization. Further details are given in Section C.1 of Appendix.

We consider two performance measures, Cost and Balance. The former assesses clustering utility, while the latter measures fairness level. For a given assignment function $\mathcal{A}$, let $\tilde{\mathcal{A}}_s(\mathbf{x})_k :=$

---

[1]https://archive.ics.uci.edu/datasets

$\mathbb{1}(\arg\max_{k'} \mathcal{A}_s(\mathbf{x})_{k'} = k)$ be a deterministic version of $\mathcal{A}$. Then, for given cluster centers $\boldsymbol{\mu} = \{\mu_1, \ldots, \mu_K\}$, the Cost and Balance on $\mathcal{D}$ are defined as $\text{Cost} = \frac{1}{n} \sum_{(\mathbf{x},s)\in\mathcal{D}} \|\mathbf{x} - \mu_{k(\mathbf{x},s)}\|^2$ and $\text{Balance} = \min_{k\in[K]} \min_{s\in\{0,1\}} \left( \frac{1}{n_s} \sum_{\mathbf{x}\in\mathcal{X}_s} \tilde{\mathcal{A}}_s(\mathbf{x})_k \big/ \frac{1}{n} \sum_{(\mathbf{x},s)\in\mathcal{D}} \tilde{\mathcal{A}}_s(\mathbf{x})_k \right)$, respectively, where $k(\mathbf{x}, s)$ is the index $k$ satisfying $\tilde{\mathcal{A}}_s(\mathbf{x})_k = 1$. Note that we use $\tilde{\mathcal{A}}$ instead of $\mathcal{A}$ itself for fair comparison since existing algorithms use deterministic assignment functions.

**Baseline algorithms**   For the baseline algorithms compared with FCA, we consider four methods: a pre-processing (fairlet-based) method SFC from Backurs et al. (2019), two post-processing methods FCBC from Esmaeili et al. (2021) and FRAC from Gupta et al. (2023), and an in-processing method VFC from Ziko et al. (2021), which differs from the other baselines as it is specifically designed to control the trade-off between fairness level Balance and clustering utility Cost. For implementation details of these baselines, refer to Section C.2 of Appendix.

**Implementation details**   When solving the linear program (i.e., finding the coupling matrix $\Gamma$), we use the POT library (Flamary et al., 2021). For finding cluster centers, we adopt the scikit-learn library (Pedregosa et al., 2011) to run $K$-means algorithm. We specifically choose the $K$-means++ (Arthur & Vassilvitskii, 2007) for $K$-means algorithm. An ablation study comparing the $K$-means++ and a gradient descent-based algorithm (i.e., Adam (Kingma & Ba, 2014)) for finding cluster centers is provided in Section 5.4. The computations are performed on several Intel Xeon Silver CPU cores and an additional RTX 4090 GPU processor. The maximum number of iterations is set to 100 for all cases, and we select the best iteration when Cost is minimized.

## 5.2 PERFORMANCE COMPARISON RESULTS

**Trade-off: fairness level vs. clustering utility**   First, we compare FC algorithms in terms of their ability to achieve the optimal trade-off between fairness level (Balance) and clustering utility (Cost). Specifically, we compare FCA with three baselines: SFC, FCBC, and FRAC, all of which are designed to achieve perfect fairness. Table 1 presents the Balance and Cost values of the four methods, where FCA consistently attains the lowest Cost (i.e., the highest clustering utility).

Table 1: Comparison of the trade-off between Balance and Cost on ADULT, BANK, and CENSUS datasets. We underline the Balance values for the cases of near-perfect fairness (i.e., Balance > 0.95) and use **bold face for the lowest Cost** value among those cases.

| C = Cost, B = Balance | ADULT | | BANK | | CENSUS | |
|---|---|---|---|---|---|---|
| **With $L_2$ normalization** | C ($\downarrow$) | B ($\uparrow$) | C ($\downarrow$) | B ($\uparrow$) | C ($\downarrow$) | B ($\uparrow$) |
| Standard (fair-unaware) | 0.295 | 0.451 | 0.208 | 0.500 | 0.403 | 0.025 |
| FCBC (Esmaeili et al., 2021) | 0.314 | 0.938 | 0.685 | 0.947 | 1.006 | 0.956 |
| SFC (Backurs et al., 2019) | 0.534 | 0.989 | 0.410 | 0.974 | 1.015 | 0.967 |
| FRAC (Gupta et al., 2023) | 0.340 | 0.998 | 0.307 | 0.995 | 0.537 | 0.997 |
| FCA ✓ | **0.328** | 0.997 | **0.264** | 0.998 | **0.477** | 0.993 |
| **Without $L_2$ normalization** | C ($\downarrow$) | B ($\uparrow$) | C ($\downarrow$) | B ($\uparrow$) | C ($\downarrow$) | B ($\uparrow$) |
| Standard (fair-unaware) | 1.620 | 0.418 | 1.510 | 0.602 | 28.809 | 0.031 |
| FCBC (Esmaeili et al., 2021) | 1.851 | 0.931 | 2.013 | 0.945 | 59.988 | 0.955 |
| SFC (Backurs et al., 2019) | 3.399 | 0.954 | 3.236 | 0.957 | 69.437 | 0.973 |
| FRAC (Gupta et al., 2023) | 2.900 | 0.998 | 2.716 | 0.995 | 38.430 | 0.997 |
| FCA ✓ | **1.875** | 0.997 | **1.859** | 0.998 | **33.472** | 0.990 |

Notably, FCA outperforms SFC by achieving higher Balance and lower Cost, highlighting the effectiveness of finding optimal matchings via in-processing. FCA is also superior to SFC for $K$-median clustering (Table 12 in Section C.3.8 of Appendix). Moreover, while FCA requires slightly more time (Table 10 in Section C.3.7 of Appendix), it still outperforms SFC even when the number of iteration is set to be comparable to SFC (Table 11 in Section C.3.7 of Appendix). FCBC and FRAC offer lower Balance and higher Cost in most cases (although FCBC shows slightly lower Cost on ADULT dataset, it fails to attain satisfactorily high Balance, remaining below 0.95). These results suggest that FCA is the most effective at maximizing utility under near-perfect fairness.

**Numerical stability** Second, we compare the two in-processing algorithms, FCA and VFC, in terms of their ability to achieve a high given fairness level (i.e., `Balance` $\approx 1$) without numerical instability. To do so, we obtain the maximum achievable `Balance` for each algorithm. We also evaluate the robustness to data pre-processing: the impact of $L_2$ normalization on numerical stability.

Table 2: Comparison of the two in-processing algorithms, FCA and VFC, in terms of numerical stability when achieving the maximum `Balance`, on ADULT, BANK, and CENSUS datasets. **Bold**-faced results indicate the highest values of `Balance`.

| | ADULT | BANK | CENSUS |
|---|---|---|---|
| With $L_2$ normalization | | `Balance`(`Cost`) | |
| VFC (Ziko et al., 2021) | 0.889 (0.310) | 0.875 (0.221) | 0.773 (0.432) |
| FCA ✓ | **0.997** (0.328) | **0.998** (0.264) | **0.993** (0.477) |
| Without $L_2$ normalization | | `Balance`(`Cost`) | |
| VFC (Ziko et al., 2021) | 0.629 (1.688) | 0.849 (1.549) | Failed |
| FCA ✓ | **0.997** (1.875) | **0.998** (1.859) | **0.990** (33.472) |

The results presented in Table 2 show that VFC achieves `Balance` values no higher than 0.9. While FCA is explicitly designed to achieve perfect fairness, VFC is designed to control `Balance` using a hyper-parameter. However, even with large hyper-parameter values, VFC fails to achieve perfect fairness (i.e., `Balance` remains significantly below 1). Moreover, the performance gap between FCA and VFC becomes greater without $L_2$ normalization than with it, and further fails on CENSUS dataset due to an overflow (see Section C.3.1 of Appendix for details on this failure). For an additional comparison on a larger dataset ($n = 10^6$), see Section C.3.9 of Appendix, which highlights FCA's superior scalability and outperformance over VFC.

**Control of Balance** In addition to the previous analysis of the scenario where `Balance` $\approx 1$, we also compare FCA-C and VFC in terms of controlling `Balance` across various levels lower than 1. To this end, we assess the ability to achieve reasonable `Cost` while controlling `Balance`. Figure 3 displays the performance of FCA-C and VFC across various fairness levels. It shows that FCA-C effectively manages this trade-off across the wide range of `Balance` (by controlling $\epsilon$), with `Cost` similar to that of VFC. The orange dashed vertical line is the maximum achievable `Balance` by VFC, and FCA-C achieves `Balance` beyond this point well. Refer to Section C.2 of Appendix for details on selecting $\epsilon$ for FCA and the hyper-parameters used in VFC. In addition, Table 9 in Section C.3.7 of Appendix compares the computation times of FCA-C and FCA.

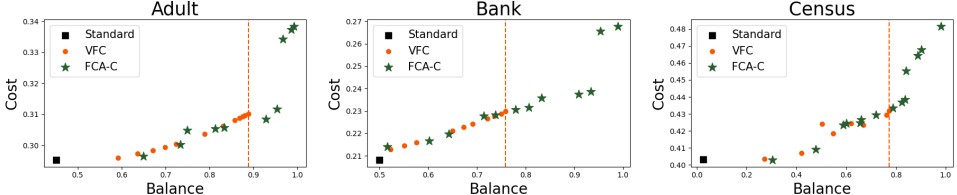

Figure 3: `Balance` vs. `Cost` trade-offs for (left) ADULT, (center) BANK, and (right) CENSUS datasets. Black squares (■) are from the standard clustering, orange circles (●) are from VFC, green stars (★) are from FCA-C, and orange dashed lines (- -) are the maximums of `Balance` that VFC can achieve. For similar results without $L_2$ normalization, see Figure 4 in Section C.3.1 of Appendix.

### 5.3 APPLICABILITY TO VISUAL CLUSTERING

We further evaluate FCA's applicability to visual clustering using two image datasets: (i) REVERSE MNIST, a mixture of the original MNIST and a color-reversed version (in which black and white are swapped), and (ii) OFFICE-31, consisting of images from two domains (amazon and webcam) with 31 classes, commonly used in visual FC methods (Li et al., 2020; Zeng et al., 2023).

We compare FCA with SFC, VFC and two visual FC baselines: DFC (Li et al., 2020) and FCMI (Zeng et al., 2023). Note that DFC and FCMI are end-to-end algorithms that simultaneously perform

clustering and learn latent space using autoencoder with fairness constraints. In contrast, FCA, SFC, and VFC are applied to a latent space pre-trained by autoencoder without any fairness constraints. The clustering utility is evaluated using `ACC` (accuracy calculated based on assigned cluster indices and ground-truth labels) and `NMI` (normalized mutual information between ground-truth label distribution and assigned cluster distribution), as in Zeng et al. (2023).

Table 3: Comparison of clustering utility (`ACC` and `NMI`) and fairness level (`Balance`) on two image datasets. 'Standard (fair-unaware)' indicates autoencoder + $K$-means. First-place values are **bold**, and second-place values are underlined. The performances of baselines reflect the best results from our re-implementations or those from Zeng et al. (2023).

| Dataset | ‖ | REVERSE MNIST | | ‖ | OFFICE-31 | | |
|---|---|---|---|---|---|---|---|
| A = ACC, N = NMI, B = Balance ‖ | A (↑) | N (↑) | B (↑) ‖ | A (↑) | N (↑) | B (↑) |
| Standard (fair-unaware) | 41.0 | 52.8 | 0.000 | 63.8 | 66.8 | 0.701 |
| SFC (Backurs et al., 2019) | 51.3 | 49.1 | **1.000** | 61.6 | 61.2 | 0.932 |
| VFC (Ziko et al., 2021) | 38.1 | 42.7 | 0.000 | 64.8 | 70.4 | 0.826 |
| DFC (Li et al., 2020) | 49.9 | 68.9 | 0.800 | 69.0 | 70.9 | 0.584 |
| FCMI (Zeng et al., 2023) | 88.4 | **86.4** | 0.998 | **70.0** | **71.2** | 0.926 |
| FCA ✓ | **89.0** | 79.0 | 0.999 | 67.6 | 70.5 | **0.967** |

Table 3 presents the comparison results, showing that FCA performs similarly to the state-of-the-art method FCMI, while mostly outperforming the other three baselines – SFC, VFC, and DFC. Although DFC achieves slightly higher `ACC` and `NMI` values on OFFICE-31 dataset, its `Balance` is significantly lower (0.584), while FCA achieves the highest (0.967). These results are notable because FCA operates as a two-step approach (i.e., pre-training the latent space and then performing FCA on the pre-trained latent space) for visual clustering, whereas DFC and FCMI are end-to-end methods.

Moreover, FCA offers practical advantages: (i) it requires fewer hyper-parameters than the end-to-end methods, which would face challenges in tuning hyper-parameters based on `ACC` or `NMI` in practice, due to the absence of ground-truth labels during training; and (ii) it can leverage any pre-trained latent space, enhancing adaptability. Further details are provided in Section C.3.2 of Appendix, including Table 5 comparing the fairlet-based method and FCA in visual clustering.

## 5.4 ABLATION STUDIES

**(1) Selection of the partition size** $m$**:** We empirically validate the efficacy of the partitioning technique in Section 4.3, by investigating the convergence of `Balance` and `Cost` with respect to the partition size $m$, by varying $m \in \{256, 512, 1024, 2048, 4096, \text{Full}\}$. In Section C.3.3 of Appendix, Figure 5 suggests that $m = 1024$ yields reasonable results, while Table 6 shows a significant reduction in computation time. Moreover, Figure 6 confirms that computation time increases linearly in $m^2$, as discussed in Section 4.3. **(2) Optimization algorithm and initialization of cluster centers:** We also conduct additional ablation studies on the stability with respect to (i) the choice of optimization algorithm for finding cluster centers, and (ii) the initialization of cluster centers. Refer to Sections C.3.4 and C.3.5 of Appendix, which confirm that FCA is robust to both factors. **(3) Varying** $K$**:** We further confirm the uniform effectiveness of FCA with respect to $K$. As shown in Figure 7 in Section C.3.6 of Appendix, FCA outperforms the baseline methods across all $K \in \{5, 10, 20, 40\}$.

## 6 CONCLUSION AND DISCUSSION

This paper has proposed FCA, an in-processing algorithm for fair clustering. FCA is motivated by the theoretical result that finding the optimal perfectly fair clustering is equivalent to finding centers in the aligned space. FCA algorithm is based on two well-known algorithms, the $K$-means++ algorithm and linear programming, making it transparent and stable. We have empirically demonstrated that FCA achieves near-perfect fairness and superior clustering utility, when compared to existing methods. Moreover, FCA is robust to the choice of data pre-processing method, optimization algorithm, and initialization. Additionally, we developed FCA-C, a variant of FCA, to control Balance.

Possible directions for future work are extending FCA for multiple protected groups (see Section A.3 of Appendix for a feasible approach), or applying FCA to other clustering algorithms such as Gaussian mixture, which we will pursue in the near future.

## REPRODUCIBILITY STATEMENT

The full proofs of the theoretical results are rigorously presented in Section B of Appendix. Key details regarding the experimental implementation, including the datasets, libraries, and hyper-parameters used, are outlined in Section 5.1. Further information on the datasets and algorithm implementations can be found in Sections C.1 and C.2 for the three tabular datasets, and in Section C.3.2 for the two image datasets, of Appendix. The source code will be made publicly available after acceptance.

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

# A SUPPLEMENTARY DISCUSSION

## A.1 FAIRLET-BASED METHODS

Fairlet decomposition was first introduced in Chierichetti et al. (2017), providing a pre-processing method for fair clustering. A fairlet is defined as a subset (as small as possible) of data in which the proportion of the sensitive attribute is balanced within all subsets. It is important to note that the fairlets are not built arbitrarily; instead, the sum of distances among instances within each fairlet is minimized (i.e., each fairlet consists of similar instances). Finding such fairlets can be done by solving the minimum cost flow problem.

Notably, when $n_0 = n_1$, building fairlets is equivalent to finding the optimal coupling $\Gamma$ in the Kantorovich problem, which can be understood as a special case of minimum cost problem (Peyré & Cuturi, 2020; Chen et al., 2022). Hence, we can say that the fairlet-based methods find the coupling *only once*, then apply the standard clustering algorithm. On the contrary, our proposed approach jointly optimizes both the coupling (i.e., the matching map $\mathbf{T}$) and the cluster centers, to ensure that they correspond to each other optimally.

After building the fairlets, each fairlet is then deterministically assigned to a cluster. Since fairness is implicitly guaranteed within each fairlet, the resulting clustering directly becomes also fair. The representatives of each fairlet can be arbitrarily chosen when $n_0 = n_1$, or chosen by the medoids (or possibly the centroids) when $n_0 \neq n_1$, as suggested in Chierichetti et al. (2017). Then, a standard clustering algorithm is applied to the set of these chosen representatives. However, these choices of representatives result in approximation and suboptimality in view of clustering utility.

On the other hand, the computational cost is high, with most of the time spent finding the fairlets due to the quadratic complexity of the minimum cost flow problem. To address this issue, SFC (Backurs et al., 2019) has proposed a scalable algorithm for fairlet decomposition by using a reduction approach using metric embedding and trees. See Section C.2.2 for details on the SFC algorithm.

## A.2 OPTIMAL TRANSPORT PROBLEM

The notion of Optimal Transport (OT) provides a geometric view of the discrepancy between two probability measures. For two given probability measures $\mathcal{P}_1$ and $\mathcal{P}_2$, a map $\mathbf{T} : \mathrm{Supp}(\mathcal{P}_1) \rightarrow \mathrm{Supp}(\mathcal{P}_2)$ is defined as 'transport map' if $\mathbf{T}_{\#}\mathcal{P}_1 = \mathcal{P}_2$, where $\mathbf{T}_{\#}\mathcal{P}_1(A) = \mathcal{P}_1(\mathbf{T}^{-1}(A)), \forall A$, is the push-forward measure. The OT map is the minimizer of transport cost among all transport maps. That is, the OT map from $\mathcal{P}_1$ to $\mathcal{P}_2$ is the solution of $\min_{\mathbf{T}:\mathbf{T}_{\#}\mathcal{P}_1=\mathcal{P}_2} \mathbb{E}_{\mathbf{X}\sim\mathcal{P}_1}\left(c\left(\mathbf{X}, \mathbf{T}(\mathbf{X})\right)\right)$ for a pre-specified cost function $c$ (e.g., $L_2$ distance), which is so-called the Monge problem.

Kantorovich relaxed the Monge problem by seeking the optimal coupling (joint distribution) between two distributions. The Kantorovich problem is mathematically formulated as $\inf_{\pi\in\Pi(\mathcal{P}_1,\mathcal{P}_2)} \mathbb{E}_{\mathbf{X},\mathbf{Y}\sim\pi}\left(c(\mathbf{X}, \mathbf{Y})\right)$ where $\Pi(\mathcal{P}_1,\mathcal{P}_2)$ is the set of all joint measures of $\mathcal{P}_1$ and $\mathcal{P}_2$. For two empirical measures, this problem can be solved by the use of linear programming as follows. For given two empirical distributions on $\mathcal{X}_0 = \{\mathbf{x}_i^{(0)}\}_{i=1}^{n_0}$ and $\mathcal{X}_1 = \{\mathbf{x}_j^{(1)}\}_{j=1}^{n_1}$, the cost matrix between the two is defined by $\mathbf{C} := [c_{i,j}] \in \mathbb{R}_+^{n_0 \times n_1}$ where $c_{i,j} = \|\mathbf{x}_i^{(0)} - \mathbf{x}_j^{(1)}\|^2$. Then, the optimal joint distribution is defined by the matrix $\Gamma = [\gamma_{i,j}] \in \mathbb{R}_+^{n_0 \times n_1}$ that solves the following objective:

$$\min_{\Gamma} \|\mathbf{C} \odot \Gamma\|_1 = \min_{\gamma_{i,j}} c_{i,j}\gamma_{i,j} \text{ s.t. } \sum_{i=1}^{n_0} \gamma_{i,j} = \frac{1}{n_1}, \sum_{j=1}^{n_1} \gamma_{i,j} = \frac{1}{n_0}, \gamma_{i,j} \geq 0. \tag{7}$$

This problem can be solved by the use of linear programming. For the case of large $n$ with $n_0 \neq n_1$, various feasible estimators have been developed (Cuturi, 2013; Genevay et al., 2016), e.g., Sinkhorn algorithm (Cuturi, 2013). Note only practical implementations, but theoretical aspects such as minimax estimation have also discussed deeply (Deb et al., 2021; Hütter & Rigollet, 2021; Seguy et al., 2018; Yang & Uhler, 2019). Recently, the OT map is utilized in diverse domains, for example, economics (Galichon, 2016; Chiappori et al., 2010), domain adaptation (Damodaran et al., 2018; Forrow et al., 2019), and computer vision (Su et al., 2015; Salimans et al., 2018). Several studies, including Jiang et al. (2020); Chzhen et al. (2020); Gordaliza et al. (2019), have also employed the OT theory in the field of algorithmic fairness for supervised learning.

### A.3 EXTENSION TO MULTIPLE PROTECTED GROUPS

In this paper, we mainly focus on the case of two protected groups, for ease of discussion. However, it is possible to extend FCA to handle multiple (more than two) protected groups. The idea is equivalent to the case of two protected groups: matching multiple individuals from different protected groups.

**The case of equal sample sizes** Let $G$ be the number of protected groups, and denote $s^* \in \{0, \ldots, G-1\}$ as a fixed (reference) sensitive attribute. For the case of $n_0 = \ldots = n_{G-1}$, we can similarly decompose the objective function in terms of matching map, as follows. First, we decompose the clustering objective similar to the proof of Theorem 3.1:

$$C(\boldsymbol{\mu}, \mathcal{A}_0, \ldots, \mathcal{A}_{G-1}) = \mathbb{E} \sum_{k=1}^{K} \mathcal{A}_S(\mathbf{X}) \|\mathbf{X} - \mu_k\|^2$$

$$= \frac{1}{G} \mathbb{E}_{s^*} \sum_{k=1}^{K} \mathcal{A}_{s^*}(\mathbf{X})_k \left( \|\mathbf{X} - \mu_k\|^2 + \sum_{s' \neq s^*} \|\mathbf{T}_{s'}(\mathbf{X}) - \mu_k\|^2 \right), \tag{8}$$

where $\mathbf{T}_{s'}$ is the one-to-one matching map from $\mathcal{X}_{s^*}$ to $\mathcal{X}_{s'}$ for all $s' \in \{0, \ldots, G-1\}$. Note that $\mathbf{T}_{s'}$ then becomes the identity map from $\mathcal{X}_{s^*}$ to $\mathcal{X}_{s^*}$ for $s' = s^*$. Then for any given $\mathbf{x}$ and $\mu_k$, we can decompose $\|\mathbf{x} - \mu_k\|^2$ by

$$\left\| \frac{\sum_{s' \neq s^*}(\mathbf{x} - \mathbf{T}_{s'}(\mathbf{x}))}{G} \right\|^2 + \left\| \frac{\mathbf{x} + \sum_{s' \neq s^*} \mathbf{T}_{s'}(\mathbf{x})}{G} - \mu_k \right\|^2$$

$$+ 2 \left\langle \frac{\sum_{s' \neq s^*}(\mathbf{x} - \mathbf{T}_{s'}(\mathbf{x}))}{G}, \frac{\mathbf{x} + \sum_{s \neq s^*} \mathbf{T}_{s'}(\mathbf{x})}{G} - \mu_k \right\rangle. \tag{9}$$

We similarly decompose $\|\mathbf{T}_{s'}(\mathbf{x}) - \mu_k\|^2$ by

$$\left\| \frac{\sum_{s'' \neq s', s'' \in \{0, \ldots, G-1\}}(\mathbf{T}_{s'}(\mathbf{x}) - \mathbf{T}_{s''}(\mathbf{x}))}{G} \right\|^2 + \left\| \frac{\mathbf{x} + \sum_{s' \neq s^*} \mathbf{T}_{s'}(\mathbf{x})}{G} - \mu_k \right\|^2$$

$$+ 2 \left\langle \frac{\sum_{s'' \neq s', s'' \in \{0, \ldots, G-1\}}(\mathbf{T}_{s'}(\mathbf{x}) - \mathbf{T}_{s''}(\mathbf{x}))}{G}, \frac{\mathbf{x} + \sum_{s' \neq s^*} \mathbf{T}_{s'}(\mathbf{x})}{G} - \mu_k \right\rangle. \tag{10}$$

By summing up the above $G$ many terms, we have the final result that

$$C(\boldsymbol{\mu}, \mathcal{A}_0, \ldots, \mathcal{A}_{G-1})$$

$$= \frac{1}{G} \mathbb{E}_{s^*} \sum_{k=1}^{K} \mathcal{A}_{s^*}(\mathbf{X})_k \left( \sum_{s=0}^{G-1} \left\| \frac{\sum_{s' \neq s}(\mathbf{T}_s(\mathbf{X}) - \mathbf{T}_{s'}(\mathbf{X}))}{G} \right\|^2 + \left\| \frac{\mathbf{X} + \sum_{s' \neq s^*} \mathbf{T}_{s'}(\mathbf{X})}{G} - \mu_k \right\|^2 \right). \tag{11}$$

Note that this result holds for any $s^* \in \{0, \ldots, G-1\}$.

**The case of unequal sample sizes** For the case of unequal sample sizes, we also derive a similar decomposition (i.e., eq. (12)). We first define the alignment map as: $\mathbf{T}^{\mathrm{A}}(\mathbf{x}_0, \ldots, \mathbf{x}_{G-1}) := \pi_0 \mathbf{x}_0 + \ldots + \pi_{G-1} \mathbf{x}_{G-1}$. Then, we find the joint distribution and cluster centers by minimizing the following objective:

$$\mathbb{E}_{(\mathbf{X}_0, \ldots, \mathbf{X}_{G-1}) \sim \mathbb{Q}} \left( \sum_{s=0}^{G-1} G \pi^G \left\| \sum_{s' \neq s} (\mathbf{X}_s - \mathbf{X}_{s'}) \right\|^2 + \min_k \| \mathbf{T}^{\mathrm{A}}(\mathbf{X}_0, \ldots, \mathbf{X}_{G-1}) - \mu_k \|^2 \right), \quad (12)$$

where $\pi^G := \prod_{s=0}^{G-1} \pi_s$.

The proof idea is similar to the case of two protected groups, in Theorem 3.3. We first recall the original $K$-means objective: $C(\boldsymbol{\mu}, \mathcal{A}_0, \ldots, \mathcal{A}_{G-1}) := \mathbb{E} \sum_{k=1}^{K} \mathcal{A}_S(\mathbf{X})_k \| \mathbf{X} - \mu_k \|^2 = \sum_{s=0}^{G-1} \pi_s \mathbb{E}_s \sum_{k=1}^{K} \mathcal{A}_s(\mathbf{X}_s)_k \| \mathbf{X}_s - \mu_k \|^2$. Consider a set of joint distributions of $\mathbf{X}_0, \ldots, \mathbf{X}_{G-1}$ given $\boldsymbol{\mu}$ as $\mathcal{Q} := \{ \mathbb{Q}(\{\mathbf{x}_0, \ldots, \mathbf{x}_{G-1}\}|\boldsymbol{\mu}) : \mathbf{x}_s \in \mathcal{X}_s, s \in \{0, \ldots, G-1\} \}$. Then, we can show that there exists a $\mathbb{Q} = \mathbb{Q}(\{\mathbf{x}_0, \ldots, \mathbf{x}_{G-1}\}|\boldsymbol{\mu}) \in \mathcal{Q}$ satisfying

$$C(\boldsymbol{\mu}, \mathcal{A}_0, \ldots, \mathcal{A}_{G-1}) = \sum_{s=0}^{G-1} \pi_s \mathbb{E}_s \sum_{k=1}^{K} \mathcal{A}_s(\mathbf{X}_s)_k \| \mathbf{X}_s - \mu_k \|^2$$

$$= \mathbb{E}_{(\mathbf{X}_0, \ldots, \mathbf{X}_{G-1}) \sim \mathbb{Q}} \left( \sum_{s=0}^{G-1} \pi_s \| \mathbf{X}_s - \mu_k \|^2 \right), \quad (13)$$

by using the same logic in the proof of Theorem 3.3. Furthermore, we can reformulate it as

$$\mathbb{E}_{(\mathbf{X}_0, \ldots, \mathbf{X}_{G-1}) \sim \mathbb{Q}} \left( \sum_{s=0}^{G-1} G \pi^G \left\| \sum_{s' \neq s} (\mathbf{X}_s - \mathbf{X}_{s'}) \right\|^2 + \min_k \| \mathbf{T}^{\mathrm{A}}(\mathbf{X}_0, \ldots, \mathbf{X}_{G-1}) - \mu_k \|^2 \right), \quad (14)$$

with the assignment functions for fair clustering given as

$$\mathcal{A}_s(\mathbf{x}_s)_k := \mathbb{Q} \left( \arg\min_{k'} \| \pi_s \mathbf{x}_s + \sum_{s' \neq s} \pi_{s'} \mathbf{X}_{s'} - \mu_{k'} \|^2 = k \middle| \mathbf{X}_s = \mathbf{x}_s \right), \forall s \in \{0, \ldots, G-1\}. \quad (15)$$

Furthermore, since finding the joint distribution for multiple protected groups is technically similar to the case of two protected groups, the optimization of the objective (12) can be solved using a linear program.

## A.4 EXTENSION TO $K$-CLUSTERING WITH THE $L_p$ NORM ($p \geq 1$)

For a given $K$-clustering with the $L_p$ norm ($p \geq 1$), we can derive a decomposition result similar to the $K$-means clustering problem, which is mainly discussed in this paper.

The objective of $K$-clustering for the $L_p$ norm ($p \geq 1$) is given by $\mathbb{E} \sum_{k=1}^{K} \mathcal{A}_S(\mathbf{X})_k \|\mathbf{X} - \mu_k\|_p$. For the simple case where $n_0 = n_1$, using the triangle inequality, we can derive an upper bound for the objective similar to Theorem 3.1:

$$\mathbb{E} \sum_{k=1}^{K} \mathcal{A}_S(\mathbf{X})_k \|\mathbf{X} - \mu_k\|_p \leq \mathbb{E}_s \sum_{k=1}^{K} \mathcal{A}_S(\mathbf{X})_k \left( \left\| \frac{\mathbf{X} - \mathbf{T}(\mathbf{X})}{2} \right\|_p + \left\| \frac{\mathbf{X} + \mathbf{T}(\mathbf{X})}{2} - \mu_k \right\|_p \right).$$

Hence, we can minimize this upper bound of the fair $K$-clustering objective for the $L_p$ norm. Note that using the $L_1$ norm corresponds to the $K$-median clustering.

See Section C.3.8 for experiments based on this approach, showing the outperformance FCA over SFC in view of the $K$-median clustering.

## A.5 OTHER FAIRNESS NOTIONS IN CLUSTERING

Apart from the group fairness discussed in this paper, several other fairness notions have been explored in clustering problems.

**Proportional fairness** *Proportional fairness*, initially introduced by Chen et al. (2019), is a fairness notion that ensures that cluster sizes are proportionally balanced. It aims to find a clustering, where any group of at least $n/K$ data points should not be able to form a new cluster center that improves the clustering cost for all points in the group. The primary focus in proportionally fair clustering is to study the approximation errors. For example, Micha & Shah (2020) provided a better approximation error than that of the algorithm from Chen et al. (2019), while Aziz et al. (2024) showed that there exists a proportionally fair clustering that achieves a $(1 + \sqrt{2})$-approximation.

**Individual fairness** *Individual fairness*, which was initially introduced by Jung et al. (2019), is a fairness notion that ensures that every data point has an assigned cluster center within the radius of the smallest ball around the data point containing at least $n/K$ points. Initial studies by Jung et al. (2019) focused on approximation guarantees for individually fair clustering. More recently, Mahabadi & Vakilian (2020) studied more general approximation errors, while Han et al. (2022) considered a specific scenario where outliers exist in given data.

Notably, there have been several efforts to explore the relationships between different fairness notions in clustering. For example, Kellerhals & Peters (2024) demonstrated that algorithms for individual fairness and proportional fairness can be compatible for some sense. We believe that investigating further connections among various fairness notions including group, proportional, and individual fairness would be a promising direction for future research.

# B PROOFS OF THE THEOREMS

## B.1 PROOF OF THEOREM 3.1

**Theorem 3.1** For any given perfectly fair deterministic assignment function $\mathcal{A}$ and cluster centers $\boldsymbol{\mu}$, there exists a one-to-one matching map $\mathbf{T} : \mathcal{X}_s \to \mathcal{X}_{s'}$ such that, for any $s \in \{0, 1\}$,

$$C(\boldsymbol{\mu}, \mathcal{A}_0, \mathcal{A}_1) = \mathbb{E}_s \sum_{k=1}^{K} \mathcal{A}_s(\mathbf{X})_k \left( \frac{\|\mathbf{X} - \mathbf{T}(\mathbf{X})\|^2}{4} + \left\| \frac{\mathbf{X} + \mathbf{T}(\mathbf{X})}{2} - \mu_k \right\|^2 \right). \quad (16)$$

*Proof of Theorem 3.1.* Without loss of generality, let $s = 0$. First, it is clear that we can construct a one-to-one map $\mathbf{T}$ that maps each $\mathbf{x} \in \mathcal{X}^{k,0} := \{\mathbf{x} \in \mathcal{X}_0 : \mathcal{A}_0(\mathbf{x})_k = 1\}$ to a unique $\mathbf{x}' \in \mathcal{X}^{k,1} := \{\mathbf{x}' \in \mathcal{X}_1 : \mathcal{A}_1(\mathbf{x}')_k = 1\}$ for all $k \in [K]$. That is, $\{\mathbf{T}(\mathbf{x}) : \mathbf{x} \in \mathcal{X}^{k,0}\} = \mathcal{X}^{k,1}, \forall k \in [K]$ and $\{\mathbf{T}(\mathbf{x}) : \mathbf{x} \in \mathcal{X}_0\} = \mathcal{X}_1$.

Then, for the given $\mathbf{T}$, we can rewrite the clustering cost as

$$C(\boldsymbol{\mu}, \mathcal{A}_0, \mathcal{A}_1) = \mathbb{E} \sum_{k=1}^{K} \mathcal{A}_S(\mathbf{X})_k \|\mathbf{X} - \mu_k\|^2 = \frac{1}{2} \mathbb{E}_0 \sum_{k=1}^{K} \mathcal{A}_0(\mathbf{X})_k \left( \|\mathbf{X} - \mu_k\|^2 + \|\mathbf{T}(\mathbf{X}) - \mu_k\|^2 \right). \quad (17)$$

For any given $\mathbf{x}$ and $\mu_k$, we decompose $\|\mathbf{x} - \mu_k\|^2$ as:

$$\|\mathbf{x} - \mu_k\|^2 = \left\| \mathbf{x} - \frac{\mathbf{x} + \mathbf{T}(\mathbf{x})}{2} + \frac{\mathbf{x} + \mathbf{T}(\mathbf{x})}{2} - \mu_k \right\|^2$$

$$= \frac{\|\mathbf{x} - \mathbf{T}(\mathbf{x})\|^2}{4} + \left\| \frac{\mathbf{x} + \mathbf{T}(\mathbf{x})}{2} - \mu_k \right\|^2 + 2 \left\langle \mathbf{x} - \frac{\mathbf{x} + \mathbf{T}(\mathbf{x})}{2}, \frac{\mathbf{x} + \mathbf{T}(\mathbf{x})}{2} - \mu_k \right\rangle.$$

We similarly decompose $\|\mathbf{T}(\mathbf{x}) - \mu_k\|^2$ as:

$$\|\mathbf{T}(\mathbf{x}) - \mu_k\|^2 = \left\| \mathbf{T}(\mathbf{x}) - \frac{\mathbf{x} + \mathbf{T}(\mathbf{x})}{2} + \frac{\mathbf{x} + \mathbf{T}(\mathbf{x})}{2} - \mu_k \right\|^2$$

$$= \frac{\|\mathbf{x} - \mathbf{T}(\mathbf{x})\|^2}{4} + \left\| \frac{\mathbf{x} + \mathbf{T}(\mathbf{x})}{2} - \mu_k \right\|^2 + 2 \left\langle \mathbf{T}(\mathbf{x}) - \frac{\mathbf{x} + \mathbf{T}(\mathbf{x})}{2}, \frac{\mathbf{x} + \mathbf{T}(\mathbf{x})}{2} - \mu_k \right\rangle.$$

Adding the two terms, we have

$$2 \left( \frac{\|\mathbf{x} - \mathbf{T}(\mathbf{x})\|^2}{4} + \left\| \frac{\mathbf{x} + \mathbf{T}(\mathbf{x})}{2} - \mu_k \right\|^2 \right).$$

Finally, we conclude that

$$\frac{1}{2} \mathbb{E}_0 \sum_{k=1}^{K} \mathcal{A}_0(\mathbf{X})_k \left( \|\mathbf{X} - \mu_k\|^2 + \|\mathbf{T}(\mathbf{X}) - \mu_k\|^2 \right)$$

$$= \frac{1}{2} \mathbb{E}_0 \sum_{k=1}^{K} \mathcal{A}_0(\mathbf{X})_k \cdot 2 \left( \frac{\|\mathbf{X} - \mathbf{T}(\mathbf{X})\|^2}{4} + \left\| \frac{\mathbf{X} + \mathbf{T}(\mathbf{X})}{2} - \mu_k \right\|^2 \right). \quad (18)$$

$\square$

### B.2 PROOF OF THEOREM 3.3

**Theorem 3.3** Let $\boldsymbol{\mu}^*$ and $\mathbb{Q}^*$ be the cluster centers and joint distribution solving

$$\min_{\boldsymbol{\mu}, \mathbb{Q} \in \mathcal{Q}} \mathbb{E}_{(\mathbf{X}_0, \mathbf{X}_1) \sim \mathbb{Q}} \left( 2\pi_0 \pi_1 \|\mathbf{X}_0 - \mathbf{X}_1\|^2 + \min_k \|\mathbf{T}^A(\mathbf{X}_0, \mathbf{X}_1) - \mu_k\|^2 \right), \tag{19}$$

where $\mathbf{T}^A(\mathbf{x}_0, \mathbf{x}_1) = \pi_0 \mathbf{x}_0 + \pi_1 \mathbf{x}_1$, which is called the alignment map.
Then, $(\boldsymbol{\mu}^*, \mathcal{A}_0^*, \mathcal{A}_1^*)$ is the solution of the perfectly fair $K$-means clustering, where $\mathcal{A}_0^*(\mathbf{x})_k :=$
$\mathbb{Q}^* \left( \arg \min_{k'} \|\mathbf{T}^A(\mathbf{x}, \mathbf{X}_1) - \mu_{k'}\|^2 = k | \mathbf{X}_0 = \mathbf{x} \right)$ and $\mathcal{A}_1^*(\mathbf{x})_k$ is defined similarly.

*Proof of Theorem 3.3.* Let $\mathbf{X}_s = \mathbf{X}|S = s, s \in \{0, 1\}$. For given $(\boldsymbol{\mu}, \mathcal{A}_0, \mathcal{A}_1)$, recall the original $K$-means objective: $C(\boldsymbol{\mu}, \mathcal{A}_0, \mathcal{A}_1) := \mathbb{E} \sum_{k=1}^K \mathcal{A}_S(\mathbf{X})_k \|\mathbf{X} - \mu_k\|^2 = \pi_0 \mathbb{E}_0 \sum_{k=1}^K \mathcal{A}_0(\mathbf{X}_0)_k \|\mathbf{X}_0 - \mu_k\|^2 + \pi_1 \mathbb{E}_1 \sum_{k=1}^K \mathcal{A}_1(\mathbf{X}_1)_k \|\mathbf{X}_1 - \mu_k\|^2$.

Consider a set of joint distributions of $\mathbf{X}_0, \mathbf{X}_1$ given $\boldsymbol{\mu}$ as $\mathcal{Q} := \{\mathbb{Q}(\{\mathbf{x}_0, \mathbf{x}_1\}|\boldsymbol{\mu}) : \mathbf{x}_0 \in \mathcal{X}_0, \mathbf{x}_1 \in \mathcal{X}_1\}$. We show that there exists a $\mathbb{Q} = \mathbb{Q}(\{\mathbf{x}_0, \mathbf{x}_1\}|\boldsymbol{\mu}) \in \mathcal{Q}$ satisfying

$$C(\boldsymbol{\mu}, \mathcal{A}_0, \mathcal{A}_1) = \pi_0 \mathbb{E}_0 \sum_{k=1}^K \mathcal{A}_0(\mathbf{X}_0)_k \|\mathbf{X}_0 - \mu_k\|^2 + \pi_1 \mathbb{E}_1 \sum_{k=1}^K \mathcal{A}_1(\mathbf{X}_1)_k \|\mathbf{X}_1 - \mu_k\|^2 \\ = \mathbb{E}_{(\mathbf{X}_0, \mathbf{X}_1) \sim \mathbb{Q}} (\pi_0 \|\mathbf{X}_0 - \mu_k\|^2 + \pi_1 \|\mathbf{X}_1 - \mu_k\|^2). \tag{20}$$

Let

$$\mathbb{Q}(\{\mathbf{x}_0, \mathbf{x}_1\}|\boldsymbol{\mu}) = \sum_{k=1}^K \frac{\mathcal{A}_0(\mathbf{x}_0)_k \mathcal{A}_1(\mathbf{x}_1)_k}{\mathcal{C}_k} \mathbb{P}_0(\{\mathbf{x}_0\}) \mathbb{P}_1(\{\mathbf{x}_1\}), \tag{21}$$

where $\mathcal{C}_k := \mathbb{E} \mathcal{A}_S(\mathbf{X})_k = \mathbb{E}_0 \mathcal{A}_0(\mathbf{X}_0)_k = \mathbb{E}_1 \mathcal{A}_1(\mathbf{X}_1)_k$. Then,

$$\mathbb{E}_{(\mathbf{X}_0, \mathbf{X}_1) \sim \mathbb{Q}} (\pi_0 \|\mathbf{X}_0 - \mu_k\|^2 + \pi_1 \|\mathbf{X}_1 - \mu_k\|^2) \\ = \sum_{k=1}^K \left( \sum_{\mathbf{x}_0} \mathcal{A}_0(\mathbf{x}_0)_k \pi_0 \|\mathbf{x}_0 - \mu_k\|^2 \mathbb{P}_0(\{\mathbf{x}_0\}) + \sum_{\mathbf{x}_1} \mathcal{A}_1(\mathbf{x}_1)_k \pi_1 \|\mathbf{x}_1 - \mu_k\|^2 \mathbb{P}_1(\{\mathbf{x}_1\}) \right) \\ = \sum_{k=1}^K \left( \mathbb{E}_0 \pi_0 \mathcal{A}_0(\mathbf{X}_0)_k \|\mathbf{X}_0 - \mu_k\|^2 + \mathbb{E}_1 \pi_1 \mathcal{A}_1(\mathbf{X}_1)_k \|\mathbf{X}_1 - \mu_k\|^2 \right) = C(\boldsymbol{\mu}, \mathcal{A}_0, \mathcal{A}_1), \tag{22}$$

which concludes eq. (20). Our original aim is to find $(\boldsymbol{\mu}, \mathcal{A}_0, \mathcal{A}_1)$ minimizing $C(\boldsymbol{\mu}, \mathcal{A}_0, \mathcal{A}_1)$. Let $\mu(\mathbf{x}) = \mu_{k^*}$, where $k^* = \arg \min_k \|\mathbf{x} - \mu_k\|^2$. Let $\tilde{\mu}(\mathbf{x}_0, \mathbf{x}_1) := \mu_{k'}$, where $k' = \arg \min_k \|\pi_0 \mathbf{x}_0 + \pi_1 \mathbf{x}_1 - \mu_k\|^2$. For given $\mathbb{Q}$ defined in eq. (21), similar to Theorem 3.1, we can reformulate

$$\mathbb{E}_{(\mathbf{X}_0, \mathbf{X}_1) \sim \mathbb{Q}} (\pi_0 \|\mathbf{X}_0 - \mu(\mathbf{X}_0)\|^2 + \pi_1 \|\mathbf{X}_1 - \mu(\mathbf{X}_1)\|^2) \\ = \mathbb{E}_{(\mathbf{X}_0, \mathbf{X}_1) \sim \mathbb{Q}} \left( 2\pi_0 \pi_1 \|\mathbf{X}_0 - \mathbf{X}_1\|^2 + \|\pi_0 \mathbf{X}_0 + \pi_1 \mathbf{X}_1 - \tilde{\mu}(\mathbf{X}_0, \mathbf{X}_1)\|^2 \right). \tag{23}$$

In turn, the assignment functions are given as

$$\mathcal{A}_0(\mathbf{x}_0)_k := \mathbb{Q} \left( \arg \min_{k'} \|\pi_0 \mathbf{x}_0 + \pi_1 \mathbf{X}_1 - \mu_{k'}\|^2 = k \Big| \mathbf{X}_0 = \mathbf{x}_0 \right) \tag{24}$$

and

$$\mathcal{A}_1(\mathbf{x}_1)_k := \mathbb{Q} \left( \arg \min_{k'} \|\pi_0 \mathbf{X}_0 + \pi_1 \mathbf{x}_1 - \mu_{k'}\|^2 = k \Big| \mathbf{X}_1 = \mathbf{x}_1 \right). \tag{25}$$

Hence, $C(\boldsymbol{\mu}, \mathcal{A}_0, \mathcal{A}_1) = \mathbb{E}_{(\mathbf{X}_0, \mathbf{X}_1) \sim \mathbb{Q}} \left( 2\pi_0 \pi_1 \|\mathbf{X}_0 - \mathbf{X}_1\|^2 + \min_k \|\pi_0 \mathbf{X}_0 + \pi_1 \mathbf{X}_1 - \mu_k\|^2 \right)$.

**(About Remark 3.4)** As discussed in Section 3.1 and Remark 3.4, we additionally show that the optimal perfectly fair assignment function is deterministic when $n_0 = n_1$. For a given $\mathbf{x}_i \in \mathcal{X}_0$, note that $\gamma_{i,j} > 0$ for a unique $j \in [n_1]$, meaning that $\mathbb{Q}$ corresponds to the one-to-one matching map $\mathbf{T}$ in Theorem 3.1 ($\because$ finding $\mathbb{Q}$ becomes equivalent to finding the optimal permutation when $n_0 = n_1$, see Remark 2.4 in Peyré & Cuturi (2020) for the theoretical evidence). Also, note that $\pi_0 = \pi_1 = 1/2$. Then, we have $\mathcal{A}_0^*(\mathbf{x}_i)_k := \mathbb{Q}^* \left( \arg \min_{k'} \|\mathbf{T}^A(\mathbf{x}_i, \mathbf{X}_1) - \mu_{k'}\|^2 = k | \mathbf{X}_0 = \mathbf{x}_i \right) = \mathbb{1}(\arg \min_{k'} \|\frac{\mathbf{x}_i + \mathbf{T}(\mathbf{x}_i)}{2} - \mu_{k'}\|^2 = k)$, which is deterministic.

$\square$

### B.3 PROOF OF THEOREM 4.1

**Theorem 4.1** Minimizing the FCA-C objective $\tilde{C}(\mathbb{Q}, \mathcal{W}, \boldsymbol{\mu})$ with the corresponding assignment function defined in eq. (6) is equivalent to minimizing $C(\boldsymbol{\mu}, \mathcal{A}_0, \mathcal{A}_1)$ subject to $(\mathcal{A}_0, \mathcal{A}_1) \in \mathbf{A}_\epsilon$.

*Proof of Theorem 4.1.* It suffices to show the followings: A and B.

    A. For given $\mathbb{Q}, \mathcal{W}, \boldsymbol{\mu}$, let $\mathcal{A}_0$ and $\mathcal{A}_1$ be the constructed assignment functions, defined in eq. (6). Then, we have $\tilde{C}(\mathbb{Q}, \mathcal{W}, \boldsymbol{\mu}) = C(\boldsymbol{\mu}, \mathcal{A}_0, \mathcal{A}_1)$ and $(\mathcal{A}_0, \mathcal{A}_1) \in \mathbf{A}_\epsilon$.

    B. (i) For given $\boldsymbol{\mu}$ and $(\mathcal{A}_0, \mathcal{A}_1) \in \mathbf{A}_\epsilon$, there exist $\mathbb{Q}$ and $\mathcal{W}$ s.t. $\tilde{C}(\mathbb{Q}, \mathcal{W}, \boldsymbol{\mu}) \leq C(\boldsymbol{\mu}, \mathcal{A}_0, \mathcal{A}_1)$.

• *Proof of A.*

For given $\mathbb{Q}, \mathcal{W}, \boldsymbol{\mu}$, we construct the assignment functions as in eq. (6). In other words, we define

$$
\begin{aligned}
\mathcal{A}_0(\mathbf{x}_0)_k \quad &= \mathbb{P}_1(\{\arg\min_{k'} \|\mathbf{T}^{\mathrm{A}}(\mathbf{x}_0, \mathbf{X}_1) - \mu_{k'}\|^2 = k, (\mathbf{x}_0, \mathbf{X}_1) \in \mathcal{W}^c\}) \\
&\quad + \mathbb{1}\left(\arg\min_{k'} \|\mathbf{x}_0 - \mu_{k'}\|^2 = k\right) \cdot \mathbb{P}_1(\{(\mathbf{x}_0, \mathbf{X}_1) \in \mathcal{W}\}),
\end{aligned}
$$

and the assignment function $\mathcal{A}_1$ is defined similarly. Then, we have $\tilde{C}(\mathbb{Q}, \mathcal{W}, \boldsymbol{\mu}) = C(\boldsymbol{\mu}, \mathcal{A}_0, \mathcal{A}_1)$ by its definition.

Furthermore,

$$
\begin{aligned}
&\sum_{k=1}^K |\mathbb{E}_0 \mathcal{A}_0(\mathbf{X})_k - \mathbb{E}_1 \mathcal{A}_1(\mathbf{X})_k| \\
&= \sum_{k=1}^K \Bigg| \mathbb{E}_0 \mathbb{Q}_{\mathbf{X}_1} \left( \arg\min_{k'} \|\mathbf{T}^{\mathrm{A}}(\mathbf{X}_0, \mathbf{X}_1) - \mu_{k'}\|^2 = k, (\mathbf{X}_0, \mathbf{X}_1) \in \mathcal{W}^c \right) \\
&\qquad - \mathbb{E}_1 \mathbb{Q}_{\mathbf{X}_0} \left( \arg\min_{k'} \|\mathbf{T}^{\mathrm{A}}(\mathbf{X}_0, \mathbf{X}_1) - \mu_{k'}\|^2 = k, (\mathbf{X}_0, \mathbf{X}_1) \in \mathcal{W}^c \right) \\
&\qquad + \mathbb{E}_0 \mathbb{Q}_{\mathbf{X}_1} ((\mathbf{X}_0, \mathbf{X}_1) \in \mathcal{W}) \, \mathbb{1}(\arg\min_{k'} \|\mathbf{X}_0 - \mu_{k'}\|^2 = k) \\
&\qquad - \mathbb{E}_1 \mathbb{Q}_{\mathbf{X}_0} ((\mathbf{X}_0, \mathbf{X}_1) \in \mathcal{W}) \, \mathbb{1}(\arg\min_{k'} \|\mathbf{X}_1 - \mu_{k'}\|^2 = k) \Bigg| \\
&= \sum_{k=1}^K \Bigg| \mathbb{E}_0 \mathbb{Q}_{\mathbf{X}_1} ((\mathbf{X}_0, \mathbf{X}_1) \in \mathcal{W}) \, \mathbb{1}(\arg\min_{k'} \|\mathbf{X}_0 - \mu_{k'}\|^2 = k) \\
&\qquad - \mathbb{E}_1 \mathbb{Q}_{\mathbf{X}_0} ((\mathbf{X}_0, \mathbf{X}_1) \in \mathcal{W}) \, \mathbb{1}(\arg\min_{k'} \|\mathbf{X}_1 - \mu_{k'}\|^2 = k) \Bigg| \\
&= \sum_{k=1}^K \left| \mathbb{E}_{\mathbf{X}_0, \mathbf{X}_1} \mathbb{1}((\mathbf{X}_0, \mathbf{X}_1) \in \mathcal{W}) \cdot \left( \mathbb{1}(\arg\min_{k'} \|\mathbf{X}_0 - \mu_{k'}\|^2 = k) - \mathbb{1}(\arg\min_{k'} \|\mathbf{X}_1 - \mu_{k'}\|^2 = k) \right) \right| \\
&\leq \sum_{k=1}^K \mathbb{E}_{\mathbf{X}_0, \mathbf{X}_1} \left( \left| \mathbb{1}(\arg\min_{k'} \|\mathbf{X}_0 - \mu_{k'}\|^2 = k) - \mathbb{1}(\arg\min_{k'} \|\mathbf{X}_1 - \mu_{k'}\|^2 = k) \right| \big| (\mathbf{X}_0, \mathbf{X}_1) \in \mathcal{W} \right) \\
&\qquad \cdot \mathbb{P}_{\mathbf{X}_0, \mathbf{X}_1} \left| \mathbb{1}((\mathbf{X}_0, \mathbf{X}_1) \in \mathcal{W}) \right| \\
&\leq \epsilon,
\end{aligned}
\tag{26}
$$

which implies $(\mathcal{A}_0, \mathcal{A}_1) \in \mathbf{A}_\epsilon$.

• *Proof of B.*

For each $k \in [K]$, let $\delta_k = \min\{\mathbb{E}_0(\mathcal{A}_0(\mathbf{X}))_k, \mathbb{E}_1(\mathcal{A}_1(\mathbf{X}))_k\}$. We decompose $\mathcal{A}_s(\cdot)_k = \tilde{\mathcal{A}}_s(\cdot)_k + \mathcal{A}_s^c(\cdot)_k$, where $\tilde{\mathcal{A}}_s(\cdot)_k = \delta_k \mathcal{A}_s(\cdot)_k / \mathbb{E}_s(\mathcal{A}_s(\mathbf{X}))_k$. Then, $\mathbb{E}_0 \tilde{\mathcal{A}}_0(\mathbf{X})_k = \mathbb{E}_1 \tilde{\mathcal{A}}_1(\mathbf{X})_k$ for all $k \in [K]$. Define $\tilde{\mathcal{A}}_s(\mathbf{X})_{K+1} := \sum_{k=1}^K \mathcal{A}_s^c(\mathbf{X})_k$. Note that $\mathbb{E}_s \tilde{\mathcal{A}}_s(\mathbf{X})_{K+1} \leq \epsilon$.

Now, for given $\boldsymbol{\mu}$, we define a probability measure on $\mathcal{X}_0 \times \mathcal{X}_1$ by

$$\mathbb{Q}(d\mathbf{x}_0, d\mathbf{x}_1|\boldsymbol{\mu}) = \sum_{k=1}^{K} \frac{\tilde{\mathcal{A}}_0(\mathbf{x}_0)_k \tilde{\mathcal{A}}_1(\mathbf{x}_1)_k}{\delta_k} p_0(\mathbf{x}_0) p_1(\mathbf{x}_1) + \frac{\tilde{\mathcal{A}}_0(\mathbf{x}_0)_{K+1} \tilde{\mathcal{A}}_1(\mathbf{x}_1)_{K+1}}{\delta_{K+1}} p_0(\mathbf{x}_0) p_1(\mathbf{x}_1),$$

where $\delta_{K+1} = 1 - \sum_{k=1}^{K} \delta_k$ and $p_0, p_1$ are the densities of $\mathbb{P}_0, \mathbb{P}_1$, respectively.

Note that $\delta_{K+1} = \mathbb{E}_s \tilde{\mathcal{A}}_s(\mathbf{X})_{K+1}$ and thus $\delta_{K+1} \leq \epsilon$. In addition, for given $(\mathbf{x}_0, \mathbf{x}_1) \in \mathcal{X}_0 \times \mathcal{X}_1$, we define a binary random variable $R(\mathbf{x}_0, \mathbf{x}_1)$ such that

$$\Pr(R(\mathbf{x}_0, \mathbf{x}_1) = 1) = \frac{\frac{\tilde{\mathcal{A}}_0(\mathbf{x}_0)_{K+1} \tilde{\mathcal{A}}_1(\mathbf{x}_1)_{K+1}}{\delta_{K+1}} p_0(\mathbf{x}_0) p_1(\mathbf{x}_1)}{\mathbb{Q}(d\mathbf{x}_0, d\mathbf{x}_1|\boldsymbol{\mu})}.$$

For given $\mathbf{x} \in \mathcal{X}_s$, let $\mu(\mathbf{x}) = \mu_{k^*}$, where $k^* = \operatorname{argmin}_k \|\mathbf{x} - \mu_k\|^2$. For given $(\mathbf{x}_0, \mathbf{x}_1) \in \mathcal{X}_0 \times \mathcal{X}_1$, let $\tilde{\mu}(\mathbf{x}_0, \mathbf{x}_1) = \mu_{k'}$, where $k' = \arg\min_k \|\pi_0 \mathbf{x}_0 + \pi_1 \mathbf{x}_1 - \mu_k\|^2$. Then, it holds that $\tilde{C}(\mathbb{Q}, R, \mu) \leq C(\boldsymbol{\mu}, \mathcal{A}_0, \mathcal{A}_1)$, where

$$\tilde{C}(\mathbb{Q}, R, \mu) := \mathbb{E}_{\mathbb{Q},R}\left[\left\{2\pi_0\pi_1 \|\mathbf{X}_0 - \mathbf{X}_1\|^2 + \|\pi_0 \mathbf{X}_0 + \pi_1 \mathbf{X}_1 - \tilde{\mu}(\mathbf{X}_0, \mathbf{X}_1)\|^2\right\}(1 - R(\mathbf{X}_0, \mathbf{X}_1))\right]$$

$$+ \mathbb{E}_{\mathbb{Q},R}\left[\left\{\pi_0 \|\mathbf{X}_0 - \mu(\mathbf{X}_0)\|^2 + \pi_1 \|\mathbf{X}_1 - \mu(\mathbf{X}_1)\|^2\right\} R(\mathbf{X}_0, \mathbf{X}_1)\right].$$

The final mission is to find $\mathcal{W} \subset \mathcal{X}_0 \times \mathcal{X}_1$ such that $R(\mathbf{x}_0, \mathbf{x}_1) = \mathbb{1}((\mathbf{x}_0, \mathbf{x}_1) \in \mathcal{W})$. For this, define $\eta(\mathbf{x}_0, \mathbf{x}_1) = 2\pi_0\pi_1 \|\mathbf{x}_0 - \mathbf{x}_1\|^2 + \min_k \|\mathbf{T}^A(\mathbf{x}_0, \mathbf{x}_1) - \mu_k\|^2$. Let $\eta_\epsilon$ be the $\epsilon$th upper quantile of $\eta(\mathbf{X}_0, \mathbf{X}_1)$ and let

$$\tilde{\mathcal{W}} = \{(\mathbf{x}_0, \mathbf{x}_1) : \eta(\mathbf{x}_0, \mathbf{x}_1) > \eta_\epsilon\}. \tag{27}$$

Then, we can find $\mathcal{W} \supset \tilde{\mathcal{W}}$ such that $\tilde{C}(\mathbb{Q}, \mathcal{W}, \mu) \leq \tilde{C}(\mathbb{Q}, R, \mu)$ and $\mathbb{Q}(\mathcal{W}) = \epsilon$ (see Remark B.1 below), which completes the proof.

$\square$

**Remark B.1** (The case when the densities do not exist). *When the distribution of $\eta(\mathbf{X}_0, \mathbf{X}_1)$ is strictly increasing, we have $\mathbb{Q}(\mathcal{W}) = \epsilon$. If not, we can find $\mathcal{W}$ such that $\mathcal{W} \supset \{(\mathbf{x}_0, \mathbf{x}_1) : \eta(\mathbf{x}_0, \mathbf{x}_1) \geq \epsilon\}$ with $\mathbb{Q}(\mathcal{W}) = \epsilon$ when $\mathbb{Q}$ has its density.*

*When $\mathbb{Q}$ is discrete, the situation is tricky. When $n_0 = n_1$, the measure $\mathbb{Q}$ minimizing $\tilde{C}(\mathbb{Q}, \mathcal{W}, \mu)$ has masses $1/n_0$ on $n_0$ many pairs of $(\mathbf{x}_0, \mathbf{x}_1)$ among $\mathcal{X}_0 \times \mathcal{X}_1$. In this case, whenever $n_0\epsilon$ is an integer, we can find $\mathcal{W}$ such that $\mathbb{Q}(\mathcal{W}) = \epsilon$.*

*Otherwise, we could consider a random assignment. Let $F_\eta$ be the distribution of $\eta(\mathbf{X}_0, \mathbf{X}_1)$. Suppose that $F_\eta$ has a jump at $\eta_\epsilon$. In that case, $\mathbb{Q}(\mathcal{W}) < \epsilon$. Let $(\mathbf{x}_0^*, \mathbf{x}_1^*)$ be the element such that $\eta(\mathbf{x}_0^*, \mathbf{x}_1^*) = \eta_\epsilon$. Then, we can let $R(\mathbf{x}_0^*, \mathbf{x}_1^*) = 1$ with probability $(\epsilon - \mathbb{Q}(\mathcal{W}))/\mathbb{Q}(\{\mathbf{x}_0^*, \mathbf{x}_1^*\})$, $R(\mathbf{x}_0, \mathbf{x}_1) = 1$ for $(\mathbf{x}_0, \mathbf{x}_1) \in \mathcal{W}$ and $R(\mathbf{x}_0, \mathbf{x}_1) = 0$ for $(\mathbf{x}_0, \mathbf{x}_1) \in (\mathcal{W} \cup \{\mathbf{x}_0^*, \mathbf{x}_1^*\})^c$. The current FCA-C algorithm can be modified easily for this random assignment.*

## B.4 PROOF OF COROLLARY 4.2

**Corollary 4.2** The assignment functions in $\mathbf{A}_\epsilon$ satisfies $|\text{Balance} - 1| \leq C\epsilon$, where $C = \max_{s,k} \frac{\pi_s}{\mathbb{E}(\mathcal{A}_S(\mathbf{X})_k)}$.

*Proof of Corollary 4.2.* By the fact that for all $k \in [K]$, $\mathbb{E}(\mathcal{A}_S(\mathbf{X})_k) = \pi_0 \mathbb{E}_0(\mathcal{A}_0(\mathbf{X})_k) + \pi_1 \mathbb{E}_1(\mathcal{A}_1(\mathbf{X})_k)$, we have

$$
\left| \frac{\mathbb{E}_0(\mathcal{A}_0(\mathbf{X})_k)}{\mathbb{E}(\mathcal{A}_S(\mathbf{X})_k)} - \frac{\mathbb{E}_1(\mathcal{A}_1(\mathbf{X})_k)}{\mathbb{E}(\mathcal{A}_S(\mathbf{X})_k)} \right| = \left| \frac{\mathbb{E}_s(\mathcal{A}_s(\mathbf{X})_k)}{\mathbb{E}(\mathcal{A}_S(\mathbf{X})_k)} - \frac{1}{\pi_{s'}} \left( 1 - \pi_s \frac{\mathbb{E}_s(\mathcal{A}_s(\mathbf{X})_k)}{\mathbb{E}(\mathcal{A}_S(\mathbf{X})_k)} \right) \right|
$$
$$
= \frac{1}{\pi_{s'}} \left| \frac{\mathbb{E}_s(\mathcal{A}_s(\mathbf{X})_k)}{\mathbb{E}(\mathcal{A}_S(\mathbf{X})_k)} - 1 \right|, \forall s \in \{0,1\}. \tag{28}
$$

Combining (28) and the definition $\mathbf{A}_\epsilon = \{(\mathcal{A}_0, \mathcal{A}_1) : \sum_{k=1}^{K} |\mathbb{E}_0(\mathcal{A}_0(\mathbf{X})_k) - \mathbb{E}_1(\mathcal{A}_1(\mathbf{X})_k)| \leq \epsilon\}$, we have

$$
\left| \frac{\mathbb{E}_s(\mathcal{A}_s(\mathbf{X})_k)}{\mathbb{E}(\mathcal{A}_S(\mathbf{X})_k)} - 1 \right| = \frac{\pi_{s'}}{\mathbb{E}(\mathcal{A}_S(\mathbf{X})_k)} |\mathbb{E}_0(\mathcal{A}_0(\mathbf{X})_k) - \mathbb{E}_1(\mathcal{A}_1(\mathbf{X})_k)| \leq \frac{\pi_{s'}}{\mathbb{E}(\mathcal{A}_S(\mathbf{X})_k)} \epsilon, \forall s \in \{0,1\}. \tag{29}
$$

Since this bound holds for any $s \in \{0,1\}$ and $k \in [K]$, we conclude that $|\text{Balance} - 1| = 1 - \text{Balance} = 1 - \min_{k \in [K]} \min_{s \in \{0,1\}} \frac{\mathbb{E}_s(\mathcal{A}_s(\mathbf{X})_k)}{\mathbb{E}(\mathcal{A}_S(\mathbf{X})_k)} \leq C\epsilon$, where $C = \max_{s,k} \frac{\pi_s}{\mathbb{E}(\mathcal{A}_S(\mathbf{X})_k)}$. $\qquad \square$

## B.5 APPROXIMATION GUARANTEE OF FCA-C

We establish the approximation guarantee of our proposed algorithm: **The cost of our fair clustering solution is at most $\tau + 2$ times of the cost of the global optimal fair clustering solution** (where $\tau$ is the approximation error of the algorithm used for $K$-means clustering). In other words, FCA-C has an approximation error of $\tau + 2$ for any fairness level.

Recall the $K$-means clustering cost $C(\mu, \mathcal{A}_0, \mathcal{A}_1) := \mathbb{E} \sum_{k=1}^{K} \mathcal{A}_S(\mathbf{X})_k \|\mathbf{X} - \mu_k\|^2$, where $\mu = \{\mu_k\}_{k=1}^{K}$. Note that this is equivalently written as $\sum_{i=1}^{n} \sum_{k=1}^{K} \mathcal{A}_{s_i}(\mathbf{x}_i)_k \|\mathbf{x}_i - \mu_k\|^2$. For a given $\epsilon$, let $\tilde{\mu}, \tilde{\mathcal{A}}_0, \tilde{\mathcal{A}}_1$ be the optimal solution of $\min_{\mu, \mathcal{A}_0, \mathcal{A}_1} C(\mu, \mathcal{A}_0, \mathcal{A}_1)$ subject to $(\mathcal{A}_0, \mathcal{A}_1) \in \mathbf{A}_\epsilon$, i.e., optimal fair clustering solution for fairness level $\epsilon$. Let $C^* = \min_{\mu, \mathcal{A}_0, \mathcal{A}_1} C(\mu, \mathcal{A}_0, \mathcal{A}_1)$ be the global optimal clustering cost without fairness constraint.

Then, we have the following results: Theorem B.2 and Proposition B.3.

**Theorem B.2.** *Given a $\tau$-approximation algorithm for (standard) $K$-means clustering, let $\mu^\diamond, \mathcal{A}_0^\diamond, \mathcal{A}_1^\diamond$ be the (approximated) solution of $\min_{\mu, \mathcal{A}_0, \mathcal{A}_1} C(\mu, \mathcal{A}_0, \mathcal{A}_1)$, found by the $\tau$-approximation algorithm. For fixed $\mu^\diamond$, we have that $C(\mu^\diamond, \mathcal{A}_0', \mathcal{A}_1') \leq (\tau + 2)C^*$, where $\mathcal{A}_0', \mathcal{A}_1'$ are the assignment functions corresponding to $\mathbb{Q}', \mathcal{W}' = \arg\min_{\mathbb{Q}, \mathcal{W}} \tilde{C}(\mathbb{Q}, \mathcal{W}, \mu^\diamond)$, i.e., the solution of FCA-C under fixed $\mu^\diamond$.*

*Proof of Theorem B.2.* It suffices to show that there exist $\mathcal{A}_0^+$ and $\mathcal{A}_1^+$ such that (i) $(\mathcal{A}_0^+, \mathcal{A}_1^+) \in \mathbf{A}_\epsilon$, (ii) $C(\mu^\diamond, \mathcal{A}_0^+, \mathcal{A}_1^+) \leq (\tau + 2)C^*$, and (iii) $C(\mu^\diamond, \mathcal{A}_0', \mathcal{A}_1') \leq C(\mu^\diamond, \mathcal{A}_0^+, \mathcal{A}_1^+)$.

Recall that $\tilde{\mu} = \{\tilde{\mu}_k\}_{k=1}^{K}$ and $\mu^\diamond = \{\mu_k^\diamond\}_{k=1}^{K}$. For all $k \in [K]$, we define the set of nearest neighbors of $\mu_k^\diamond$ as $N(\mu_k^\diamond) := \{\tilde{\mu}_k \in \tilde{\mu} : \arg\min_{\mu_{k'}^\diamond \in \mu^\diamond} \|\tilde{\mu}_k - \mu_{k'}^\diamond\|^2 = \mu_k^\diamond\}$. First, for $\mathbf{x} \in \mathcal{X}$, we define $\mathcal{A}_s^+(\mathbf{x})_k := \sum_{k':\tilde{\mu}_{k'} \in N(\mu_k^\diamond)} \tilde{\mathcal{A}}_s(\mathbf{x})_{k'}$.

By the definitions of $\tilde{\mathcal{A}}_0$ and $\tilde{\mathcal{A}}_1$, we have $\sum_{k=1}^{K} \left| \frac{1}{n_0} \sum_{i=1}^{n_0} \tilde{\mathcal{A}}_0(\mathbf{x}_i^{(0)})_k - \frac{1}{n_1} \sum_{j=1}^{n_1} \tilde{\mathcal{A}}_1(\mathbf{x}_j^{(1)})_k \right| \leq \epsilon$. Then,

$$\sum_{k=1}^{K} \left| \frac{1}{n_0} \sum_{i=1}^{n_0} \mathcal{A}_0^+(\mathbf{x}_i^{(0)})_k - \frac{1}{n_1} \sum_{j=1}^{n_1} \mathcal{A}_1^+(\mathbf{x}_j^{(1)})_k \right|$$

$$= \sum_{k=1}^{K} \left| \sum_{k':\tilde{\mu}_{k'} \in N(\mu_k^\diamond)} \left( \frac{1}{n_0} \sum_{i=1}^{n_0} \tilde{\mathcal{A}}_0(\mathbf{x}_i^{(0)})_{k'} - \frac{1}{n_1} \sum_{j=1}^{n_1} \tilde{\mathcal{A}}_1(\mathbf{x}_j^{(1)})_{k'} \right) \right|$$

$$\leq \sum_{k=1}^{K} \sum_{k':\tilde{\mu}_{k'} \in N(\mu_k^\diamond)} \left| \frac{1}{n_0} \sum_{i=1}^{n_0} \tilde{\mathcal{A}}_0(\mathbf{x}_i^{(0)})_{k'} - \frac{1}{n_1} \sum_{j=1}^{n_1} \tilde{\mathcal{A}}_1(\mathbf{x}_j^{(1)})_{k'} \right|$$

$$= \sum_{k'=1}^{K} \left| \frac{1}{n_0} \sum_{i=1}^{n_0} \tilde{\mathcal{A}}_0(\mathbf{x}_i^{(0)})_{k'} - \frac{1}{n_1} \sum_{j=1}^{n_1} \tilde{\mathcal{A}}_1(\mathbf{x}_j^{(1)})_{k'} \right| \leq \epsilon,$$

where the last equality holds because $\cup_{k=1}^{K} \cup_{k':\tilde{\mu}_{k'} \in N(\mu_k^\diamond)} \{k'\} = \{k'\}_{k'=1}^{K}$. Hence, (i) is proved.

Second, for a given $\mathbf{x} \in \mathcal{X}$, let $\mu_k^\diamond(\mathbf{x}) := \arg\min_{\mu_k^\diamond \in \mu^\diamond} \|\mathbf{x} - \mu_k^\diamond\|^2$ be the cluster center that $\mathbf{x}$ is assigned to, found by $\tau$ approximation algorithm for $K$-means clustering. Note that $\mathcal{A}_0^\diamond$ and $\mathcal{A}_1^\diamond$ are deterministic, hence we can represent $\mu_k^\diamond$ as above. Then, we have that

$$C(\mu^\diamond, \mathcal{A}_0^+, \mathcal{A}_1^+) = \sum_{k=1}^{K} \mathcal{A}_s^+(\mathbf{x})_k \|\mathbf{x} - \mu_k^\diamond\|^2 = \sum_{k=1}^{K} \sum_{k':\tilde{\mu}_{k'} \in N(\mu_k^\diamond)} \tilde{\mathcal{A}}_s(\mathbf{x})_{k'} \|\mathbf{x} - \mu_k^\diamond\|^2$$

$$\leq \sum_{k=1}^{K} \sum_{k':\tilde{\mu}_{k'} \in N(\mu_k^\diamond)} \tilde{\mathcal{A}}_s(\mathbf{x})_{k'} \left( \|\mathbf{x} - \tilde{\mu}_{k'}\|^2 + \|\tilde{\mu}_{k'} - \mu_k^\diamond\|^2 \right)$$

$$\leq \sum_{k=1}^{K} \sum_{k':\tilde{\mu}_{k'} \in N(\mu_k^\diamond)} \tilde{\mathcal{A}}_s(\mathbf{x})_{k'} \left( \|\mathbf{x} - \tilde{\mu}_{k'}\|^2 + \|\tilde{\mu}_{k'} - \mu_k^\diamond(\mathbf{x})\|^2 \right)$$

$$\leq \sum_{k=1}^{K} \sum_{k':\tilde{\mu}_{k'} \in N(\mu_k^{\diamond})} \tilde{\mathcal{A}}_s(\mathbf{x})_{k'} \left( 2\|\mathbf{x} - \tilde{\mu}_{k'}\|^2 + \|\mathbf{x} - \mu_k^{\diamond}(\mathbf{x})\|^2 \right)$$

$$= 2 \sum_{k'=1}^{K} \tilde{\mathcal{A}}_s(\mathbf{x})_{k'} \|\mathbf{x} - \tilde{\mu}_{k'}\|^2 + \sum_{k=1}^{K} \sum_{k':\tilde{\mu}_{k'} \in N(\mu_k^{\diamond})} \tilde{\mathcal{A}}_s(\mathbf{x})_{k'} \|\mathbf{x} - \mu_k^{\diamond}(\mathbf{x})\|^2$$

$$= 2C(\tilde{\mu}, \tilde{\mathcal{A}}_0, \tilde{\mathcal{A}}_1) + \|\mathbf{x} - \mu_k^{\diamond}(\mathbf{x})\|^2 = 2C(\tilde{\mu}, \tilde{\mathcal{A}}_0, \tilde{\mathcal{A}}_1) + \min_{\mu_k^{\diamond} \in \mu^{\diamond}} \|\mathbf{x} - \mu_k^{\diamond}\|^2$$

$$\leq 2C(\tilde{\mu}, \tilde{\mathcal{A}}_0, \tilde{\mathcal{A}}_1) + \tau C^* \leq 2C(\tilde{\mu}, \tilde{\mathcal{A}}_0, \tilde{\mathcal{A}}_1) + \tau C(\tilde{\mu}, \tilde{\mathcal{A}}_0, \tilde{\mathcal{A}}_1) = (\tau + 2)C(\tilde{\mu}, \tilde{\mathcal{A}}_0, \tilde{\mathcal{A}}_1).$$

Hence, (ii) is proved.

Finally, it is clear that $C(\mu^{\diamond}, \mathcal{A}_0', \mathcal{A}_1') \leq C(\mu^{\diamond}, \mathcal{A}_0^+, \mathcal{A}_1^+)$, since $\mathcal{A}_0'$ and $\mathcal{A}_1'$ are the minimizers of of $C(\mu, \mathcal{A}_0, \mathcal{A}_1)$ subject to $(\mathcal{A}_0, \mathcal{A}_1) \in \mathbf{A}_\epsilon$ given $\mu = \mu^{\diamond}$ (by Theorem 4.1). Hence, (iii) is proved. $\square$

**Proposition B.3.** *For given $\mathcal{A}_0', \mathcal{A}_1'$, there exists $\mu'$ such that $C(\mu', \mathcal{A}_0', \mathcal{A}_1') \leq C(\mu^{\diamond}, \mathcal{A}_0', \mathcal{A}_1')$.*

*Proof of Proposition B.3.* Letting $\mu_k' := \sum_{i=1}^{n} \mathcal{A}_s(\mathbf{x}_i)_k \mathbf{x}_i / \sum_{i=1}^{n} \mathcal{A}_s(\mathbf{x}_i)_k$, which solves $\min_\mu C(\mu, \mathcal{A}_0', \mathcal{A}_1')$, concludes the proof. $\square$

**Remark B.4** (Implication of Theorem B.2 and Proposition B.3)**.** *Theorem B.2 shows that finding fair clustering (assignments) using FCA-C with fixed cluster centers (found by an approximation algorithm for $K$-means clustering) has a guaranteed approximation error. In other words, for the fair assignment functions $\mathcal{A}_0'$ and $\mathcal{A}_1'$ obtained by FCA-C, it is guaranteed that cost of fair clustering obtained by FCA-C (for fixed cluster centers) $\leq (\tau + 2)\times$ cost of the optimal fair clustering.*

*Proposition B.3 subsequently presents a straightforward result that updating cluster centers also guarantees at least the same approximation error. In other words, once the cluster centers are updated by minimizing the clustering cost given the assignment functions, it follows that: cost of fair clustering obtained by FCA-C (after one cluster center update) $\leq$ cost of fair clustering obtained by FCA-C (for fixed cluster centers). This implies that the fair clustering obtained by FCA-C with updated cluster centers also has the $(\tau + 2)$-approximation guarantee.*

## C EXPERIMENTS

### C.1 DATASETS

ADULT dataset (Becker & Kohavi, 1996) can be downloaded from `https://archive.is.uci/ml/datasets/adult`. We use 5 continuous variables (`age`, `fnlwgt`, `education`, `capital-gain`, `capital-loss`, `hours-per-week`). The total sample size is 32,561. The sample size for $S = 0$ and $S = 1$ (resp. female and male) are 10,771 and 21,790, respectively.

BANK dataset (Moro et al., 2012) can be downloaded from `https://archive.ics.uci.edu/ml/datasets/Bank+Marketing`. We use 6 continuous variables (`age`, `duration`, `euribor3m`, `nr.employed`, `cons.price.idx`, `campaign`). The total sample size is 41,108. The sample size for $S = 0$ and $S = 1$ (resp. not married and married) are 16,180 and 24,928, respectively.

CENSUS dataset (Meek et al.) can be downloaded from `https://archive.ics.uci.edu/ml/datasets/US+Census+Data+(1990)`. We use 66 continuous variables (`dAncstry1`, `dAncstry2`, `iAvail`, `iCitizen`, `iClass`, `dDepart`, `iDisabl1`, `iDisabl2`, `iEnglish`, `iFeb55`, `iFertil`, `dHispanic`, `dHour89`, `dHours`, `iImmigr`, `dIncome1`, `dIncome2`, `dIncome3`, `dIncome4`, `dIncome5`, `dIncome6`, `dIncome7`, `dIncome8`, `dIndustry`, `iKorean`, `iLang1`, `iLooking`, `iMarital`, `iMay75880`, `iMeans`, `iMilitary`, `iMobility`, `iMobillim`, `dOccup`, `iOthrserv`, `iPerscare`, `dPOB`, `dPoverty`, `dPwgt1`, `iRagechld`, `dRearning`, `iRelat1`, `iRelat2`, `iRemplpar`, `iRiders`, `iRlabor`, `iRownchld`, `dRpincome`, `iRPOB`, `iRrelchld`, `iRspouse`, `iRvetserv`, `iSchool`, `iSept80`, `iSubfam1`, `iSubfam2`, `iTmpabsnt`, `dTravtime`, `iVietnam`, `dWeek89`, `iWork89`, `iWorklwk`, `iWWII`, `iYearsch`, `iYearwrk`, `dYrsserv`). We subsample 20,000-many data, as done in Chierichetti et al. (2017); Bera et al. (2019); Esmaeili et al. (2021). The sample size for $S = 0$ and $S = 1$ (resp. not married and married) are 9,844 and 10,156 respectively.

For each dataset, we use only the numerical variables as introduced above, consistent with previous studies, e.g., Bera et al. (2019); Ziko et al. (2021), to name a few. The variables of all datasets are scaled with zero mean and unit variance.

### C.2 IMPLEMENTATION DETAILS

#### C.2.1 PROPOSED ALGORITHMS

**FCA**  We use the `POT` library (Flamary et al., 2021) to find the optimal joint distribution $\mathbb{Q}$ (i.e., $\Gamma$). For updating cluster centers, we adopt the $K$-means++ algorithm (Arthur & Vassilvitskii, 2007) from the implementation of `scikit-learn` package Pedregosa et al. (2011). The iterative process of updating cluster centers and the joint distribution is run for 100 iterations, with the result where `Cost` is minimized being selected.

**FCA-C**  To control `Balance`, we vary $\epsilon$, which is the size of $\mathcal{W}$. The value of $\epsilon$ is adjusted in increments of 0.05, ranging from 0.1 to 0.9. We also use a similar partitioning technique in Section 4.3 for FCA-C with $m = 2048$. For stability, at the first iteration step, we find $\mathcal{W}$ based on fixed cluster centers $\boldsymbol{\mu}$, initialized using the $K$-means++ algorithm.

The source code for our proposed algorithms will be made publicly available after acceptance.

#### C.2.2 BASELINE ALGORITHMS

**Scalable Fair Clustering (SFC) (Backurs et al., 2019)**  We directly use the official source code of SFC, available on the authors' GitHub[2]. SFC provides a fast and scalable algorithm for fairlet decomposition, which builds the fairlets in nearly linear time. The given data are first embedded to a tree structure (hierarchically well-separated tree; HST) using probabilistic metric embedding, where we seek for optimal edges (as well as their nodes) to be activated that satisfy `Balance` $\approx 1$. Then, the linked nodes are then aggregated as a fairlet. This process can build fairlets in nearly linear time while minimizing the cost of building them. After building these fairlets using SFC, we apply

---

[2]`https://github.com/talwagner/fair_clustering`

the standard $K$-means algorithm on the fairlet space (i.e., the set of representatives of the obtained fairlets).

**Fair Clustering under a Bounded Cost (FCBC) (Esmaeili et al., 2021)**  We use the official source code of FCBC, available on the authors' GitHub[3]. FCBC maximizes `Balance` under a cost constraint, where the cost constraint is defined by the Price of Fairness (PoF) - 'cost of fair clustering (the solution) / cost of standard clustering'. However, since the authors mentioned the constrained optimization problem is NP-hard, they reduced the problem to a post-processing approach (i.e., fairly assigning data under the cost constraint, with pre-specified centers opened by a standard clustering algorithm). We set the value of PoF to 1.2 to achieve `Balance` $\approx 1$.

**Fair Round-Robin Algorithm for Clustering (FRAC) (Gupta et al., 2023)**  We directly follow the official source code of FRAC, available on the authors' GitHub[4]. FRAC provides an in-loop post-processing approach to fairly assign data from each protected group to given cluster centers found by a standard clustering algorithm. In other words, the fair assignment problem is solved at each iteration of standard clustering algorithm.

**Variational Fair Clustering (VFC) (Ziko et al., 2021)**  We use the official source code of VFC, available on the authors' GitHub[5]. The overall objective of VFC is $\mathbb{E} \min_k \|\mathbf{X} - \mu_k\|^2 + \lambda \cdot \sum_{k=1}^{K} \mathrm{KL} \left( [\pi_0, \pi_1] \| \left[ \frac{\pi_0 \mathbb{E}_0 \mathcal{A}_0(\mathbf{X})_k}{\mathbb{E} \mathcal{A}_S(\mathbf{X})_k}, \frac{\pi_1 \mathbb{E}_1 \mathcal{A}_1(\mathbf{X})_k}{\mathbb{E} \mathcal{A}_S(\mathbf{X})_k} \right] \right)$, where $\lambda$ is a hyper-parameter to control `Balance`. A higher $\lambda$ results in higher `Balance`, with $\mathrm{KL} = 0 \iff$ `Balance` $= 1$. We run the code with multiple trials, by varying the values of $\lambda$. For Table 2, we select the best $\lambda$ that achieves the highest `Balance` and report the corresponding performance for the chosen hyper-parameter, as the authors have done. For Figure 3, we present all the results obtained using the various values of $\lambda$. Table 4 below provides the values of VFC's hyper-parameters used in our experiments.

Table 4: The hyper-parameters used in VFC for searching a clustering with maximum achievable `Balance` for each dataset. The **bold** faces are the recommended ones by the authors. The underlined values are the ones we use for maximum achievable `Balance`.

| Dataset | $L_2$ normalization | Hyper-parameters |
|---------|---------------------|------------------|
| ADULT | O | $\{5000, 7000, \mathbf{9000}, 10000, 11000, 12000, 13000, 13600, \underline{14200}\}$ |
|  | X | $\{5000, 7000, 9000, 10000, 11000, 12000, 13000, 15000, 20000, 22000, \underline{23000}\}$ |
| BANK | O | $\{5000, \mathbf{6000}, 7000, 9000, 10000, 11000, 12000, \underline{12300}\}$ |
|  | X | $\{10000, 12000, 13000, 15000, 17000, 19000, 25200, \underline{26000}\}$ |
| CENSUS | O | $\{100, 200, 500, 700, 1000, 1500, \underline{2000}\}$ |
|  | X | *Failed* |

Note that VFC's superior performance over other two well-known FC algorithms from Bera et al. (2019); Kleindessner et al. (2019) has already been shown in Ziko et al. (2021), which is why we omit these two methods as baselines in our experiments.

---

[3] https://github.com/Seyed2357/Fair-Clustering-Under-Bounded-Cost
[4] https://github.com/shivi98g/Fair-k-means-Clustering-via-Algorithmic-Fairness
[5] https://github.com/imtiazziko/Variational-Fair-Clustering

## C.3 Omitted experimental results

### C.3.1 Performance comparison results - Control of Balance (Section 5.2)

We here provide the trade-off for various fairness levels, when the data are not $L_2$-normalized. Similar to Figure 3, FCA and VFC show similar trade-offs, while VFC cannot achieve certain high values of Balance.

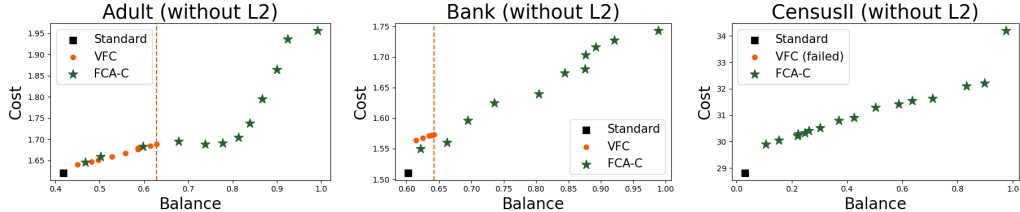

Figure 4: Balance vs. Cost trade-offs for (left) ADULT, (center) BANK, and (right) CENSUS datasets. Black squares (■) are from the standard clustering, orange circles (●) are from VFC, green stars (⋆) are from FCA-C, and orange dashed lines (- -) are the maximums of Balance that VFC can achieve. The data are not $L_2$-normalized.

In specific, on CENSUS dataset without $L_2$ normalization, VFC fails, i.e., it does not achieve higher Balance than the standard clustering for any $\lambda$. We find that the failure of VFC on CENSUS dataset (without $L_2$ normalization) is due to explosion of an exponential term calculated in its algorithm. There exists an exponential term with respect to the clustering cost in the calculation of the optimal assignment vector in the VFC algorithm, and thus when the clustering cost becomes too large, VFC fails due to an overflow. Note that the input dimension of CENSUS dataset is 66, while those of ADULT and BANK are 5 and 6, respectively. While the clustering cost with $L_2$ normalization is bounded regardless of the dimension, the clustering cost without $L_2$ normalization is proportional to the input dimension. This is why VFC fails only for CENSUS dataset without $L_2$ normalization. In contrast, as FCA does not fail at all regardless of the $L_2$ normalization because there is no exponential term in the algorithm, meaning that it is numerically more stable than VFC.

### C.3.2 APPLICABILITY TO VISUAL CLUSTERING (SECTION 5.3)

**Settings: datasets, baselines, and measures**  REVERSE MNIST is a mixture of two image digit datasets: the original MNIST and a color-reversed version (where black and white are swapped). OFFICE-31 is a mixture of two datasets from two domains (amazon and webcam) with 31 classes. Both datasets are used in state-of-the-art visual FC methods (Li et al., 2020; Zeng et al., 2023).

For the baseline, we consider a state-of-the-art FC method in vision domain called FCMI from Zeng et al. (2023). FCMI learns a fair autoencoder with two additional loss terms: (i) clustering loss on the latent space and (ii) mutual information between latent vector and color. While FCMI is an end-to-end method that learns the fair latent vector and perform clustering on the fair latent space simultaneously, FCA is applied to a pre-trained latent space obtained by learning an autoencoder with the reconstruction loss only. We also report the performances of DFC (Li et al., 2020) which was the main baseline in the FCMI paper, along with SFC (Backurs et al., 2019) and VFC (Ziko et al., 2021), even though these two methods are not specifically designed for the vision domain.

The clustering performance for the two image datasets is evaluated by two classification measures ACC (accuracy calculated based on assigned cluster indices and ground-truth labels) and NMI (Normalized Mutual Information between ground-truth label distribution and assigned cluster distribution), which are consistently used in Li et al. (2020); Zeng et al. (2023), as datasets involve ground-truth labels (e.g., $0, 1, \ldots, 9$ for REVERSE MNIST and the 31 classes for OFFICE-31). The fairness performance is evaluated by Balance.

**Results**  Table 3 in the main body shows that FCA performs similar to FCMI, which is the state-of-the-art, while outperforming the other baselines with large margins in terms of Balance. Note that SFC, VFC, and FCA are two-step methods, i.e., find fair clustering on the pre-trained (fair-unaware) latent space, and FCA is the best among those. Furthermore, on the other hand, DFC and FCMI are end-to-end methods so it is surprising that FCA outperforms DFC and performs similarly to FCMI.

**Further comparison between the fairlet-based method and FCA in visual clustering**  We compare the fairlet-based method and FCA using OFFICE-31 dataset, considering (i) not only the overall clustering utility, (ii) but also the similarity of matched features. For a clear comparison, we sample a balanced subset (w.r.t. the label and sensitive attribute) from the original (imbalanced) dataset. Specifically, we ensure that the number of samples with the same label is equal across the two protected groups (i.e., the two domains). As a result, we have 795 images from both the amazon and webcam domains. We then find fairlets or apply FCA on this balanced dataset. Note that finding fairlets when $n_0 = n_1$ is equivalent to finding the optimal transport map (Peyré & Cuturi, 2020).

Table 5 below presents the comparison results using various measures, including performance measures with respect to matching (the matching cost and how much images with the same label are matched) and clustering (Cost, ACC, and NMI). While the fairlets tend to match more similar features (i.e., the matching cost is lower) as expected by the definition of fairlets, FCA exhibits a better ability to collect images with same labels into clusters (i.e., lower Cost, higher ACC, and higher NMI). Moreover, in visual clustering, matching similar features in the pre-trained latent space does not always guarantee that images with the same label are matched (FCA achieves a higher proportion of matchings where images share the same label compared to fairlets). These results suggest that while fairlets provide optimal matchings in terms of feature similarity, however, they may be suboptimal in terms of label similarity and overall clustering utility.

Table 5: Comparison of the fairlet-based method and FCA. 'Matching cost' is defined by the average distance between two matched features. 'Matching = Label' is defined by the average ratio of images with the same label being matched. **Bold**-faced values indicate the best performance.

| Matching method | Matching performance | | Clustering performance | | |
| --- | --- | --- | --- | --- | --- |
| | Matching cost ($\downarrow$) | Matching = Label ($\uparrow$) | Cost ($\downarrow$) | ACC ($\uparrow$) | NMI ($\uparrow$) |
| Fairlet-based | **0.211** | 0.595 | 0.278 | 65.8 | 71.0 |
| FCA ✓ | 0.241 | **0.631** | **0.269** | **69.3** | **72.2** |

### C.3.3   ABLATION STUDY: SELECTION OF THE PARTITION SIZE $m$ (SECTION 5.4)

Figure 5 indicates that using a partition size of around 1000 yields reasonable results. Specifically, using $m$ values greater than 1000 shows similar performance compared to those obtained with $m = 1024$.

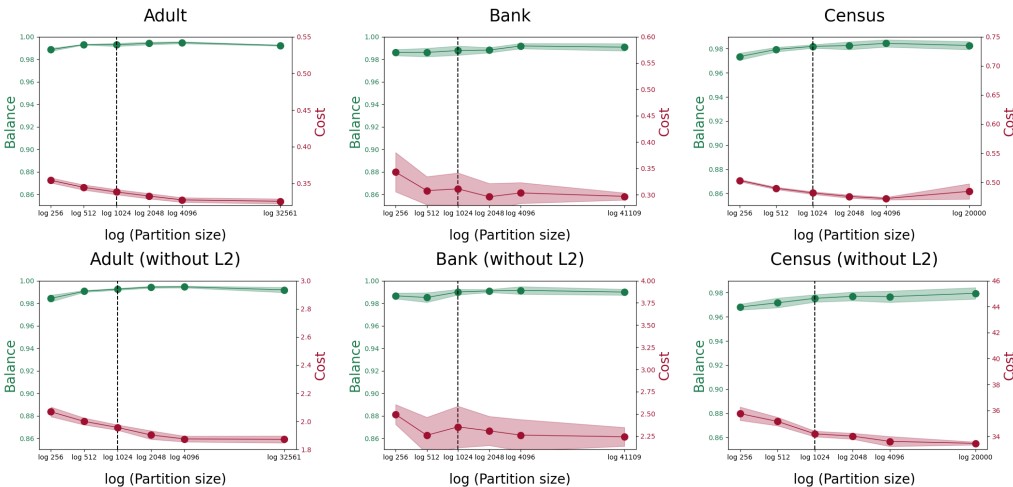

Figure 5: Variations of `Cost` and `Balance` with respect to the partition size. (Left, Center, Right) = (ADULT, BANK, CENSUS). (Top, Bottom) = (with $L_2$ normalization, without $L_2$ normalization).

We further provide the elapsed computation time for various partition sizes, up to using the full dataset. Using $m = 1024$ as the baseline, we calculate the relative computation time (%) for other partition sizes. The comparison, presented in Table 6 below, shows that using $m = 1024$ leads to a significant reduction in computation time.

Table 6: Comparison of computation time with different partition sizes up to using the full dataset. For each partition size and dataset, we provide the averaged relative elapsed time per a single iteration, when compared to computation time spent for $m = 1024$.

| (Relative) elapsed time per iteration | Partition size $m$ | | | | | |
|---|---|---|---|---|---|---|
| | 256 | 512 | 1024 ✓ | 2048 | 4096 | Full |
| ADULT ($n = 32561, d = 5$) | 17% | 58% | 100% | 140% | 344% | 3,184% |
| BANK ($n = 41108, d = 6$) | 14% | 43% | 100% | 161% | 288% | 3,308% |
| CENSUS ($n = 20000, d = 66$) | 23% | 52% | 100% | 176% | 375% | 1,064% |

Additionally, we observe that computation time is linear in $m^2$, as shown in Figure 6 below.

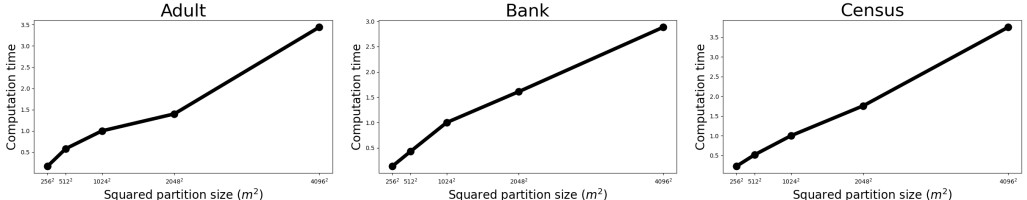

Figure 6: Squared partition size $m^2$ vs. Relative computation time. (Left, Center, Right) = (ADULT, BANK, CENSUS).

### C.3.4 ABLATION STUDY: OPTIMIZATION ALGORITHM OF CLUSTER CENTERS (SECTION 5.4)

We analyze how the FCA's performance varies depending on the optimization algorithm for finding cluster centers. For $K$-means algorithm, we use the $K$-means++ initialization (Arthur & Vassilvitskii, 2007) from the implementation of `scikit-learn` package Pedregosa et al. (2011). Note that we use the algorithm of Lloyd (1982) for $K$-means algorithm. We consider not only $K$-means++ initialization but additionally consider the random initialization of centers in this ablation study. For the gradient-based algorithm, we use Adam optimizer (Kingma & Ba, 2014). We set a learning rate of 0.005 for CENSUS dataset with $L_2$ normalization, and 0.05 for all other cases. To accelerate convergence, 20 gradient steps of updating the centers are performed per iteration.

Table 7 presents the results comparing these three approaches, showing that FCA is robust to the choice of the optimization algorithm for finding cluster centers. While the gradient-based algorithm is also effective and accurate, it requires additional practical considerations such as selections of the learning rate and optimizer.

Table 7: Comparison of performance with respect to optimization algorithms for finding cluster centers with $L_2$ normalization (top) and without $L_2$ normalization (bottom). '$K$-means random' indicates that the initial centers are set randomly at the first iteration, then apply the algorithm from Lloyd (1982). '$K$-means++' indicates that the initial centers are set according to the $K$-means++ initialization in the first iteration, then apply the algorithm from Lloyd (1982). 'Gradient-based' indicates that the initial centers are set randomly, and the centers are subsequently updated using the Adam optimizer.

| With $L_2$ normalization | ADULT | | BANK | | CENSUS | |
|---|---|---|---|---|---|---|
| | Cost | Balance | Cost | Balance | Cost | Balance |
| FCA ($K$-means random) | 0.331 | 0.997 | 0.265 | 0.998 | 0.477 | 0.992 |
| FCA ($K$-means++) | 0.328 | 0.997 | 0.264 | 0.998 | 0.477 | 0.993 |
| FCA (Gradient-based) | 0.339 | 0.993 | 0.254 | 0.986 | 0.478 | 0.979 |

| Without $L_2$ normalization | ADULT | | BANK | | CENSUS | |
|---|---|---|---|---|---|---|
| | Cost | Balance | Cost | Balance | Cost | Balance |
| FCA ($K$-means random) | 1.882 | 0.997 | 1.864 | 0.997 | 32.913 | 0.989 |
| FCA ($K$-means++) | 1.875 | 0.997 | 1.859 | 0.998 | 33.472 | 0.990 |
| FCA (Gradient-based) | 1.943 | 0.993 | 1.967 | 0.991 | 34.121 | 0.976 |

### C.3.5 ABLATION STUDY: INITIALIZATION OF CLUSTER CENTERS (SECTION 5.4)

Moreover, we empirically assess the robustness of FCA and FCA-C to the initialization of centers, and compare them with the standard $K$-means++ algorithm. For FCA and the standard $K$-means++ algorithm, we run each algorithm five times with five different random initial centers. For FCA-C, we use five random initial centers for each $\epsilon \in \{0.1, 0.15, \ldots, 0.85, 0.9\}$, and calculate the averages as well as standard deviations. Then, we divide the standard deviation by average 17 times (corresponding to 17 $\epsilon$s), and take average.

Table 8 below reports the coefficient of variation (= standard deviation divided by average) of `Cost` and `Balance`. The results show that the variations of all the three algorithms are similar, indicating that FCA and FCA-C are as stable as the standard $K$-means++ with respect to the choice of initial cluster centers. That is, aligning data to build fair clustering does not affect the stability of the overall algorithm at all.

Table 8: Standard deviations divided by averages (i.e., coefficient of variation) with respect to five random different choices of initial centers.

| FCA | ADULT | | BANK | | CENSUS | |
|---|---|---|---|---|---|---|
| | Cost | Balance | Cost | Balance | Cost | Balance |
| with $L_2$ | 0.012 | 0.001 | 0.093 | 0.006 | 0.004 | 0.001 |
| without $L_2$ | 0.010 | 0.001 | 0.081 | 0.003 | 0.006 | 0.003 |

| FCA-C | ADULT | | BANK | | CENSUS | |
|---|---|---|---|---|---|---|
| | Cost | Balance | Cost | Balance | Cost | Balance |
| with $L_2$ | 0.015 | 0.001 | 0.083 | 0.007 | 0.011 | 0.004 |
| without $L_2$ | 0.011 | 0.001 | 0.088 | 0.002 | 0.010 | 0.002 |

| $K$-means++ | ADULT | | BANK | | CENSUS | |
|---|---|---|---|---|---|---|
| | Cost | Balance | Cost | Balance | Cost | Balance |
| with $L_2$ | 0.009 | 0.001 | 0.078 | 0.005 | 0.008 | 0.002 |
| without $L_2$ | 0.011 | 0.000 | 0.090 | 0.004 | 0.010 | 0.002 |

### C.3.6 ABLATION STUDY: VARYING THE NUMBER OF CLUSTERS $K$ (SECTION 5.4)

We analyze the impact of $K$ to the performance of FC algorithms. On ADULT dataset, we evaluate the FC algorithms with $K \in \{5, 10, 20, 40\}$. The results are presented in Figure 7 below, which show that FCA outperforms existing FC algorithms across all values of $K$. In specific, FCA achieves lower values of `Cost` than baselines for most values of $K$, while maintaining the highest fairness level of `Balance` $\approx 1$.

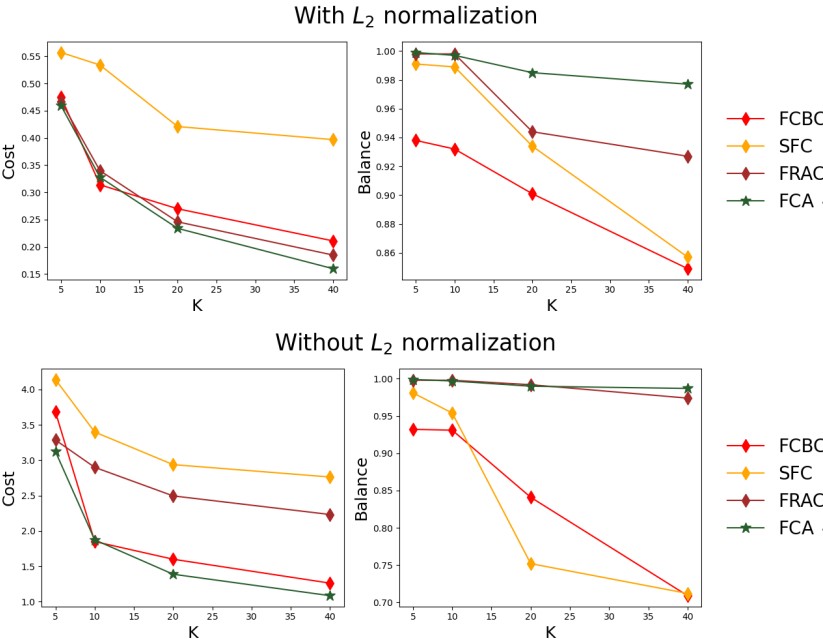

Figure 7: Performance of FC algorithms in terms of `Cost` and `Balance` for $K \in \{5, 10, 20, 40\}$. (Top, Bottom) = (With $L_2$ normalization, Without $L_2$ normalization). (Left, Right) = ($K$ vs. `Cost`, $K$ vs. `Balance`).

### C.3.7 COMPARISON OF COMPUTATIONAL COST

**FCA versus FCA-C**   We compare the computation times of FCA and FCA-C, as FCA-C technically involves an additional step of optimizing $\mathcal{W}$. Table 9 below shows that while FCA-C requires slightly more computation time than FCA, the increase is not substantial (a maximum of 3.8%). For this analysis, the data are $L_2$-normalized and the batch size is fixed as 1024.

Table 9: Comparison of computation times (seconds) of FCA and FCA-C per each iteration. The reported results are averages and standard deviations taken over five runs.

| Average (Standard deviation) | ADULT | BANK | CENSUS |
|---|---|---|---|
| FCA | 5.67 (0.39) | 7.31 (0.78) | 16.10 (2.70) |
| FCA-C | 5.72 (0.47) | 7.58 (0.32) | 16.46 (1.23) |

**FCA versus SFC**   We consider two scenarios to compare FCA and SFC as follows. For this analysis, the data are $L_2$-normalized and the batch size is fixed as 1024 for FCA. Note that FCA consists of an outer iteration (updating cluster centers and joint distribution alternately), and an inner iteration when using $K$-means algorithm.

1. We compare the number of iterations until convergence. For SFC, we calculate the number of iterations in the $K$-means algorithm (after finding fairlets). For FCA, we calculate the sum of the number of iterations (inner iteration) in $K$-means algorithm for each outer iteration (updating cluster centers and joint distribution). In this analysis, note that we report the elapsed time for FCA with 10 iterations of the outer iteration, because the performance of FCA with 10 iterations of the outer iteration is not significantly different from FCA with 100 iterations of the outer iteration.

   As a result, SFC requires smaller number of iterations, compared to FCA (see Table 10), primarily because FCA involves the outer iterations. On the other hand, FCA is almost linear with respect to the number of K-means algorithm iterations until convergence.

2. To further analyze whether using early-stopping in FCA can maintain reasonable performance, we conduct an additional analysis: we fix the number of $K$-means iterations to 1 per outer iteration of FCA, then perform a total of 10 outer iterations. With this setup, the total number of iterations for FCA becomes 10, which is comparable to or smaller than that of SFC (15 for Adult, 10 for Bank, and 32 for Census dataset, as shown in Table 10). We observe that the performance of FCA with this early-stopping is slightly worse than the original FCA (where $K$-means algorithm runs until convergence), but it still outperforms SFC (see Table 11). However, note that this early-stopping approach would be not recommended, at least, for the datasets used in our experiments. This is because running $K$-means algorithm takes less than a second or a few seconds, while the computation time of finding joint distribution dominates the time of running $K$-means algorithm until convergence.

Table 10: Comparison of computational costs between FCA and SFC: Total number of iterations in the $K$-means algorithm.

| Total number of iterations | ADULT | BANK | CENSUS |
|---|---|---|---|
| SFC (Backurs et al., 2019) | 15 | 10 | 32 |
| FCA | 57 | 110 | 93 |

Table 11: Performance comparison of SFC, FCA with a total of 10 iterations, and the original FCA (updating until convergence).

| Dataset | ADULT | | BANK | | CENSUS | |
|---|---|---|---|---|---|---|
| C = Cost, B = Balance | C ($\downarrow$) | B ($\uparrow$) | C ($\downarrow$) | B ($\uparrow$) | C ($\downarrow$) | B ($\uparrow$) |
| SFC (Backurs et al., 2019) | 3.399 | 0.954 | 3.236 | 0.957 | 69.437 | 0.973 |
| FCA (total 10 iterations) | 1.923 | 0.996 | 1.992 | 0.995 | 33.967 | 0.990 |
| FCA (original) | 1.875 | 0.997 | 1.859 | 0.998 | 33.472 | 0.990 |

### C.3.8 Comparison of FCA and SFC based on $K$-median clustering cost

As SFC is originally designed based on the $K$-median clustering objective, for a more fair comparison, we compare FCA and SFC based on the $K$-median clustering cost (i.e., $L_1$ cost). In specific, FCA for $K$-median clustering cost is modified as follows: (i) The $L_2$ norm in eq. (3) is replaced by the $L_1$ norm. (ii) The cluster centers and found by minimizing the $L_1$ distance, as we discuss in Section A.4 of Appendix.

The results are presented in Table 12, which show that FCA still outperforms SFC. It implies that the fairlet-based method still may not always find the optimal matching, even when a clustering objective more suited to fairlet-based approaches (e.g., $L_1$ norm) is considered.

Let $\texttt{Cost}_1 = \frac{1}{n} \sum_{(\mathbf{x},s) \in \mathcal{D}} \|\mathbf{x} - \mu_{k(\mathbf{x},s)}\|_1$ be the $K$-median clustering cost.

Table 12: Comparison of $\texttt{Cost}_1$ and $\texttt{Balance}$ of FCA and SFC. The data are not $L_2$-normalized.

| Dataset | ADULT | | BANK | | CENSUS | |
|---|---|---|---|---|---|---|
| $\texttt{C}_1 = \texttt{Cost}_1, \texttt{B} = \texttt{Balance}$ | $\texttt{C}_1 (\downarrow)$ | $\texttt{B} (\uparrow)$ | $\texttt{C}_1 (\downarrow)$ | $\texttt{B} (\uparrow)$ | $\texttt{C}_1 (\downarrow)$ | $\texttt{B} (\uparrow)$ |
| Standard (fair-unaware) | 1.788 | 0.418 | 1.989 | 0.620 | 21.402 | 0.043 |
| SFC (Backurs et al., 2019) | 2.979 | 0.950 | 3.056 | 0.827 | 29.597 | 0.913 |
| FCA ✓ | 2.032 | 0.993 | 2.383 | 0.993 | 22.927 | 0.978 |

### C.3.9   PERFORMANCE COMPARISON ON A LARGE DATASET

In this section, we evaluate FCA's scalability for larger datasets, while the main experiments in Section 5 are conducted on real datasets with sample sizes of around $20,000$ to $40,000$. In specific, we apply FCA on a synthetic dataset with an extremely large sample size (around a million).

**Dataset generation**   We generate a large ($n = 10^6$) synthetic dataset in $\mathbb{R}^d$ using a $J$-component Gaussian mixture, as follows:

1. (Mean vectors) We sample $J$ many $d$-dimensional vectors $m_j, j \in \{1, \ldots, J\}$ from a uniform distribution $\mathrm{Unif}(-20, 20)$. To ensure diversity, the distance between any two vectors is at least 1. These vectors are used as the mean vectors for the Gaussian components.

2. (Covariance matrix) Each $j$th Gaussian component is assigned a covariance matrix $\Sigma_j = \sigma_j^2 \mathbb{I}$, where $\sigma_j \sim \mathrm{Unif}(1, 3)$.

3. (Weights) Component weights, denoted as $\phi_j, j \in \{1, \ldots, J\}$, are sampled from a Dirichlet distribution $\mathrm{Diri}(\alpha_1, \ldots, \alpha_J)$ for given parameters $\alpha_1, \ldots, \alpha_J$.

4. (Completion) The Gaussian mixture model is now completed as $\sum_{j=1}^{J} \phi_j \mathcal{N}(m_j, \Sigma_j)$. We set $J$ as an even number, and sample data for $S = 0$ from $J/2$ components and for $S = 1$ from the remaining $J/2$ components.

Using this procedure, we construct a dataset with $n = 10^6, d = 2, J = 20$, and $\alpha_j = 1, \forall j \in \{1, \ldots, J\}$. The resulting dataset contains 320,988 samples for $S = 0$ and 679,012 samples for $S = 1$. All features are scaled to have zero mean and unit variance. Figure 8 provides a visualization of this synthetic dataset.

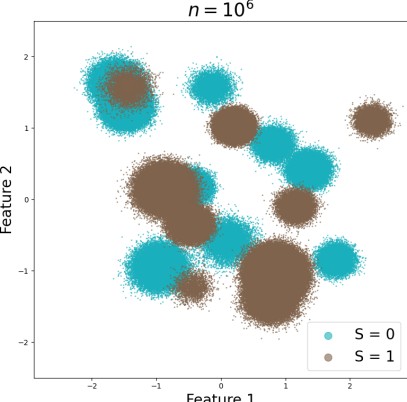

Figure 8: The large synthetic dataset with $n = 10^6, d = 2, J = 20$, and $\alpha_j = 1, \forall j \in \{1, \ldots, J\}$.

**Results**   We fix the number of clusters to $K = 10$. We compare FCA and VFC in this analysis, as other baselines incur extremely high computational costs for this dataset. Table 13 presents the results, showing that FCA can be scaled-up and is still a favorable FC algorithm for large datasets. In contrast, VFC still fails to achieve near-perfect fairness, while FCA-C also outperforms VFC at the maximum achievable `Balance` for VFC.

Table 13: Comparison of `Cost` and `Balance` between FCA (or FCA-C) and VFC on the large synthetic dataset in Figure 8.

| C = Cost, B = Balance | C (↓) | B (↑) |
|---|---|---|
| Standard (fair-unaware) | 0.058 | 0.000 |
| VFC (Ziko et al., 2021) ($\lambda = 51000$) | 0.111 | 0.154 |
| FCA-C ($\epsilon = 0.65$) ✓ | 0.107 | 0.155 |
| FCA ✓ | 0.669 | 0.997 |

