# OpenReview forum: "Fair Clustering via Alignment"
_ICLR.cc/2025/Conference — Submitted to ICLR 2025_

### Official Review · Reviewer_C9w4 · 2024-10-29

**Soundness:** 2
**Presentation:** 3
**Contribution:** 3
**Rating:** 5
**Confidence:** 5

**Summary:**

This paper studies the fair clustering problem and introduces a new algorithm, FCA, to address the trade-off between fairness and clustering utility in clustering algorithms. FCA aligns data from different protected groups into a common space, allowing for optimal clustering while adhering to fairness constraints. The authors prove that this approach achieves better clustering utility and stability compared to existing methods. Empirically, FCA outperforms baseline algorithms in cost-fairness tradeoff.

**Strengths:**

- The paper presents the FCA algorithm, which effectively balances fairness and clustering utility, achieving optimal results without compromising either aspect. This addresses a significant challenge in fair clustering.

- The paper provides a theoretical analysis of the FCA algorithm, proving that optimal fair clustering can be achieved through alignment in a common space.

- Empirically, FCA is shown to be robust against variations in data preprocessing, optimization algorithms, and initialization of cluster centers, ensuring consistent performance across different scenarios.

**Weaknesses:**

The FCA algorithm's optimality is not clear. It seems that the algorithm does not find the optimal fair clustering. I suggest including a detailed comparison of the notion of optimality and the commonly used optimal solution.

- Sec 3 mainly discusses the case of perfect fair clustering, in which we can construct mappings between different groups. However, it is more common that there is a group-wise lower and upper bound for the ratio of each group in a cluster. How do we handle this case?

**Questions:**

- Could you give examples of what settings FCA can actually find an optimal fair clustering and what settings it does not?

- Instead of perfect fair clustering, e.g., we only require the balance to be at least 0.8, is there any theoretical result?

---

> ### Author Response · Authors · 2024-11-22
> **Point-by-point responses**
>
> ### Weaknesses
>
> - - - - - -
>
> > **W1**: The FCA algorithm's optimality is not clear. It seems that the algorithm does not find the optimal fair clustering. I suggest including a detailed comparison of the notion of optimality and the commonly used optimal solution.
>
>
> *Answer*
>
>    As Lloyd's $K$-means algorithm has the **descent property (i.e., the cost decreases with each iteration; Lloyd (1982); Selim \& Ismail (1984)), FCA inherits this property as well and thus converges to an optimum.**
>         The solution would be a local optimum, because FCA depends on the initial cluster centers (as well as Lloyd's algorithm) and involves optimizing the joint distribution.
>
>    To clarify this point, we modify the abstract and introduction by adding the term `(local) optimum' where appropriate.
>
>    Experimentally, we observe that using **Lloyd's algorithm with $K$-means++ initialization in FCA results in more effective performance** compared to baseline methods (FCA tends to find a good local optimum).
>         Moreover, in Table 7 in Section C.3.4 of Appendix, we compare three algorithms: $K$-means (Lloyd's) algorithm with $K$-means++ initialization and random initialization (added newly in the revised paper), and a gradient-based optimization.
>         The results show that $K$-means++ initialization provides superior performance.
>
> - - - - - -
>
> > **W2**: Sec 3 mainly discusses the case of perfect fair clustering, in which we can construct mappings between different groups. However, it is more common that there is a group-wise lower and upper bound for the ratio of each group in a cluster. How do we handle this case?
>
>
> *Answer*
>
>    While FCA (in Section 4.1) finds the perfectly fair optimal clustering, **FCA-C (in Section 4.2) is specifically designed to handle such non-perfect fairness cases.**
>         FCA-C controls (non-perfect) fairness level by applying fairness constraint only to a subset of the aligned space, as formulated in eq. (5) and (6).
>         **The fairness level is controlled by controlling the size of this subset.**
>
>    Moreover, we would like to emphasize that **FCA-C can find the optimal clustering that minimizes cost for a given (non-perfect) fairness level, as shown in Theorem 4.1.**
>         This optimality is achieved by optimizing the selection of members in the subset.
>
>
> - - - - - -
> - - - - - -
> ### Questions
>
> - - - - - -
>
> > **Q1**: Could you give examples of what settings FCA can actually find an optimal fair clustering and what settings it does not?
>
>
> *Answer*
>
>
>
>    FCA would find a local optimum, since Lloyd's algorithm for $K$-means clustering is used.
>    Even though the solution of FCA would be a local optimum, we experimentally confirm that the solution of FCA is consistently better than that of baseline algorithms, in terms of the trade-off between cost and fairness level.
>    Please also see our response to W1.
>
> - - - - - -
>
> > **Q2**: Instead of perfect fair clustering, e.g., we only require the balance to be at least 0.8, is there any theoretical result?
>
>
> *Answer*
>
>
>    In fact, Theorem 4.1 is the theoretical result showing that FCA-C can optimally control the trade-off between cost and (non-perfect, Balance $< 1$) fairness level.
>    Please also see our response to W2.

---

> ### Author Response · Authors · 2024-11-28
>
> *Dear reviewer C9w4*:
>
> Once again, thank you very much for your review, which has been greatly helpful for our work.
> We hope this post doesn't cause any inconvenience, but we have about 5-6 hours left to make revisions to the paper's PDF (although discussions can continue for a few more days).
>
> We would like to ask whether our responses have addressed your concerns or if there are any remaining discussion points that require further revisions to the paper (minor points can still be discussed over the next few days!).
>
> Thank you.

---

### Official Review · Reviewer_7dQE · 2024-10-31

**Soundness:** 3
**Presentation:** 3
**Contribution:** 2
**Rating:** 5
**Confidence:** 4

**Summary:**

This paper studies fair clustering problem and gives an algorithmic framework that takes a distributional view and transform the fair clustering problem into a variant of optimal transport problem. Once the relation to the optimal transport is established, one can formulate an LP and solve it in poly(n) time. Actually, I think the paper gives a new fair clustering objective/formulation (which is different to the widely used proportional fairness), particularly a notion called ``balance’’, which measures how close the conditional expectation with respect to any given sensitive attribute assignment, to the unconditional one. The new algorithm is designed for this objective.

This paper also conducts experiments to validate the effectiveness of the proposed algorithm, and it compares with several recent approximation algorithms for fair clustering as baselines. It seems the new algorithm can outperform baselines in their experiment setup. A further applications to visual clustering is showcased.

**Strengths:**

- The paper offers a new view to fair clustering
- Both theoretical and experimental results are presented
- This new formulation seems to evade the NP-hardness of proportional fair clustering, which may be a useful feature, though I might be wrong

**Weaknesses:**

- The relation to the widely considered proportional fairness is not formally addressed
- It seems the method can only deal with a binary sensitive attribute, whereas many previous works can systematically solve arbitrary groups and sensitive attributes
- The case of general “balance” seems to be weak (Corollary 4.2), and no worst-case bound that only depends on the magnitude of input is given.

**Questions:**

- Your problem requires the input of a distribution of input. In many other fair clustering papers, the input is simply a point set and the optimization is done only on one input. How are the two settings related? Notice that in your experiments you do a comparison to those algorithms, then how do you set the input accordingly?

- For your phase 2, you mentioned that one could use k-means++ to update \mu. However, is there a formal treatment of this? In particular, does the O(\log k) approximation ratio carries on to your case?

- The FCA-C algorithm seems to use many more steps than FCA. How efficient can we implement this?

- How do you measure the balance of the baselines in the experiments? I don’t think they work with a distributional input

**Details Of Ethics Concerns:**

None.

---

> ### Author Response · Authors · 2024-11-22
> **Point-by-point responses #1**
>
> ### Weaknesses
>
> - - - - - -
>
> > **W1**: The relation to the widely considered proportional fairness is not formally addressed.
>
>
> - *Answer*
>
>    We acknowledge that there are various fairness notions in clustering, such as group fairness, proportional fairness, and individual fairness.
>    Among these, this paper focuses on the group fairness notion (fairness with respect to the ratio of protected groups within each cluster).
>
>    While our proposed framework would not be directly extended to other notions, our core idea of alignment could potentially be adapted for them.
>    We believe exploring such extensions or relationships would be a very promising topic for future research.
>
>    Instead, in the revised paper, **we add a new section (Section A.5 of Appendix) introducing other fairness notions in clustering, including proportional fairness and individual fairness.**
>    Due to the page limit, this section is included in Appendix, rather than the main body.
>
> - - - - - -
>
> > **W2**: It seems the method can only deal with a binary sensitive attribute, whereas many previous works can systematically solve arbitrary groups and sensitive attributes
>
>
> - *Answer*
>
> In fact, **we can extend FCA to handle multiple (more than two) protected groups.**
> Note that this paper mainly focuses on the case of two protected groups, for ease of discussion.
> The main idea is similar: matching multiple individuals from different protected groups.
>
> Let $G$ be the number of protected groups, and denote $s\^{\*} \\in \\{0, \ldots, G-1\\}$ as a fixed (reference) sensitive attribute.
> For the case of equal sample sizes that $n\_{0} = \ldots = n\_{G-1},$ we can similarly decompose the objective function in terms of matching map as follows, which we prove in Section A.3 of Appendix:
> $$ C(\mu, \mathcal{A}\_{0}, \ldots, \mathcal{A}\_{G-1})
>                 = \mathbb{E} \sum\_{k=1}\^{K} \mathcal{A}\_{S}(\mathbf{X}) \Vert \mathbf{X} - \mu\_{k} \Vert\^{2}
> $$
> $$
> = \\mathbb{E}\_{s\^{\*}} \\sum\_{k=1}\^{K} \\mathcal{A}\_{s\^{\*}} (\\mathbf{X})\_{k} \\frac{1}{G}
> ( \\sum\_{s=0}\^{G-1} \\Vert \\frac{ \\sum\_{s’ \\neq s} (\\mathbf{T}\_{s}(\\mathbf{X}) - \\mathbf{T}\_{s’}(\\mathbf{X})) }{G} \\Vert\^{2} + \\Vert \\frac{ \\mathbf{X} + \\sum\_{s\^{’} \\neq s\^{\*}} \\mathbf{T}\_{s\^{’}}(\\mathbf{X}) }{G} - \\mu\_{k} \\Vert\^{2}  ),
> $$
>
> where $\mathbf{T}\_{s’}$ is the one-to-one matching map from
> $\mathcal{X}\_{s\^{\*}}$ to $\mathcal{X}\_{s’}$ for $s' \\in \\{0, \ldots, G-1\\}.$
> Note that this result holds for any $s\^{\*} \\in \\{0, \ldots, G-1 \\}.$
>
> For the case of unequal sample sizes, we expect that a similar decomposition (i.e., eq (A) below) would be valid.
> We first define the alignment map as:
>         $
>         \mathbf{T}\^{\textup{A}} (\mathbf{x}\_{0}, \ldots, \mathbf{x}\_{G-1})
>         := \pi\_{0} \mathbf{x}\_{0} + \ldots + \pi\_{G-1} \mathbf{x}\_{G-1}.
>         $
> Then, we find the joint distribution and cluster centers by minimizing the following objective (A):
>         $$
>             \mathbb{E}\_{ (\mathbf{X}\_{0}, \ldots, \mathbf{X}\_{G-1}) \sim \mathbb{Q}}
>             \left(
>             \sum\_{s = 0}\^{G-1} G \pi\^{G} \bigg\Vert \sum_{s' \neq s} \left( \mathbf{X}\_{s} - \mathbf{X}\_{s’} \right) \bigg\Vert\^{2}
>             + \min\_{k} \Vert \mathbf{T}\^{\textup{A}} ( \mathbf{X}\_{0}, \ldots, \mathbf{X}\_{G-1} ) - \mu\_{k} \Vert^{2}
>             \right),
>             $$
> where $\pi\^{G} := \prod\_{s=0}\^{G-1} \pi\_{s}.$
> Furthermore, since finding the joint distribution for multiple protected groups is technically similar to the case of two protected groups, the optimization of the objective in eq. (A) can be solved using a linear program.
> Unfortunately, we have not succeeded in proving (A), which we are trying now.
>
> We newly add this feasible extension of FCA, with details in Section A.3 of Appendix.
>
> - - - - - -
>
> > **W3**: The case of general “balance” seems to be weak (Corollary 4.2), and no worst-case bound that only depends on the magnitude of input is given.
>
>
> - *Answer*
>
>    In fact, the **Balance defined in Definition 2.1 explicitly captures the worst-case with respect to clusters $\times$ protected groups**.
>         That is, Balance is defined by $\min_{s} \min_{k}$ of (the probability of assigning to the $k$th cluster for group $s$ $\div$ the probability of assigning to the $k$th cluster for the entire dataset).
>
>    Moreover, as we mentioned in Section 2, this definition is widely used in the fair clustering literature for group fairness (Chierichetti et al., 2017; Bera et al., 2019; Backurs et al., 2019; Esmaeili et al., 2021; Ziko et al., 2021; Zeng et al., 2023), so we believe this measure is appropriate for evaluating group fairness.

---

> ### Author Response · Authors · 2024-11-22
> **Point-by-point responses #2**
>
> ### Questions
>
>
> - - - - - -
>
> > **Q1**: Your problem requires the input of a distribution of input. In many other fair clustering papers, the input is simply a point set and the optimization is done only on one input. How are the two settings related? Notice that in your experiments you do a comparison to those algorithms, then how do you set the input accordingly?
>
>
> - *Answer*
>
>    We would like to clarify that **there is no difference between FCA and existing methods in terms of input: FCA also takes a given dataset as input.** The term `distribution' defined in this paper simply refers to the empirical distribution defined over the given dataset.
>
>    The distribution is defined in this paper to discuss (i) probabilistic matching, which is similar to the optimal transport problem (defined over two probability measures), and (ii) subsequent probabilistic assignments for the case of unequal sample sizes.
>
>
>    To avoid such confusion, we newly add the following statement in Section 2: *We specifically define these distributions to discuss the (probabilistic) matching between two protected groups of different sizes, and subsequent probabilistic assignments for fair clustering.*
>
> - - - - - -
>
> > **Q2**: For your phase 2, you mentioned that one could use k-means++ to update $\mu$. However, is there a formal treatment of this? In particular, does the $O(\log k)$ approximation ratio carries on to your case?
>
>
> - *Answer*
>
>    As FCA alternately updates the joint distribution and cluster centers, we would not guarantee the approximation ratio of $K$-means++.
>    However, as Lloyd's $K$-means algorithm has **the descent property** (i.e., the cost decreases with each iteration; Lloyd (1982); Selim \& Ismail (1984)), FCA inherits this property as well and thus converges to an optimum.
>    The solution would be a local optimum, because FCA depends on the initial cluster centers (as well as Lloyd's algorithm) and involves optimizing the joint distribution.
>    To clarify this point, we modify the abstract and introduction by adding the term `(local) optimum' where appropriate.
>
>    Experimentally, we observe that using Lloyd's algorithm with **$K$-means++ initialization in FCA results in more effective performance compared to baseline methods (FCA tends to find a good local optimum).**
>    Moreover, in Table 7 in Section C.3.4 of Appendix, we compare three algorithms: $K$-means (Lloyd's) algorithm with $K$-means++ initialization and random initialization (added newly in the revised paper), and a gradient-based optimization.
>    The results show that $K$-means++ initialization provides superior performance.
>
>    Developing an algorithm with a low approximation error or the capability to find a global optimum, building upon FCA, would be an interesting topic for future work.
>
>
> - - - - - -
>
> > **Q3**: The FCA-C algorithm seems to use many more steps than FCA. How efficient can we implement this?
>
>
> - *Answer*
>
>    Compared to FCA, FCA-C technically performs only one additional step: constructing $\mathcal{W},$ the set of data points to which the non-fair cost (i.e., $K$-means cost) is applied.
>    The construction of $\mathcal{W}$ involves just the calculation of FCA cost (i.e., $\eta$ in line 322) followed by a single sorting operation, which is straightforward.
>
>    We also experimentally confirm that, **FCA-C does not require significantly more computational cost compared to FCA**:
>    Table 9 in Section C.3.7 of Appendix shows that the computation time of FCA-C is higher than FCA by at most 3.8\%.
>
>
> - - - - - -
>
> > **Q4**: How do you measure the balance of the baselines in the experiments? I don’t think they work with a distributional input.
>
>
> *Answer*
>
>    The **calculation of the balance is done once the clusters (i.e., the assignments for each data point) are given, regardless of algorithm used or data distributions**. Please also see our response to Q1.

---

> > ### Comment · Reviewer_7dQE · 2024-11-22
> >
> > Thanks for the response. Your answer about “distribution” might suggest that I have a major misunderstanding. So let me ask the following followup questions.
> >
> > 1. How does your notion of fairness compare with that used in Chierichetti et al., 2017, Bera et al., 2019? Can yours imply theirs, i.e., if some clustering is fair in your notion, then is it fair (w.r.t. some parameter) in their notion? Moreover, does your result imply any new approximation algorithm under their setting of “fair clustering”, and does existing results, such as Bera et al. 2019, imply any approximation algorithm for your setting?
> >
> > (By the way, in my previous post, by “proportionally fair”, I actually meant Chierichetti et al., 2017 line of research.)
> >
> > 2. For the new part about multiple groups, it does not look very conclusive to me. You say that you may be able to solve (A), but you also said you have not succeeded in proving (A) — what does this mean? Also, if you solve (A), then how do we generate a final fair clustering?

---

> > > ### Author Response · Authors · 2024-11-23
> > > **Follow-up response #1**
> > >
> > > *Reviewer 7dQE:*
> > > Thank you for the follow-up discussion!
> > > We hope our response below resolves your concerns.
> > >
> > > -------
> > > > **1.** How does your notion of fairness compare with that used in Chierichetti et al., 2017, Bera et al., 2019? Can yours imply theirs, i.e., if some clustering is fair in your notion, then is it fair (w.r.t. some parameter) in their notion? Moreover, does your result imply any new approximation algorithm under their setting of “fair clustering”, and does existing results, such as Bera et al. 2019, imply any approximation algorithm for your setting?
> > >
> > > - *Answer*:
> > >
> > >   - **(About the fairness notion)**
> > >    **The fairness notion considered in this paper is conceptually equivalent to the notion of Chierichetti et al., (2017) and subsequent research in this line.**
> > >         Specifically, for any given fair clusters under the existing notion (i.e., where the proportion of $S=0$ and $S=1$ in each cluster equals $n_{0} / n_{1}$),
> > >         the clusters are also fair under our notion (i.e., our balance = 1), and vice versa.
> > >
> > >         Furthermore, **the balance defined in this paper is simply a normalized version of the balance defined in prior works, and is therefore not fundamentally different.**
> > >         Note that DFC (Li et al., 2020), which considers the same fairness notion as Chierichetti et al., (2017), used a definition of balance similar to ours.
> > >         The reason for the normalization is as follows:
> > >         under the assumption that $n_0 < n_1$, the balance value based on the existing definition has a maximum of $n_0 / n_1$, making it dependent on the given dataset.
> > >         In contrast, our definition of balance tightly lies in $[0, 1]$ regardless of the given dataset, i.e., can achieve a maximum value of 1.
> > >         This provides a clearer presentation (closer to 1 indicates greater fairness, and closer to 0 indicates greater unfairness).
> > >
> > >    - **(About the approximation)**
> > >         Thank you for the important discussion.
> > >         We can investigate the relationship between our framework and the approximation algorithm in Bera et al. (2019) as follows:
> > >
> > >         The approximation algorithm in Bera et al., (2019) finds fair assignments by solving a relaxed linear programming (LP) problem, for given fixed (opened) cluster centers.
> > >         In the case of equal sample sizes ($n_{0} = n_{1}$) and for perfect fairness, solving the LP of Bera et al., (2019) finds the optimal (in terms of minimizing the cost) fair assignments, while the fairness is ensured by implicitly matching data from the two protected groups (for satisfying the fairness constraint).
> > >         Hence, this mechanism is conceptually equivalent to ours (when the cluster centers are fixed).
> > >
> > >         However, the gap between fair assignment (with fixed cluster centers) and fair clustering results in additional approximation error (as shown in Lemma 3 of Bera et al., (2019)), because the cluster centers are not optimized for the fair assignment problem.
> > >         In contrast, our proposed algorithm FCA iteratively optimizes both cluster centers and matching until convergence (rather than performing this process only once).
> > >         As a result, the cost of FCA could be no larger than the cost of Bera et al. (2019), i.e., FCA could have an approximation error that is at most equal to that of Bera et al. (2019).

---

> > > ### Author Response · Authors · 2024-11-23
> > > **Follow-up response #2**
> > >
> > > > **2.** For the new part about multiple groups, it does not look very conclusive to me. You say that you may be able to solve (A), but you also said you have not succeeded in proving (A) — what does this mean? Also, if you solve (A), then how do we generate a final fair clustering?
> > >
> > > - *Answer*:
> > >
> > >   - **(About the proof for (A))**
> > >    The phrase 'have not succeeded' in our previous response does not imply 'failed'.
> > >         We simply posted the response before fully completing the proof to initiate this rebuttal discussion as quickly as possible, with the plan to update the proof before this discussion period ends.
> > >
> > >         Fortunately, we have now completed the proof, which is a straightforward extension of the proof for Theorem 3.3.
> > >         It is now added to Section A.3 of Appendix.
> > >
> > >   - **(Completion of fair clustering based on (A))**
> > >    After solving (A), the fair clustering is constructed similarly to the case of two protected groups, as follows.
> > >    Let $\mathbf{x}$ be a data point in $\\mathcal{X}\_{s}$ (i.e., from the $s$th protected group).
> > >    Without loss of generality, let $s = 0.$
> > >    Then, for the cluster centers $\mu\^{\*}$ and joint distribution $\mathbb{Q}\^{\*}$ that minimize (A), we assign $\mathbf{x}$ to $k$th cluster with probability $\mathbb{Q}\^{\*}(argmin\_{k'} \Vert \mathbf{T}\^{\textup{A}}(\mathbf{x}, \mathbf{X}\_{1}, \ldots, \mathbf{X}\_{G-1}) \Vert\^{2} = k \vert \mathbf{X}\_{0} = \mathbf{x}).$
> > >    Note that this probability represents the ratio of aligned points assigned to $k$th cluster among all $\\prod\_{s = 1}\^{G-1} n\_{s}$-many aligned points (with positive weights) of $\\mathbf{x},$
> > >    i.e., $\\{ \pi\_{0} \mathbf{x} + \sum\_{s = 1}^{G-1} \pi\_{s} \mathbf{x}\_{i\_{s}}\^{(s)} \\}\_{i\_{s} \in [n\_{s}], s \in \\{1, \ldots, G-1\\} },$
> > >     and the fairness is satisfied through the use of aligned space.
> > >
> > >     We also add this point to Section A.3 of Appendix.

---

> > > > ### Comment · Reviewer_7dQE · 2024-11-25
> > > >
> > > > Thanks for the response. Now I have a more clear view of your setting/result.
> > > >
> > > > I agree that your notion of fairness is related to Chierichetti et al., (2017). Your paper starts with the perfectly balanced case, and the idea is to use optimal transport + k-means++ for the optimization. This sounds like a new approach, but unfortunately the worst-case guarantee, i.e., approximation ratio, is not established. The approach seems to generalize to the case of allowing some lower bound t \leq 1 in the balance/fairness. The presentation can be improved to emphasize better the relation to the previous notions of fairness.
> > > >
> > > > By the way, in my mind, I was always thinking about a generalized version of Chierichetti et al., (2017), given by Bera et al. (2019): for the i-th cluster (i \in [k]), the proportion of S = 0 and S = 1 is between some given [\alpha_i, \beta_i]. Your notion seems to be weaker than this, but if you do not care about the ratio, then your overall approach may still generalize.
> > > >
> > > > Bottomline: this paper does not propose a brand new fairness notion (and the notion is essentially Chierichetti et al., (2017)), and the focus is to compute the fair clustering more efficiently. However, the proposed algorithm does not have a worst-case guarantee of approximation ratio, which is the main weakness, especially considering Chierichetti et al., (2017) already designed an algorithm that has O(1)-approximation ratio (albeit it is parameterized by the balance factor). I can see from the experiments the new algorithm outperforms many existing baselines, but the improvement does not seem to be significantly better.
> > > >
> > > > Overall, I would slightly increase my rating, but I still incline to rejection.

---

> > > > > ### Author Response · Authors · 2024-11-26
> > > > > **Proof of Theorem A #1**
> > > > >
> > > > > #### **(Full proof for Theorem A – part 1)**
> > > > >
> > > > > It suffices to show that there exist $\mathcal{A}\_{0}\^{+}$ and $\mathcal{A}\_{1}\^{+}$ such that
> > > > > (i) $(\mathcal{A}\_{0}\^{+}, \mathcal{A}\_{1}\^{+}) \in \mathbf{A}\_{\epsilon}$,
> > > > > (ii) $C( \mu\^{\diamond}, \mathcal{A}\_{0}\^{+}, \mathcal{A}\_{1}\^{+} ) \le (\tau + 2) C\^{*},$
> > > > > and
> > > > > (iii) $C( \mu\^{\diamond}, \mathcal{A}\_{0}', \mathcal{A}\_{1}' ) \le C( \mu\^{\diamond}, \mathcal{A}\_{0}\^{+}, \mathcal{A}\_{1}\^{+} ).$
> > > > >
> > > > > Recall that $\tilde{\mu} = \\{ \tilde{\mu}\_{k} \\}\_{k=1}\^{K} $ and $\mu\^{\diamond} = \\{ \mu\_{k}\^{\diamond} \\}\_{k=1}\^{K}.$
> > > > > For all $k \in [K], $ we define the set of nearest neighbors of $\mu\_{k}\^{\diamond}$ as $N(\mu\_{k}\^{\diamond}) := \\{ \tilde{\mu}\_{k} \in \tilde{\mu} : argmin\_{\mu\_{k'}\^{\diamond} \in \mu\^{\diamond}} \Vert \tilde{\mu}\_{k} - \mu\_{k'}\^{\diamond} \Vert\^{2} = \mu\_{k}\^{\diamond} \\}.$
> > > > > First, for $\mathbf{x} \in \mathcal{X},$ we define $\mathcal{A}\_{s}\^{+}(\mathbf{x})\_{k} := \sum\_{k': \tilde{\mu}\_{k'} \in N(\mu\_{k}\^{\diamond}) } \tilde{\mathcal{A}}\_{s}(\mathbf{x})\_{k'}.$
> > > > >
> > > > > By the definitions of $\tilde{\mathcal{A}}\_{0}$ and $\tilde{\mathcal{A}}\_{1},$ we have $\sum\_{k=1}\^{K} \left\vert \frac{1}{n_{0}} \sum\_{i=1}\^{n_{0}} \tilde{\mathcal{A}}\_{0}(\mathbf{x}\_{i}\^{(0)})\_{k} - \frac{1}{n\_{1}} \sum\_{j=1}\^{n\_{1}} \tilde{\mathcal{A}}\_{1} (\mathbf{x}\_{j}\^{(1)})\_{k} \right\vert \le \epsilon.$
> > > > > Then,
> > > > > $$ \sum\_{k=1}\^{K} \left\vert \frac{1}{n_{0}}  \sum\_{i=1}\^{n_{0}} \mathcal{A}\_{0}\^{+}(\mathbf{x}\_{i}\^{(0)})\_{k} - \frac{1}{n_{1}} \sum\_{j=1}\^{n_{1}} \mathcal{A}\_{1}\^{+} (\mathbf{x}\_{j}\^{(1)})\_{k} \right\vert = \sum\_{k=1}\^{K} \left\vert \sum_{k': \tilde{\mu}\_{k'} \in N(\mu\_{k}\^{\diamond})} \left( \frac{1}{n_{0}} \sum\_{i=1}\^{n_{0}} \tilde{\mathcal{A}}\_{0}(\mathbf{x}\_{i}\^{(0)})\_{k'} - \frac{1}{n_{1}} \sum\_{j=1}\^{n\_{1}} \tilde{\mathcal{A}}\_{1} (\mathbf{x}\_{j}\^{(1)})\_{k'}  \right) \right\vert $$
> > > > > $$ \le \sum\_{k=1}\^{K} \sum\_{k': \tilde{\mu}\_{k'} \in N(\mu\_{k}\^{\diamond})} \left\vert \frac{1}{n\_{0}} \sum\_{i=1}\^{n\_{0}} \tilde{\mathcal{A}}\_{0}(\mathbf{x}\_{i}\^{(0)})\_{k'} - \frac{1}{n_{1}} \sum\_{j=1}\^{n_{1}} \tilde{\mathcal{A}}\_{1} (\mathbf{x}\_{j}\^{(1)})\_{k'} \right\vert = \sum\_{k'=1}\^{K} \left\vert \frac{1}{n\_{0}} \sum\_{i=1}\^{n_{0}} \tilde{\mathcal{A}}\_{0}(\mathbf{x}\_{i}\^{(0)})\_{k'} - \frac{1}{n\_{1}} \sum\_{j=1}\^{n\_{1}} \tilde{\mathcal{A}}\_{1} (\mathbf{x\}_{j}\^{(1)})\_{k'} \right\vert \le \epsilon, $$
> > > > > where the last equality holds because $\cup\_{k=1}\^{K} \cup\_{k': \tilde{\mu}\_{k'} \in N(\mu\_{k}\^{\diamond})} \\{ k' \\} = \\{ k' \\}\_{k'=1}\^{K}.$
> > > > > Hence, (i) is proved.

---

> > > > > ### Author Response · Authors · 2024-11-26
> > > > > **Proof of Theorem A #2**
> > > > >
> > > > > #### **(Full proof for Theorem A – part 2)**
> > > > >
> > > > > Second, for a given $\mathbf{x} \in \mathcal{X},$ let $\mu\_{k}\^{\diamond}(\mathbf{x}) := argmin\_{\mu\_{k}\^{\diamond} \in \mu\^{\diamond}} \Vert \mathbf{x} - \mu\_{k}\^{\diamond} \Vert\^{2}$ be the cluster center that $\mathbf{x}$ is assigned to, found by $\tau$ approximation algorithm for $K$-means clustering.
> > > > > Note that $\mathcal{A}\_{0}\^{\diamond}$ and $\mathcal{A}\_{1}\^{\diamond}$ are deterministic, hence we can represent $\mu\_{k}\^{\diamond}$ as above.
> > > > > Then, we have that
> > > > > $$ C( \mu\^{\diamond}, \mathcal{A}\_{0}\^{+}, \mathcal{A}\_{1}\^{+} ) = \sum\_{k=1}\^{K} \mathcal{A}\_{s}\^{+}(\mathbf{x})\_{k} \Vert \mathbf{x} - \mu\_{k}\^{\diamond} \Vert\^{2} = \sum\_{k=1}\^{K} \sum\_{k': \tilde{\mu}\_{k'} \in N(\mu\_{k}\^{\diamond}) } \tilde{\mathcal{A}}\_{s}(\mathbf{x})\_{k'} \Vert \mathbf{x} - \mu\_{k}\^{\diamond} \Vert\^{2} $$
> > > > > $$ \le \sum\_{k=1}\^{K} \sum\_{k': \tilde{\mu}\_{k'} \in N(\mu\_{k}\^{\diamond}) } \tilde{\mathcal{A}}\_{s}(\mathbf{x})\_{k'} \left( \Vert \mathbf{x} - \tilde{\mu}\_{k'} \Vert\^{2} + \Vert \tilde{\mu}\_{k'} - \mu\_{k}\^{\diamond} \Vert\^{2} \right) \le \sum\_{k=1}\^{K} \sum\_{k': \tilde{\mu}\_{k'} \in N(\mu\_{k}\^{\diamond}) } \tilde{\mathcal{A}}\_{s}(\mathbf{x})\_{k'} \left( \Vert \mathbf{x} - \tilde{\mu}\_{k'} \Vert\^{2} + \Vert \tilde{\mu}\_{k'} - \mu\_{k}\^{\diamond}(\mathbf{x}) \Vert\^{2} \right) $$
> > > > > $$ \le \sum\_{k=1}\^{K} \sum\_{k': \tilde{\mu}\_{k'} \in N(\mu\_{k}\^{\diamond}) } \tilde{\mathcal{A}}\_{s}(\mathbf{x})\_{k'} \left( 2 \Vert \mathbf{x} - \tilde{\mu}\_{k'} \Vert\^{2} + \Vert \mathbf{x} - \mu\_{k}\^{\diamond}(\mathbf{x}) \Vert\^{2} \right) = 2 \sum\_{k'=1}\^{K} \tilde{\mathcal{A}}\_{s}(\mathbf{x})\_{k'} \Vert \mathbf{x} - \tilde{\mu}\_{k'} \Vert\^{2} + \sum\_{k=1}\^{K} \sum\_{k': \tilde{\mu}\_{k'} \in N(\mu\_{k}\^{\diamond}) } \tilde{\mathcal{A}}\_{s}(\mathbf{x})\_{k'} \Vert \mathbf{x} - \mu\_{k}\^{\diamond}(\mathbf{x}) \Vert\^{2}
> > > > > $$
> > > > > $$ = 2 C(\tilde{\mu}, \tilde{\mathcal{A}}\_{0}, \tilde{\mathcal{A}}\_{1})  + \Vert \mathbf{x} - \mu\_{k}\^{\diamond}(\mathbf{x}) \Vert\^{2} = 2 C(\tilde{\mu}, \tilde{\mathcal{A}}\_{0}, \tilde{\mathcal{A}}\_{1}) + min\_{\mu\_{k}\^{\diamond} \in \mu\^{\diamond}} \Vert \mathbf{x} - \mu\_{k}\^{\diamond} \Vert\^{2} $$
> > > > > $$ \le 2 C(\tilde{\mu}, \tilde{\mathcal{A}}\_{0}, \tilde{\mathcal{A}}\_{1}) + \tau C\^{*} \le  2 C(\tilde{\mu}, \tilde{\mathcal{A}}\_{0}, \tilde{\mathcal{A}}\_{1})+ \tau C(\tilde{\mu}, \tilde{\mathcal{A}}\_{0}, \tilde{\mathcal{A}}\_{1}) = (\tau + 2) C(\tilde{\mu}, \tilde{\mathcal{A}}\_{0}, \tilde{\mathcal{A}}\_{1}).$$
> > > > > Hence, (ii) is proved.
> > > > >
> > > > > Finally, it is clear that $C( \mu\^{\diamond}, \mathcal{A}\_{0}', \mathcal{A}\_{1}' ) \le C( \mu\^{\diamond}, \mathcal{A}\_{0}\^{+}, \mathcal{A}\_{1}\^{+} ),$
> > > > > since $\mathcal{A}\_{0}'$ and $\mathcal{A}\_{1}'$ are the minimizers of of $C(\mu, \mathcal{A}\_{0}, \mathcal{A}\_{1})$ subject to $(\mathcal{A}\_{0}, \mathcal{A}\_{1}) \in \mathbf{A}\_{\epsilon}$ given $\mu = \mu\^{\diamond}$ (by Theorem 4.1).
> > > > > Hence, (iii) is proved.

---

> > > > > > ### Comment · Reviewer_7dQE · 2024-11-28
> > > > > >
> > > > > > > Specifically, while Bera et al., (2019) finds a fair assignment under fixed cluster centers and controls fairness level using  and  we were able to obtain a similar approximation result for FCA-C, which controls fairness level using $\epsilon$ (in 2nd follow-up response #2). After that, we discuss several points comparing between FCA and Bera et al., (2019) (in 2nd follow-up response #3).
> > > > > >
> > > > > > Does your assignment violate the fairness constraint? And I don’t see where $\epsilon$ appears in your followup discussion and the meaning of the this notation.
> > > > > >
> > > > > > > For the fair assignment functions  and obtained by FCA-C, it is guaranteed that cost of fair clustering obtained by FCA-C (for fixed cluster centers)  cost of the optimal fair clustering (shown in Theorem A).
> > > > > >
> > > > > > I’m not sure about this claim. This claim seems to assume the cluster centers are fixed/given, and I suppose the said assignment is found for this given center set. Then how can its cost be comparable to he *optimal* fair clustering? I think it makes more sense to relate to the cost of the given center set.
> > > > > >
> > > > > > Anyway, it’s more conclusive if you can give a more direct approximation bound, such that there’s no assumption about “fixed center set”.

---

> > > > > > > ### Author Response · Authors · 2024-11-29
> > > > > > > **Proof of Theorem A' #1**
> > > > > > >
> > > > > > > ### (Proof of Theorem A’)
> > > > > > >
> > > > > > > Given a $\tau$-approximation algorithm for $K$-means clustering, let $\mu\^{\diamond}, \mathcal{A}\_{0}\^{\diamond}, \mathcal{A}\_{1}\^{\diamond}$ be the approximated solution to $min\_{\mu, \mathcal{A}\_{0}, \mathcal{A}\_{1}} C(\mu, \mathcal{A}\_{0}, \mathcal{A}\_{1}),$ obtained using the given $\tau$-approximation algorithm.
> > > > > > > Let $C\^{*} = min\_{\mu, \mathcal{A}\_{0}, \mathcal{A}\_{1}} C(\mu, \mathcal{A}\_{0}, \mathcal{A}\_{1})$ represent the global optimal clustering cost without fairness constraint.
> > > > > > >
> > > > > > > Then, we have $ C( \mu\^{\diamond}, \mathcal{A}\_{0}\^{\diamond}, \mathcal{A}\_{1}\^{\diamond} ) \le \tau C\^{*}. $
> > > > > > >
> > > > > > > We begin by showing that for given initial cluster centers $\mu\^{\diamond},$
> > > > > > > there exist $\mathcal{A}\_{0}\^{+}$ and $\mathcal{A}\_{1}\^{+}$ that satisfy
> > > > > > > the following conditions:
> > > > > > > (i) $(\mathcal{A}\_{0}\^{+}, \mathcal{A}\_{1}\^{+}) \in \mathbf{A}\_{\epsilon}$,
> > > > > > > (ii) $C( \mu\^{\diamond}, \mathcal{A}\_{0}\^{+}, \mathcal{A}\_{1}\^{+} ) \le (\tau + 2) C( \tilde{\mu}, \tilde{\mathcal{A}}\_{0}, \tilde{\mathcal{A}}\_{1} ),$
> > > > > > > and
> > > > > > > (iii) $C( \mu\^{\diamond}, \mathcal{A}\_{0}', \mathcal{A}\_{1}' ) \le C( \mu\^{\diamond}, \mathcal{A}\_{0}\^{+}, \mathcal{A}\_{1}\^{+} ),$
> > > > > > > where $\mathcal{A}\_{0}', \mathcal{A}\_{1}'$ are the fair assignment functions corresponding to
> > > > > > > $\mathbb{Q}', \mathcal{W}' = argmin\_{\mathbb{Q}, \mathcal{W}} \tilde{C}(\mathbb{Q}, \mathcal{W}, \mu\^{\diamond}).$
> > > > > > >
> > > > > > > (i)
> > > > > > > Recall that $\tilde{\mu} = \\{ \tilde{\mu}\_{k} \\}\_{k=1}\^{K} $ and $\mu\^{\diamond} = \\{ \mu\_{k}\^{\diamond} \\}\_{k=1}\^{K}.$
> > > > > > > For all $k \in [K], $ we define the set of nearest neighbors of $\mu\_{k}\^{\diamond}$ as $N(\mu\_{k}\^{\diamond}) := \\{ \tilde{\mu}\_{k} \in \tilde{\mu} : argmin\_{\mu\_{k'}\^{\diamond} \in \mu\^{\diamond}} \Vert \tilde{\mu}\_{k} - \mu\_{k'}\^{\diamond} \Vert^{2} = \mu\_{k}\^{\diamond} \\}.$
> > > > > > > For $\mathbf{x} \in \mathcal{X},$ we define $\mathcal{A}\_{s}\^{+}(\mathbf{x})\_{k} := \sum\_{k': \tilde{\mu}\_{k'} \in N(\mu\_{k}\^{\diamond}) } \tilde{\mathcal{A}}\_{s}(\mathbf{x})\_{k'}.$
> > > > > > > Then, $\mathcal{A}\_{s}\^{+}$ also becomes an assignment function since $\sum\_{k=1}\^{K} \mathcal{A}\_{s}\^{+}(\mathbf{x}) = 1$ for all $\mathbf{x} \in \mathcal{X}.$
> > > > > > >
> > > > > > > From the definitions of $\tilde{\mathcal{A}}\_{0}$ and $\tilde{\mathcal{A}}\_{1},$ we have the fact that $ \sum\_{k=1}\^{K} \left\vert \frac{1}{n\_{0}} \sum\_{i=1}\^{n\_{0}} \tilde{\mathcal{A}}\_{0}(\mathbf{x}\_{i}\^{(0)})\_{k} - \frac{1}{n\_{1}} \sum\_{j=1}\^{n\_{1}} \tilde{\mathcal{A}}\_{1} (\mathbf{x}\_{j}\^{(1)})\_{k} \right\vert \le \epsilon. $
> > > > > > > Therefore, we obtain
> > > > > > > $$
> > > > > > > \sum\_{k=1}\^{K} \left\vert \frac{1}{n\_{0}}  \sum\_{i=1}\^{n\_{0}} \mathcal{A}\_{0}\^{+}(\mathbf{x}\_{i}\^{(0)})\_{k} - \frac{1}{n\_{1}} \sum\_{j=1}\^{n\_{1}} \mathcal{A}\_{1}\^{+} (\mathbf{x}\_{j}\^{(1)})\_{k} \right\vert
> > > > > > > = \sum\_{k=1}\^{K} \left\vert \sum\_{k': \tilde{\mu}\_{k'} \in N(\mu\_{k}\^{\diamond})} \left( \frac{1}{n\_{0}} \sum\_{i=1}\^{n_{0}} \tilde{\mathcal{A}}\_{0}(\mathbf{x}\_{i}\^{(0)})\_{k'} - \frac{1}{n_{1}} \sum\_{j=1}\^{n\_{1}} \tilde{\mathcal{A}}\_{1} (\mathbf{x}\_{j}\^{(1)})\_{k'}  \right) \right\vert
> > > > > > > $$
> > > > > > > $$
> > > > > > > \le
> > > > > > > \sum\_{k=1}\^{K} \sum\_{k': \tilde{\mu}\_{k'} \in N(\mu\_{k}\^{\diamond})} \left\vert \frac{1}{n\_{0}} \sum\_{i=1}\^{n_{0}} \tilde{\mathcal{A}}\_{0}(\mathbf{x}\_{i}\^{(0)})\_{k'} - \frac{1}{n_{1}} \sum\_{j=1}\^{n\_{1}} \tilde{\mathcal{A}}\_{1} (\mathbf{x}\_{j}\^{(1)})\_{k'} \right\vert
> > > > > > > =\sum\_{k'=1}\^{K} \left\vert \frac{1}{n_{0}} \sum\_{i=1}\^{n_{0}} \tilde{\mathcal{A}}\_{0}(\mathbf{x}\_{i}\^{(0)})\_{k'} - \frac{1}{n\_{1}} \sum\_{j=1}\^{n\_{1}} \tilde{\mathcal{A}}\_{1} (\mathbf{x}\_{j}\^{(1)})\_{k'} \right\vert \le \epsilon,
> > > > > > > $$
> > > > > > > where the last equality holds because $\cup\_{k=1}\^{K} \cup\_{k': \tilde{\mu}\_{k'} \in N(\mu\_{k}\^{\diamond})} \\{ k' \\} = \\{ k' \\}\_{k'=1}\^{K}.$
> > > > > > > Thus, the condition (i) is satisfied.

---

> > > > > > > ### Author Response · Authors · 2024-11-29
> > > > > > > **Proof of Theorem A' #2**
> > > > > > >
> > > > > > > (ii)
> > > > > > > For a given $\mathbf{x} \in \mathcal{X},$ let $\mu\_{k}\^{\diamond}(\mathbf{x}) := argmin\_{\mu\_{k}\^{\diamond} \in \mu\^{\diamond}} \Vert \mathbf{x} - \mu\_{k}\^{\diamond} \Vert^{2}$ be the initial cluster center closest to $\mathbf{x}.$
> > > > > > > We then have:
> > > > > > > $$
> > > > > > > \sum\_{k=1}\^{K} \mathcal{A}\_{s}\^{+}(\mathbf{x})\_{k} \Vert \mathbf{x} - \mu\_{k}\^{\diamond} \Vert\^{2}
> > > > > > > = \sum\_{k=1}\^{K} \sum\_{k': \tilde{\mu}\_{k'} \in N(\mu\_{k}\^{\diamond}) } \tilde{\mathcal{A}}\_{s}(\mathbf{x})\_{k'}
> > > > > > > \Vert \mathbf{x} - \mu\_{k}\^{\diamond} \Vert\^{2}
> > > > > > > \le \sum\_{k=1}\^{K} \sum\_{k': \tilde{\mu}\_{k'} \in N(\mu\_{k}\^{\diamond}) } \tilde{\mathcal{A}}\_{s}(\mathbf{x})\_{k'}
> > > > > > > \left( \Vert \mathbf{x} - \tilde{\mu}\_{k'} \Vert\^{2} + \Vert \tilde{\mu}\_{k'} - \mu\_{k}\^{\diamond} \Vert\^{2} \right)
> > > > > > > $$
> > > > > > > $$
> > > > > > > \le \sum\_{k=1}\^{K} \sum\_{k': \tilde{\mu}\_{k'} \in N(\mu\_{k}\^{\diamond}) } \tilde{\mathcal{A}}\_{s}(\mathbf{x})\_{k'}
> > > > > > > \left( \Vert \mathbf{x} - \tilde{\mu}\_{k'} \Vert\^{2} + \Vert \tilde{\mu}\_{k'} - \mu\_{k}\^{\diamond}(\mathbf{x}) \Vert\^{2} \right)
> > > > > > > \le \sum\_{k=1}\^{K} \sum\_{k': \tilde{\mu}\_{k'} \in N(\mu\_{k}\^{\diamond}) } \tilde{\mathcal{A}}\_{s}(\mathbf{x})\_{k'}
> > > > > > > \left( 2 \Vert \mathbf{x} - \tilde{\mu}\_{k'} \Vert^{2} + \Vert \mathbf{x} - \mu\_{k}\^{\diamond}(\mathbf{x}) \Vert\^{2} \right)
> > > > > > > $$
> > > > > > > $$
> > > > > > > = 2 \sum\_{k'=1}\^{K} \tilde{\mathcal{A}}\_{s}(\mathbf{x})\_{k'} \Vert \mathbf{x} - \tilde{\mu}\_{k'} \Vert\^{2} + \sum\_{k=1}\^{K} \sum\_{k': \tilde{\mu}\_{k'} \in N(\mu\_{k}\^{\diamond}) } \tilde{\mathcal{A}}\_{s}(\mathbf{x})\_{k'}
> > > > > > > \Vert \mathbf{x} - \mu\_{k}\^{\diamond}(\mathbf{x}) \Vert\^{2}
> > > > > > > = 2 \sum\_{k'=1}\^{K} \tilde{\mathcal{A}}\_{s}(\mathbf{x})\_{k'}
> > > > > > > \Vert \mathbf{x} - \tilde{\mu}\_{k'} \Vert\^{2} + \Vert \mathbf{x} - \mu\_{k}\^{\diamond}(\mathbf{x}) \Vert\^{2}.
> > > > > > > $$
> > > > > > > Summing over all $\mathbf{x}$ and dividing by $n,$ we obtain:
> > > > > > > $$
> > > > > > > C( \mu\^{\diamond}, \mathcal{A}\_{0}\^{+}, \mathcal{A}\_{1}\^{+} ) \le 2 C(\tilde{\mu}, \tilde{\mathcal{A}}\_{0}, \tilde{\mathcal{A}}\_{1}) + \frac{1}{n} \sum\_{\mathbf{x} \in \mathcal{X}} min\_{k} \Vert \mathbf{x} - \mu\_{k}^{\diamond} \Vert\^{2}
> > > > > > > = 2 C(\tilde{\mu}, \tilde{\mathcal{A}}\_{0}, \tilde{\mathcal{A}}\_{1})  + C(\mu\^{\diamond}, \mathcal{A}\_{0}\^{\diamond}, \mathcal{A}\_{1}\^{\diamond})
> > > > > > > $$
> > > > > > > $$
> > > > > > > \le
> > > > > > > 2 C(\tilde{\mu}, \tilde{\mathcal{A}}\_{0}, \tilde{\mathcal{A}}\_{1}) + \tau C\^{*}
> > > > > > > \le
> > > > > > > 2 C(\tilde{\mu}, \tilde{\mathcal{A}}\_{0}, \tilde{\mathcal{A}}\_{1}) + \tau C(\tilde{\mu}, \tilde{\mathcal{A}}\_{0}, \tilde{\mathcal{A}}\_{1})
> > > > > > > = (\tau + 2) C(\tilde{\mu}, \tilde{\mathcal{A}}\_{0}, \tilde{\mathcal{A}}\_{1}),
> > > > > > > $$
> > > > > > > which concludes (ii).
> > > > > > >
> > > > > > > (iii)
> > > > > > > It is clear that $C( \mu\^{\diamond}, \mathcal{A}\_{0}', \mathcal{A}\_{1}' ) \le C( \mu\^{\diamond}, \mathcal{A}\_{0}\^{+}, \mathcal{A}\_{1}\^{+} ),$
> > > > > > > since $\mathcal{A}\_{0}'$ and $\mathcal{A}\_{1}'$ are the minimizers of of $C(\mu, \mathcal{A}\_{0}, \mathcal{A}\_{1})$ subject to $(\mathcal{A}\_{0}, \mathcal{A}\_{1}) \in \mathbf{A}\_{\epsilon}$ given $\mu = \mu\^{\diamond}$ (by Theorem 4.1).
> > > > > > > Thus, the condition (iii) is satisfied.
> > > > > > >
> > > > > > > Let $\mu\_{k}' := \sum\_{i=1}\^{n} \mathcal{A}\_{s}'(\mathbf{x}\_{i})\_{k} \mathbf{x}\_{i} / \sum\_{i=1}\^{n} \mathcal{A}\_{s}'(\mathbf{x}\_{i})\_{k}, $ which is the minimizer of $min\_{\mu} C(\mu, \mathcal{A}\_{0}', \mathcal{A}\_{1}')$ given $\mathcal{A}\_{0}'$ and $\mathcal{A}\_{1}'.$
> > > > > > > Then, it is clear that $C(\mu', \mathcal{A}\_{0}', \mathcal{A}\_{1}') \le C(\mu\^{\diamond}, \mathcal{A}\_{0}', \mathcal{A}\_{1}') \le (\tau + 2) C(\tilde{\mu}, \tilde{\mathcal{A}}\_{0}, \tilde{\mathcal{A}}\_{1}).$
> > > > > > >
> > > > > > > Finally, we iterate the above process.
> > > > > > > Using FCA-C, we find $\mathcal{A}\_{0}''$ and $\mathcal{A}\_{1}''$, which minimizes the cost given $\mu'.$
> > > > > > > Then, let $\mu\_{k}'' := \sum\_{i=1}\^{n} \mathcal{A}\_{s}''(\mathbf{x}\_{i})\_{k} \mathbf{x}\_{i} / \sum\_{i=1}\^{n} \mathcal{A}\_{s}''(\mathbf{x}\_{i})\_{k},$ which is the minimizer of $min\_{\mu} C(\mu, \mathcal{A}\_{0}'', \mathcal{A}\_{1}'')$ given $\mathcal{A}\_{0}''$ and $\mathcal{A}\_{1}''.$
> > > > > > > This results in the following inequality:
> > > > > > > $C(\mu'', \mathcal{A}\_{0}'', \mathcal{A}\_{1}'') \le C(\mu', \mathcal{A}\_{0}'', \mathcal{A}\_{1}'') \le C(\mu', \mathcal{A}\_{0}', \mathcal{A}\_{1}') \le (\tau + 2) C(\tilde{\mu}, \tilde{\mathcal{A}}\_{0}, \tilde{\mathcal{A}}\_{1}).$
> > > > > > > Hence, iterating this process until convergence guarantees the approximation error of $\tau + 2.$

---

> > > > > > > > ### Comment · Reviewer_7dQE · 2024-11-29
> > > > > > > >
> > > > > > > > Thanks for the detailed response. I’m still trying to understand your argument at a high level.
> > > > > > > >
> > > > > > > > 1. You seem to say you can find a solution (A_0, A_1) \in A_\epsilon whose cost <= (\tau + 2) times the optimal unconstrained k-means. However, is A_\epsilon the exact feasible region of fair level epsilon? If not, then you do have some violate to the fairness constraint. I ask this because I see from Corollary 4.2, it seems here we have a gap of C (which is not even an universal constant and is defined w.r.t. the dataset).
> > > > > > > >
> > > > > > > > 2. I’m somewhat convinced that if you can solve your formulation of FCA-C for a given approximate center set, then you can have the said approximation. This seems to be what you focused on in the previous responses.
> > > > > > > >
> > > > > > > > I might have missed something, but FCA-C is very complicated, and you only provided a “practical algorithm” in Section 4.2 — it is not clear if that algorithm solves FCA-C optimally (and in fact, perhaps we shouldn’t use that algorithm at all, since it also optimizes the center set which is already given). Do we know an efficient algorithm that solves FCA-C optimally, or at least achieves your claimed approximation guarantee?

---

> ### Author Response · Authors · 2024-11-26
> **2nd follow-up response #1**
>
> *Dear reviewer 7dQE*:
>
> We sincerely appreciate your valuable feedback and this important discussion!
> Thanks to your comment, we were able to derive an **approximation guarantee of our proposed algorithm**.
> We believe these results enhance the theoretical validity of FCA, particularly regarding its worst-case guarantee, and we hope this addresses your remaining concern.
> The following discussion has been also added to Section B.5 of Appendix.
>
> Specifically, while Bera et al., (2019) finds a fair assignment under fixed cluster centers and controls fairness level using $\alpha$ and $\beta,$ we were able to obtain a similar approximation result for FCA-C, which controls fairness level using $\epsilon$ (in 2nd follow-up response #2).
> After that, we discuss several points comparing between FCA and Bera et al., (2019) (in 2nd follow-up response #3).
>
> Please see our follow-up responses below.

---

> ### Author Response · Authors · 2024-11-26
> **2nd follow-up response #2**
>
> #### **(Approximation guarantee)**
>
>    We establish the approximation guarantee of our proposed algorithm:
>    **The cost of our fair clustering solution is at most $\tau + 2$ times of the cost of the global optimal fair clustering solution** (where $\tau$ is the approximation error of the algorithm used for $K$-means clustering).
>    In other words, FCA-C has an approximation error of $\tau + 2$ for any fairness level.
>
>    - **Theorem A** below shows that finding fair clustering (assignments) using FCA-C with fixed cluster centers (found by an approximation algorithm for $K$-means clustering) has a guaranteed approximation error.
>
>    - **Proposition B** below subsequently presents a straightforward result that updating cluster centers also guarantees at least the same approximation error.
>
>
> -------
>
>    The formal mathematical statements are as follows.
>    - **(Definitions)**
>        Recall the $K$-means clustering cost
>        $C(\mu, \mathcal{A}\_{0}, \mathcal{A}\_{1}) := \mathbb{E} \sum\_{k=1}\^{K} \mathcal{A}\_{S}(\mathbf{X})\_{k} \Vert \mathbf{X} - \mu\_{k} \Vert\^{2},$
>        where $\mu = \\{ \mu\_{k} \\}\_{k=1}\^{K}.$
>        Note that this is equivalently written as $\sum\_{i=1}\^{n} \sum\_{k=1}\^{K} \mathcal{A}\_{s\_{i}}(\mathbf{x}\_{i})\_{k} \Vert \mathbf{x}\_{i} - \mu\_{k} \Vert\^{2}.$
>
>         For a given $\epsilon,$ let $\tilde{\mu}, \tilde{\mathcal{A}}\_{0}, \tilde{\mathcal{A}}\_{1}$ be the optimal solution of $min\_{\mu, \mathcal{A}\_{0}, \mathcal{A}\_{1}} C(\mu, \mathcal{A}\_{0}, \mathcal{A}\_{1})$ subject to $(\mathcal{A}\_{0}, \mathcal{A}\_{1}) \in \mathbf{A}\_{\epsilon},$ i.e., optimal fair clustering solution for fairness level $\epsilon.$
>
>         Let $C^{*} = min_{\mu, \mathcal{A}\_{0}, \mathcal{A}\_{1}} C(\mu, \mathcal{A}\_{0}, \mathcal{A}\_{1})$ be the global optimal clustering cost without fairness constraint.
>
>    - **(Theorem A)**
>        Given a $\tau$-approximation algorithm for (standard) $K$-means clustering, let $\mu\^{\diamond}, \mathcal{A}\_{0}\^{\diamond}, \mathcal{A}\_{1}\^{\diamond}$ be the (approximated) solution of $min\_{\mu, \mathcal{A}\_{0}, \mathcal{A}\_{1}} C(\mu, \mathcal{A}\_{0}, \mathcal{A}\_{1}),$ found by the $\tau$-approximation algorithm.
>        For fixed $\mu\^{\diamond},$
>        we have that
>        $C( \mu\^{\diamond}, \mathcal{A}\_{0}', \mathcal{A}\_{1}' ) \le (\tau + 2) C(\tilde{\mu}, \tilde{\mathcal{A}}\_{0}, \tilde{\mathcal{A}}\_{1}),$
>        where $\mathcal{A}\_{0}', \mathcal{A}\_{1}'$ are the assignment functions corresponding to
>        $\mathbb{Q}', \mathcal{W}' = argmin\_{\mathbb{Q}, \mathcal{W}} \tilde{C}(\mathbb{Q}, \mathcal{W}, \mu\^{\diamond}),$
>        i.e., the solution of FCA-C under fixed $\mu^{\diamond}.$
>
>       - (Sketch of proof)
>                     We show that there exist $\mathcal{A}\_{0}\^{+}$ and $\mathcal{A}\_{1}\^{+}$ such that
>                     (i) $(\mathcal{A}\_{0}\^{+}, \mathcal{A}\_{1}\^{+}) \in \mathbf{A}\_{\epsilon}$,
>                     (ii) $C( \mu\^{\diamond}, \mathcal{A}\_{0}\^{+}, \mathcal{A}\_{1}\^{+} ) \le (\tau + 2) C(\tilde{\mu}, \tilde{\mathcal{A}}\_{0}, \tilde{\mathcal{A}}\_{1}),$
>                     and
>                     (iii) $C( \mu\^{\diamond}, \mathcal{A}\_{0}', \mathcal{A}\_{1}' ) \le C( \mu\^{\diamond}, \mathcal{A}\_{0}\^{+}, \mathcal{A}\_{1}\^{+} ).$
>                     Full proof is provided in the following response (= Proof of Theorem A \#1 and \#2) as well as Section B.5 of Appendix.
>
>    - **(Proposition B)**
>       For given $\mathcal{A}\_{0}', \mathcal{A}\_{1}',$ there exists $\mu'$ such that $C(\mu', \mathcal{A}\_{0}', \mathcal{A}\_{1}') \le C(\mu^{\diamond}, \mathcal{A}\_{0}', \mathcal{A}\_{1}').$
>
>       - (Sketch of proof)
>        Letting $\mu\_{k}' := \sum\_{i=1}\^{n} \mathcal{A}\_{s}(\mathbf{x}\_{i})\_{k} \mathbf{x}\_{i} / \sum\_{i=1}\^{n} \mathcal{A}\_{s}(\mathbf{x}\_{i})\_{k}, $ which solves $min\_{\mu} C(\mu, \mathcal{A}\_{0}', \mathcal{A}\_{1}'),$ concludes the proof.
>
> -------
>
>    - **(Implication)**
>
>       **For the fair assignment functions $\mathcal{A}\_{0}'$ and $\mathcal{A}\_{1}'$ obtained by FCA-C, it is guaranteed that
>       *cost of fair clustering obtained by FCA-C (for fixed cluster centers)*
>       $\le (\tau + 2) \times$
>       *cost of the optimal fair clustering* (shown in Theorem A).**
>
>       Next, once the cluster centers are updated by minimizing the clustering cost given the assignment functions, it follows that:
>       *cost of fair clustering obtained by FCA-C (after one cluster center update)*
>       $\le$
>        *cost of fair clustering obtained by FCA-C (for fixed cluster centers)* (shown in Proposition B).
>        This implies that the fair clustering obtained by FCA-C with updated cluster centers also has the $(\tau + 2)$-approximation guarantee.

---

> ### Author Response · Authors · 2024-11-26
> **2nd follow-up response #3**
>
> #### **(Comparison between our approach and Bera et al., (2019))**
>
> The fair assignment functions in our framework, obtained by the matching (transport) map, can be viewed as an alternative and efficient approach to the fair assignment algorithm of Bera et al., (2019).
> Specifically, we translate the linear programming (LP) in Bera et al., (2019) for directly finding the fair assignments into finding a matching map between different protected groups.
>
> In our framework, any matching map between different protected groups guarantees fairness of the resulting clustering, as discussed in Theorem 3.1.
> This is because fairness is implicitly satisfied through the use of matching.
> FCA focuses on finding the optimal matching (as well as cluster centers) that minimizes the clustering cost while ensuring fairness.
> On the other hand, Bera et al., (2019) requires solving a constrained LP to directly obtain a fair assignment.
>
> Several advantages of our framework (using the matching map for fair clustering) are as follows.
> 1. **(Descent property of cost)**
>         FCA alternately updates cluster centers and fair assignments (via alignment) until convergence.
>         This iterative process enables the final solution to numerically converge to a local optimum, achieving a better trade-off between cost and fairness.
>         In contrast, Bera et al., (2019) finds fair assignments under fixed cluster centers.
>
> 2. **(Numerical performance)**
>         As mentioned in lines 1333-1335 in Section C.2.2 of Appendix and supported by the VFC paper (Ziko et al., 2021), VFC outperforms Bera et al., (2019) in terms of cost-fairness trade-off on the datasets used in our experiments.
>         Consequently, FCA, which shows improved trade-offs compared to VFC, is expected to outperform Bera et al. (2019).
>
> 3. **(Possible fast computation)**
>         By leveraging regularization algorithms such as the Sinkhorn algorithm, we can reduce the computational complexity to $\mathcal{O}(n^{2})$ when finding the matching in FCA.
>         Furthermore, using the partitioning technique introduced in Section 4.3, the complexity can be further reduced to $\mathcal{O}(n m),$ where $m$ is the partition size.
>         Importantly, fairness on the entire dataset is still guaranteed in FCA when using this technique.
>
>    In contrast, the computational complexity of Bera et al., (2019) could be high, since the computational complexity of LP is approximately
>    $\mathcal{O}(n^{3}),$ in general.
>         As noted in their paper, `solver fails to find a solution in under an hour even with 40K points'.
>         Additionally, even with our partitioning technique for the LP in Bera et al. (2019), the computational complexity would be reduced to
>    $\mathcal{O}(n m^{2}),$ but we cannot guarantee fairness for the entire dataset.
>         It is because partition-wise LP may not ensure fairness on the entire dataset.

---

> ### Author Response · Authors · 2024-11-29
> **3rd follow-up response**
>
> *Dear reviewer 7dQE*:
> Thank you very much for the additional feedback.
> In this response, we provide a clearer result regarding the approximation error.
>
> ------
> > 1. Does your assignment violate the fairness constraint? And I don’t see where $\epsilon$ appears in your followup discussion and the meaning of the this notation.
>
> - *Answer*: First, $\epsilon$ represents the fairness level of a given fairness constraint (defined in line 337).
> The smaller the value of $\epsilon,$ the fairer the clustering.
> Additionally, our assignment function satisfies the fairness constraint, as its corresponding fairness level is below $\epsilon$, according to the definition in eq. (6).
>
> ------
> > 2. I’m not sure about this claim. This claim seems to assume the cluster centers are fixed/given, and I suppose the said assignment is found for this given center set. Then how can its cost be comparable to he optimal fair clustering? I think it makes more sense to relate to the cost of the given center set. Anyway, it’s more conclusive if you can give a more direct approximation bound, such that there’s no assumption about “fixed center set”.
>
> - *Answer*:
>
>    In fact, the fixed cluster centers we described in the previous response can be understood as the *initial cluster centers*, obtained using an approximation algorithm for $K$-means clustering.
>    Please note that in Bera et al., (2019), they also provide an approximation bound result for their fair clustering approach (via assignment) under the fixed cluster centers.
>
>    However, we agree that presenting a result without the term "fixed cluster centers" would be more conclusive.
>    In response, we present the following Theorem A', which guarantees that
>    **the cost of fair clustering obtained by FCA-C is at most $(\tau + 2)$ times the cost of the optimal fair clustering**.
>    This result is based on the previous results and the mechanism of FCA-C, which iteratively updates fair assignments and cluster centers to minimize the cost.
>
>    - Definitions:
>                 For a given fairness level $\epsilon$,
>                 let $\tilde{\mu}, \tilde{\mathcal{A}}\_{0}, \tilde{\mathcal{A}}\_{1}$ be the optimal solution of $\min_{\mu, \mathcal{A}\_{0}, \mathcal{A}_{1}} C(\mu, \mathcal{A}\_{0}, \mathcal{A}\_{1})$ subject to $(\mathcal{A}\_{0}, \mathcal{A}\_{1}) \in \mathbf{A}\_{\epsilon},$
>                 i.e., optimal fair clustering solution for fairness level $\epsilon.$
>
>                 Let $\mu\^{\triangle}, \mathcal{A}\_{0}\^{\triangle}, \mathcal{A}\_{1}\^{\triangle}$ be the solution of FCA-C given a fairness level $\epsilon,$ given initial cluster centers obtained by a $\tau$-approximation algorithm for $K$-means clustering.
>
>    - **Theorem A'**:
>       Given initial cluster centers obtained by a $\tau$-approximation algorithm for $K$-means clustering, FCA-C returns a $(\tau + 2)$-approximate solution for fair clustering, i.e., $C( \mu\^{\triangle}, \mathcal{A}\_{0}\^{\triangle}, \mathcal{A}\_{1}\^{\triangle} ) \le (\tau + 2) C(\tilde{\mu}, \tilde{\mathcal{A}}\_{0}, \tilde{\mathcal{A}}\_{1}).$
>
>    Note that if we use Lloyd's algorithm with the $K$-means++ initialization, we have $\tau = \mathcal{O}(\log K).$
>
>    As a result, we will make slight revisions to Section B.5 of Appendix in the final version, by updating this result and its proof accordingly.
>    The proof is provided in the responses below.

---

> ### Author Response · Authors · 2024-12-01
> **4th follow-up response**
>
> *Dear reviewer 7dQE*:
> We truly appreciate your effort to understand our claims and for continuing this discussion.
> We hope this response improves our claims and your understanding.
>
> -------
>
> > 1. You seem to say you can find a solution $(A_0, A_1) \in A_\epsilon$ whose cost $\le (\tau + 2)$ times the optimal unconstrained k-means. However, is $A_\epsilon$ the exact feasible region of fair level epsilon? ...
>
> - *Answer*:
>    - First, we would like to clarify that the theoretical result we proved in the previous response is about the bound **between the cost of a solution and the cost of optimal fair clustering (not the unconstrained $K$-means clustering)**.
>    - Second, **the fairness constrained set in FCA-C (i.e., $\mathbf{A}_{\epsilon}$) is based on the *difference* between the assignment probabilities of $S=0$ and $S=1,$** but not based on the ratio (e.g., Balance). Hence, by definition, the feasible set of fair assignment functions in FCA-C is $\mathbf{A}_{\epsilon}.$
>    - **We use the difference as the fairness constraint because we can find fair clusterings effectively by solving the Kantorovich problem, when the fairness constraint is based on the difference rather than the ratio.**
>    - We do not control balance using FCA-C, instead, FCA-C controls the difference in assignment probabilities. However, the message of Corollary 4.2 is that, after obtaining a clustering, we can at least guarantee that the balance of the clustering is bounded by $\epsilon,$ provided that the probability of assigning to each cluster is not too small.
>
> ------
>
> > 2. I'm somewhat convinced ... Do we know an efficient algorithm that solves FCA-C optimally, or at least achieves your claimed approximation guarantee?
>
> - *Answer*:
>
>    **1. Approximation error of our algorithm for FCA-C**
>    - Definitions: For a given fairness level $\epsilon,$
>    let $\tilde{\mu}, \tilde{\mathcal{A}}\_{0}, \tilde{\mathcal{A}}\_{1}$ be the optimal solution of $min\_{\mu, \mathcal{A}\_{0}, \mathcal{A}_{1}} C(\mu, \mathcal{A}\_{0}, \mathcal{A}\_{1})$ subject to $(\mathcal{A}\_{0}, \mathcal{A}\_{1}) \in \mathbf{A}\_{\epsilon},$
>    i.e., optimal fair clustering solution for fairness level $\epsilon.$
>    Let $\mu\^{\triangle}, \mathcal{A}\_{0}\^{\triangle}, \mathcal{A}\_{1}\^{\triangle}$ be the solution of FCA-C algorithm for fairness level $\epsilon,$ given initial cluster centers obtained by a $\tau$-approximation algorithm for $K$-means clustering.
>    - **Theorem A''**: Suppose that $sup\_{\mathbf{x} \in \mathcal{X}} \Vert \mathbf{x} \Vert\^{2} \le R$ for some $R > 0.$ Given initial cluster centers obtained by a $\tau$-approximation algorithm for $K$-means clustering, **FCA-C algorithm returns a $(\tau + 2)$-approximate solution with violation $3 R \epsilon$ for optimal fair clustering**, i.e., $C( \mu\^{\triangle}, \mathcal{A}\_{0}\^{\triangle}, \mathcal{A}\_{1}\^{\triangle} ) \le (\tau + 2) C(\tilde{\mu}, \tilde{\mathcal{A}}\_{0}, \tilde{\mathcal{A}}\_{1}) + 3 R \epsilon.$
>    - **Remark - comparison with our previous result**:
>     In Theorem A' from the previous response, we proved that if we can find the optimal solution for our objective given initial cluster centers, then the solution has an approximation error of $(\tau + 2).$
>    **In Theorem A'', thanks to your discussion, we could prove that our FCA-C algorithm has the same approximation error of $(\tau + 2)$ with an additional violation of $\mathcal{O}(\epsilon).$
>    The difference between the two is the term $\mathcal{O}(\epsilon).$**
>    We will make slight revisions to Section B.5 of Appendix in the final version, by accordingly adding this improved discussion and proof (provided in the response below).
>
>    **2. Comparison to Bera et al., (2019)**
>
>    Additionally, to support readers' high-level understanding, we provide a summary comparing Bera et al. (2019) with our approach.
>    We will also include this table in Section B.5.
>
> |                           | Bera et al., (2019)                                                                             | FCA-C                                                  |
> |---------------------------|-------------------------------------------------------------------------------------------------|--------------------------------------------------------|
> | Initial cluster centers   | Found by an $\tau$-approximation algorithm for standard $K$-means clustering| Found by an $\tau$-approximation algorithm for standard $K$-means clustering |
> | Approximation error bound | $(\tau + 2)$ with constant violation of $3$                                                     | $(\tau + 2)$ with violation of $\mathcal{O}(\epsilon)$ |
> | Algorithm                           | Solving LP  | Solving the Kantorovich problem [^1] |
>
> [^1]: Computationally efficient, so we can iterate many times.

---

> ### Author Response · Authors · 2024-12-01
> **Proof of Theorem A'' (part 1)**
>
> Given a $\tau$-approximation algorithm for (standard) $K$-means clustering,
> let $\mu\^{\diamond}, \mathcal{A}\_{0}\^{\diamond}, \mathcal{A}\_{1}\^{\diamond}$ be the (approximated) solution.
> We also recall that
> $\mathbb{Q}', \mathcal{W}' := argmin_{\mathbb{Q}, \mathcal{W}} \tilde{C}(\mathbb{Q}, \mathcal{W}, \mu\^{\diamond}),$
> and $\mathcal{A}\_{0}', \mathcal{A}\_{1}'$ are the assignment functions corresponding to $\mathbb{Q}'$ and $\mathcal{W}'.$
>
> It suffices to show (*Claim A*): there exist $\mathcal{Q}\^{init}, \mathcal{W}\^{init}$ such that
> $C(\mu\^{\diamond}, \mathcal{A}\_{0}\^{init}, \mathcal{A}\_{1}\^{init}) \le C(\mu\^{\diamond}, \mathcal{A}\_{0}', \mathcal{A}\_{1}') + 3 R \epsilon,$
> where $\mathcal{A}\_{0}\^{init}$ and $\mathcal{A}\_{1}\^{init}$ are the fair assignment functions corresponding to $\mathcal{Q}\^{init}$ and $\mathcal{W}\^{init}.$
>
> Then, by combining with the proof of the theorem in the previous response (approximation error under given cluster centers), we can get the same approximation error for $(\mu^{\triangle}, \mathcal{A}\_{0}\^{\triangle}, \mathcal{A}\_{1}\^{\triangle}),$ which completes the proof.
>
> *Proof of Claim A*:
> Note that we can rewrite
> $$ \tilde{C}(\mathbb{Q}, \mathcal{W}, \mu) =
> \mathbb{E}\_{\mathbf{X}\_{0}, \mathbf{X}\_{1} \sim \mathbb{Q}}
> \big( \textup{FCA cost} (\mathbf{X}\_{0}, \mathbf{X}\_{1}, \mu) \big) $$
> $$ - \mathbb{E}\_{\mathbf{X}\_{0}, \mathbf{X}\_{1} \sim \mathbb{Q}}
> \big( \textup{FCA cost} (\mathbf{X}\_{0}, \mathbf{X}\_{1}, \mu) - \textup{K-means cost} (\mathbf{X}\_{0}, \mathbf{X}\_{1}, \mu) \big)
> \mathbb{1}\left((\mathbf{X}\_{0}, \mathbf{X}\_{1}) \in \mathcal{W} \right), $$
> where $\textup{FCA cost} (\mathbf{X}\_{0}, \mathbf{X}\_{1}, \mu) = \big( 2 \pi_{0} \pi_{1} \Vert \mathbf{X}\_{0} - \mathbf{X}\_{1} \Vert^{2} + min_{k} \Vert \mathbf{T}\^{\textup{A}}(\mathbf{X}\_{0}, \mathbf{X}\_{1}) - \mu_{k} \Vert^{2} \big)$
> and $\textup{K-means cost} (\mathbf{X}\_{0}, \mathbf{X}\_{1}, \mu) = min_{k} \left( \pi_{0} \Vert \mathbf{X}\_{0} - \mu_{k} \Vert^{2} \right) + min_{k} \left( \pi_{1} \Vert \mathbf{X}\_{1} - \mu_{k} \Vert^{2} \right),$
> which are defined in eq. (5).
>
> (i): For given $\mu\^{\diamond},$
> define $\mathbb{Q}\^{init}$ as the solution of
> $ min_{\mathbb{Q} \in \mathcal{Q}} \mathbb{E}\_{\mathbf{X}\_{0}, \mathbf{X}\_{1} \sim \mathbb{Q}}
> (\textup{FCA cost} (\mathbf{X}\_{0}, \mathbf{X}\_{1}, \mu\^{\diamond})), $
> which can be found by solving the Kantorovich problem.
> Then, we have $\mathbb{E}\_{\mathbf{X}\_{0}, \mathbf{X}\_{1} \sim \mathbb{Q}\^{init}}
> \big( \textup{FCA cost} (\mathbf{X}\_{0}, \mathbf{X}\_{1}, \mu\^{\diamond}) \big) \le \mathbb{E}\_{\mathbf{X}\_{0}, \mathbf{X}\_{1} \sim \mathbb{Q}'}
> \big( \textup{FCA cost} (\mathbf{X}\_{0}, \mathbf{X}\_{1}, \mu\^{\diamond}) \big).$
>
> (ii): Let $\eta(\mathbf{x}\_{0}, \mathbf{x}\_{1}) := \textup{FCA cost}(\mathbf{x}\_{0}, \mathbf{x}\_{1}, \mu\^{\diamond}) - \textup{K-means cost}(\mathbf{x}\_{0}, \mathbf{x}\_{1}, \mu\^{\diamond}) $
> and $\eta_\epsilon$ be the $\epsilon$th upper quantile.
> Define $\mathcal{W}\^{init}=\{ (\mathbf{x}\_{0}, \mathbf{x}\_{1}) \in \mathcal{X}\_{0} \times \mathcal{X}\_{1}: \eta(\mathbf{x}\_{0}, \mathbf{x}\_{1})> \eta_\epsilon\}.$
> Then, using (i), we have
> $ \tilde{C}(\mathbb{Q}\^{init}, \mathcal{W}\^{init}, \mu\^{\diamond}) \le  \tilde{C}(\mathbb{Q}\^{init}, \emptyset, \mu\^{\diamond}) =
> \mathbb{E}\_{\mathbf{X}\_{0}, \mathbf{X}\_{1} \sim \mathbb{Q}\^{init}} (\textup{FCA-cost}(\mathbf{X}\_{0}, \mathbf{X}\_{1}, \mu\^{\diamond})) $
> $ = \mathbb{E}\_{\mathbf{X}\_{0}, \mathbf{X}\_{1} \sim \mathbb{Q}\^{init}} (\textup{FCA-cost}(\mathbf{X}\_{0}, \mathbf{X}\_{1}, \mu\^{\diamond})) $
> $ - \sup_{\mathbb{Q}, \mathcal{W} : \mathbb{Q}(\mathcal{W}) \le \epsilon} \mathbb{E}\_{\mathbf{X}\_{0}, \mathbf{X}\_{1} \sim \mathbb{Q}}
> \big( \textup{FCA cost} (\mathbf{X}\_{0}, \mathbf{X}\_{1}, \mu) - \textup{K-means cost} (\mathbf{X}\_{0}, \mathbf{X}\_{1}, \mu) \big)
> \mathbb{1}\left((\mathbf{X}\_{0}, \mathbf{X}\_{1}) \in \mathcal{W} \right) $
> $ + \sup_{\mathbb{Q}, \mathcal{W} : \mathbb{Q}(\mathcal{W}) \le \epsilon} \mathbb{E}\_{\mathbf{X}\_{0}, \mathbf{X}\_{1} \sim \mathbb{Q}}
> \big( \textup{FCA cost} (\mathbf{X}\_{0}, \mathbf{X}\_{1}, \mu) - \textup{K-means cost} (\mathbf{X}\_{0}, \mathbf{X}\_{1}, \mu) \big)
> \mathbb{1}\left((\mathbf{X}\_{0}, \mathbf{X}\_{1}) \in \mathcal{W} \right) $

---

> ### Author Response · Authors · 2024-12-01
> **Proof of Theorem A'' (part 2)**
>
> $ \le \mathbb{E}\_{\mathbf{X}\_{0}, \mathbf{X}\_{1} \sim \mathbb{Q}'} (\textup{FCA-cost}(\mathbf{X}\_{0}, \mathbf{X}\_{1}, \mu\^{\diamond})) - \mathbb{E}\_{\mathbf{X}\_{0}, \mathbf{X}\_{1} \sim \mathbb{Q}'} $
> $ \big( \textup{FCA cost} (\mathbf{X}\_{0}, \mathbf{X}\_{1}, \mu) - \textup{K-means cost} (\mathbf{X}\_{0}, \mathbf{X}\_{1}, \mu) \big)
> \mathbb{1} \left((\mathbf{X}\_{0}, \mathbf{X}\_{1}) \in \mathcal{W}' \right) $
> $ + sup_{\mathbb{Q}, \mathcal{W} : \mathbb{Q}(\mathcal{W}) \le \epsilon} \mathbb{E}\_{\mathbf{X}\_{0}, \mathbf{X}\_{1} \sim \mathbb{Q}}
> \big( \textup{FCA cost} (\mathbf{X}\_{0}, \mathbf{X}\_{1}, \mu) - \textup{K-means cost} (\mathbf{X}\_{0}, \mathbf{X}\_{1}, \mu) \big)
> \mathbb{1}\left((\mathbf{X}\_{0}, \mathbf{X}\_{1}) \in \mathcal{W} \right) $
>
> $ = \tilde{C}(\mathbb{Q}', \mathcal{W}', \mu\^{\diamond}) + sup_{\mathbb{Q}, \mathcal{W} : \mathbb{Q}(\mathcal{W}) \le \epsilon} \mathbb{E}\_{\mathbf{X}\_{0}, \mathbf{X}\_{1} \sim \mathbb{Q}} $
> $ \big( \textup{FCA cost} (\mathbf{X}\_{0}, \mathbf{X}\_{1}, \mu) - \textup{K-means cost} (\mathbf{X}\_{0}, \mathbf{X}\_{1}, \mu) \big)
> \mathbb{1}\left((\mathbf{X}\_{0}, \mathbf{X}\_{1}) \in \mathcal{W} \right). $
>
> (iii): The last term of the right-hand-side can be bounded as follows:
> First, let $\mu\^{\diamond}(\mathbf{x}) := argmin_{\mu\^{\diamond}\_{k} \in \mu\^{\diamond}} \Vert \mathbf{x} - \mu\^{\diamond}\_{k} \Vert^{2}$ for $ \mathbf{x} \in \mathcal{X}$ and
> $ \mu\^{\diamond}(\mathbf{x}\_{0}, \mathbf{x}\_{1}) := argmin_{\mu\^{\diamond}\_{k} \in \mu\^{\diamond}} \Vert \mathbf{T}\^{\textup{A}}(\mathbf{x}\_{0}, \mathbf{x}\_{1}) - \mu\^{\diamond}\_{k} \Vert^{2}$
> for $(\mathbf{x}\_{0}, \mathbf{x}\_{1}) \in \mathcal{X}\_{0} \times \mathcal{X}\_{1}.$
> Then,
> $\forall (\mathbf{x}\_{0}, \mathbf{x}\_{1}) \in \mathcal{X}\_{0} \times \mathcal{X}\_{1},$
> it holds that
> $ \textup{FCA cost}(\mathbf{x}\_{0}, \mathbf{x}\_{1}, \mu\^{\diamond}) - \textup{K-means cost}(\mathbf{x}\_{0}, \mathbf{x}\_{1}, \mu\^{\diamond})
> = 2 \pi_{0} \pi_{1} \Vert \mathbf{x}\_{0} - \mathbf{x}\_{1} \Vert^{2} + \Vert \mathbf{T}\^{\textup{A}}(\mathbf{x}\_{0}, \mathbf{x}\_{1}) - \mu\^{\diamond}(\mathbf{x}\_{0}, \mathbf{x}\_{1}) \Vert^{2} - \left( \pi_{0} \Vert \mathbf{x}\_{0} - \mu\^{\diamond}(\mathbf{x}\_{0}) \Vert^{2} \right) - \left( \pi_{1} \Vert \mathbf{x}\_{1} - \mu\^{\diamond}(\mathbf{x}\_{1}) \Vert^{2} \right) $
> $ \le 2 \pi_{0} \pi_{1} \Vert \mathbf{x}\_{0} - \mathbf{x}\_{1} \Vert^{2} + \Vert \mathbf{T}\^{\textup{A}}(\mathbf{x}\_{0}, \mathbf{x}\_{1}) - \mu\^{\diamond}(\mathbf{x}\_{0}) \Vert^{2}- \left( \pi_{0} \Vert \mathbf{x}\_{0} - \mu\^{\diamond}(\mathbf{x}\_{0}) \Vert^{2} \right) - \left( \pi_{1} \Vert \mathbf{x}\_{1} - \mu\^{\diamond}(\mathbf{x}\_{1}) \Vert^{2} \right) $
> $ \le 2 \pi_{0} \pi_{1} \Vert \mathbf{x}\_{0} - \mathbf{x}\_{1} \Vert^{2} + \pi_{0} \Vert \mathbf{x}\_{0} - \mu\^{\diamond}(\mathbf{x}\_{0}) \Vert^{2} + \pi_{1} \Vert \mathbf{x}\_{1} - \mu\^{\diamond}(\mathbf{x}\_{0}) \Vert^{2} - \left( \pi_{0} \Vert \mathbf{x}\_{0} - \mu\^{\diamond}(\mathbf{x}\_{0}) \Vert^{2} \right) - \left( \pi_{1} \Vert \mathbf{x}\_{1} - \mu\^{\diamond}(\mathbf{x}\_{1}) \Vert^{2} \right) $
> $ = 2 \pi_{0} \pi_{1} \Vert \mathbf{x}\_{0} - \mathbf{x}\_{1} \Vert^{2} + \pi_{1} ( \Vert \mathbf{x}\_{1} - \mu\^{\diamond}(\mathbf{x}\_{0}) \Vert^{2} - \Vert \mathbf{x}\_{1} - \mu\^{\diamond}(\mathbf{x}\_{1}) \Vert^{2} )
> \le 2 \pi_{0} \pi_{1} \Vert \mathbf{x}\_{0} - \mathbf{x}\_{1} \Vert^{2} + \pi\_{1} \Vert \mathbf{x}\_{1} - \mu\^{\diamond}(\mathbf{x}\_{0}) \Vert^{2} $
> $ \le \frac{1}{2} \Vert \mathbf{x}\_{0} - \mathbf{x}\_{1} \Vert\^{2} + \Vert \mathbf{x}\_{1} - \mu\^{\diamond}(\mathbf{x}\_{0}) \Vert^{2} \le \frac{1}{2} 2 R + 2R = 3R.$
>
> Hence, we conclude that
> $ sup_{\mathbb{Q}, \mathcal{W} : \mathbb{Q}(\mathcal{W}) \le \epsilon} \mathbb{E}\_{\mathbf{X}\_{0}, \mathbf{X}\_{1} \sim \mathbb{Q}} \big( \textup{FCA cost} (\mathbf{X}\_{0}, \mathbf{X}\_{1}, \mu) - \textup{K-means cost} (\mathbf{X}\_{0}, \mathbf{X}\_{1}, \mu) \big) \mathbb{1}\left((\mathbf{X}\_{0}, \mathbf{X}\_{1}) \in \mathcal{W} \right) $
> $ \le 3 R sup_{\mathbb{Q}, \mathcal{W} : \mathbb{Q}(\mathcal{W}) \le \epsilon} \mathbb{E}\_{\mathbf{X}\_{0}, \mathbf{X}\_{1} \sim \mathbb{Q}} (\mathbb{1}\left((\mathbf{X}\_{0}, \mathbf{X}\_{1}) \in \mathcal{W} \right)) = sup_{\mathbb{Q}, \mathcal{W} : \mathbb{Q}(\mathcal{W}) \le \epsilon} 3 R \mathbb{Q}(\mathcal{W}) =  3R \epsilon. $
>
> Finally, combining (ii) and (iii), we have
> $ \tilde{C}(\mathbb{Q}\^{init}, \mathcal{W}\^{init}, \mu\^{\diamond}) \le  \tilde{C}(\mathbb{Q}', \mathcal{W}', \mu\^{\diamond}) +  3R \epsilon,$
> which implies
> $C(\mu\^{\diamond}, \mathcal{A}\_{0}\^{init}, \mathcal{A}\_{1}\^{init}) \le C(\mu\^{\diamond}, \mathcal{A}\_{0}', \mathcal{A}\_{1}') + 3 R \epsilon,$
> where $\mathcal{A}\_{0}\^{init}$ and $\mathcal{A}\_{1}\^{init}$ are the fair assignment functions corresponding to $\mathcal{Q}\^{init}$ and $\mathcal{W}\^{init}.$

---

### Official Review · Reviewer_3f1D · 2024-11-03

**Soundness:** 3
**Presentation:** 3
**Contribution:** 3
**Rating:** 6
**Confidence:** 3

**Summary:**

This paper considers the fair clustering problem with the goal of optimizing a clustering objective under specifically designed fairness constraints. In practice, achieving the optimal trade-off between fairness level and clustering utility is challenging, and the existing fair clustering algorithms may yield suboptimal clustering utility or suffer from numerical instability in reaching a certain fairness level. This paper aims to address these challenges by developing a new in-processing algorithm that can achieve the optimal trade-off between fairness level and clustering utility without numerical instability. The main idea of this paper is to align data from two protected groups by transforming them into a common space (called the aligned space) and then applying a standard clustering algorithm in the aligned space. Authors prove that the optimal fair clustering is equivalent to the optimal clustering in the aligned space. Specifically, for the fair k-means clustering objective, authors propose a new fair clustering algorithm (called FCA) based on a novel decomposition. The proposed algorithm first operates by finding a joint probability distribution to align the data from different protected groups, and then optimizes cluster centers in the aligned space. The FCA algorithm ensures optimal clustering utility for any specified fairness level while avoiding the need to solve complex constrained optimization problems, thereby obtaining optimal fair clustering in practice. Experimental results show that the FCA algorithm outperforms existing baselines in terms of both the trade-off and numerical stability, and optimally controls the trade-off across various fairness levels.

**Strengths:**

This paper propose a novel decomposition method for the fair clustering cost, dividing it into two components: the transport cost with respect to a joint distribution between two protected groups, and the clustering cost with respect to cluster centers in the aligned space. Based on the proposed decomposition method, this paper develops a fair clustering algorithm FCA, which is transparent, stable, and can guarantee the convergence. Additionally, this paper theoretically proves that FCA achieves the optimal trade-off between fairness level and clustering utility.

**Weaknesses:**

While the proposed method performs well for two-group fairness, its applicability to multiple groups remains unclear. Additionally, the theoretical analysis lacks the discussions on the relationship between cost and fairness level, making it difficult to assess the trade-off achieved. Although FCA shows improved numerical stability in achieving perfect fairness, it is unclear whether this comes at the expense of clustering cost and to what extent. Finally, the experiments conducted in this paper are limited to small datasets, lacking evaluations on larger datasets and different $k$ values.

**Questions:**

Q1: While the proposed method can achieve the optimal trade-off between fairness level and clustering utility in practice on two protected groups, I am interested in whether this method can be extended to handle multiple groups, which is a more general case.

Q2: Since the proposed method provides a trade-off, is it theoretically possible to show the relation between the cost and the fairness level?

Q3: The fairlet-based method by Chierichetti et al. is for the k-center and k-median problem in metric space, whereas the proposed method in this paper is designed for the Euclidean k-means problem. In Figure 1, the proposed method demonstrates a lower cost than that of Chierichetti et al. Is this comparison fair?

Q4: In Table 2, the authors compare the numerical stability of FCA and VFC when achieving the maximum balance. The experimental results show that FCA performs better in achieving perfect fairness. Whether this is achieved by sacrificing part of the clustering cost, and if so, to what extent?

Q5: Why the experiments are conducted on smaller datasets? Is it possible to test the proposed method on larger datasets, e.g., millions or even hundreds of millions? Additionally, testing more k values across different datasets will be beneficial.

**Details Of Ethics Concerns:**

Since this is mainly a theoretical results, I don't think there are ethical issues that need to be considered.

---

> ### Author Response · Authors · 2024-11-22
> **Point-by-point responses #1**
>
> ### Weaknesses
>
> - - - - - -
>
> > **W1**: While the proposed method performs well for two-group fairness, its applicability to multiple groups remains unclear.
>
>
> - *Answer*
>
> In fact, **we can extend FCA to handle multiple (more than two) protected groups.**
> Note that this paper mainly focuses on the case of two protected groups, for ease of discussion.
> The main idea is similar: matching multiple individuals from different protected groups.
>
> Let $G$ be the number of protected groups, and denote $s\^{\*} \\in \\{0, \ldots, G-1\\}$ as a fixed (reference) sensitive attribute.
> For the case of equal sample sizes that $n\_{0} = \ldots = n\_{G-1},$ we can similarly decompose the objective function in terms of matching map as follows, which we prove in Section A.3 of Appendix:
> $$ C(\mu, \mathcal{A}\_{0}, \ldots, \mathcal{A}\_{G-1})
>                 = \mathbb{E} \sum\_{k=1}\^{K} \mathcal{A}\_{S}(\mathbf{X}) \Vert \mathbf{X} - \mu\_{k} \Vert\^{2}
> $$
> $$
> = \\mathbb{E}\_{s\^{\*}} \\sum\_{k=1}\^{K} \\mathcal{A}\_{s\^{\*}} (\\mathbf{X})\_{k} \\frac{1}{G}
> ( \\sum\_{s=0}\^{G-1} \\Vert \\frac{ \\sum\_{s’ \\neq s} (\\mathbf{T}\_{s}(\\mathbf{X}) - \\mathbf{T}\_{s’}(\\mathbf{X})) }{G} \\Vert\^{2} + \\Vert \\frac{ \\mathbf{X} + \\sum\_{s\^{’} \\neq s\^{\*}} \\mathbf{T}\_{s\^{’}}(\\mathbf{X}) }{G} - \\mu\_{k} \\Vert\^{2}  ),
> $$
>
> where $\mathbf{T}\_{s’}$ is the one-to-one matching map from
> $\mathcal{X}\_{s\^{\*}}$ to $\mathcal{X}\_{s’}$ for $s' \\in \\{0, \ldots, G-1\\}.$
> Note that this result holds for any $s\^{\*} \\in \\{0, \ldots, G-1 \\}.$
>
> For the case of unequal sample sizes, we expect that a similar decomposition (i.e., eq (A) below) would be valid.
> We first define the alignment map as:
>         $
>         \mathbf{T}\^{\textup{A}} (\mathbf{x}\_{0}, \ldots, \mathbf{x}\_{G-1})
>         := \pi\_{0} \mathbf{x}\_{0} + \ldots + \pi\_{G-1} \mathbf{x}\_{G-1}.
>         $
> Then, we find the joint distribution and cluster centers by minimizing the following objective (A):
>         $$
>             \mathbb{E}\_{ (\mathbf{X}\_{0}, \ldots, \mathbf{X}\_{G-1}) \sim \mathbb{Q}}
>             \left(
>             \sum\_{s = 0}\^{G-1} G \pi\^{G} \bigg\Vert \sum_{s' \neq s} \left( \mathbf{X}\_{s} - \mathbf{X}\_{s’} \right) \bigg\Vert\^{2}
>             + \min\_{k} \Vert \mathbf{T}\^{\textup{A}} ( \mathbf{X}\_{0}, \ldots, \mathbf{X}\_{G-1} ) - \mu\_{k} \Vert^{2}
>             \right),
>             $$
> where $\pi\^{G} := \prod\_{s=0}\^{G-1} \pi\_{s}.$
> Furthermore, since finding the joint distribution for multiple protected groups is technically similar to the case of two protected groups, the optimization of the objective in eq. (A) can be solved using a linear program.
> Unfortunately, we have not succeeded in proving (A), which we are trying now.
>
> We newly add this feasible extension of FCA, with details in Section A.3 of Appendix.
>
>
>
> - - - - - -
>
> > **W2**: Additionally, the theoretical analysis lacks the discussions on the relationship between cost and fairness level, making it difficult to assess the trade-off achieved.
>
>
> - *Answer*
>
> FCA-C, proposed in Section 4.2, is specifically designed to control the trade-off between cost and fairness level.
>
> As shown in Theorem 4.1, **FCA-C achieves the optimal trade-off: it finds the optimal fair clustering for any given fairness level.**
> In other words, for a given fairness level, the clustering obtained by FCA-C minimizes the cost, among all fair clusterings that satisfy the given fairness level.
>
>
> - - - - - -
>
> > **W3**: Although FCA shows improved numerical stability in achieving perfect fairness, it is unclear whether this comes at the expense of clustering cost and to what extent.
>
>
>
> - *Answer*
>
>    FCA does not sacrifice additional clustering cost while achieving perfect fairness.
>    In other words, FCA achieves the optimal trade-off between cost and fairness level, as theoretically shown in Theorem 3.1.
>
>    **Experimentally, FCA offers the lowest cost (i.e., achieves the best trade-off) among fair clustering algorithms capable of achieving near-perfect fairness.**
>    For example, as shown in Table 1, FCA consistently shows significantly lower costs compared to baseline algorithms.

---

> ### Author Response · Authors · 2024-11-22
> **Point-by-point responses #2**
>
> > **W4**: Finally, the experiments conducted in this paper are limited to small datasets, lacking evaluations on larger datasets and different $K$ values.
>
>
> - *Answer*
>
>    - Note that the datasets and the fixed number of clusters $K$ used in our experiments are also commonly considered in various previous works (e.g., Ziko et al. (2021)).
>    However, we agree that such additional analyses would enrich our experimental studies.
>    In response, we newly conduct two additional experiments: (i) on a larger dataset and (ii) with various values of $K.$
>
>    - **(Larger dataset)**
>    Since we could not find real data with millions of records, we generate synthetic data for this experiment.
>         A dataset with $n=10^{6}$ is constructed using a Gaussian mixture model, whose full details are provided in Section C.3.9 of Appendix.
>         FCA and the baseline algorithm (VFC) are applied to this large dataset, and the comparison results are presented in Table 13 of the same section.
>         **The result confirms that FCA can be scaled-up, outperforms VFC, and is still a favorable FC algorithm on large datasets.**
>         Moreover, the mini-batch technique is still valid even when $n$ is extremely large.
>
>    - **(Various $K$ values)**
>         We vary $K \in \{ 5, 10, 20, 40 \}$, and present the comparison results in Figure 7 in Section C.3.6 of Appendix.
>         **This results implies that FCA still outperforms baselines across various values of $K.$**
>         In specific, FCA achieves lower values of \texttt{Cost} than baselines for most values of $K,$ while maintaining the highest fairness level of $\texttt{Balance} \approx 1.$
>
> - - - - - -
> - - - - - -
>
> ### Questions
>
> - - - - - -
>
> > **Q1**: While the proposed method can achieve the optimal trade-off between fairness level and clustering utility in practice on two protected groups, I am interested in whether this method can be extended to handle multiple groups, which is a more general case.
>
> - *Answer*
>
> Please see our response to W1 above.
>
> - - - - - -
>
> > **Q2**: Since the proposed method provides a trade-off, is it theoretically possible to show the relation between the cost and the fairness level?
>
> - *Answer*
>
> Please see our response to W2 above.
>
> - - - - - -
>
> > **Q3**: The fairlet-based method by Chierichetti et al. is for the k-center and k-median problem in metric space, whereas the proposed method in this paper is designed for the Euclidean k-means problem. In Figure 1, the proposed method demonstrates a lower cost than that of Chierichetti et al. Is this comparison fair?
>
> - *Answer*
>
>    As you pointed out, since SFC (the fairlet-based method) was originally designed for $K$-median clustering, we additionally compare FCA and SFC based on the $K$-median clustering.
>
>    Fortunately, we could show that **FCA can be extended to the $K$-median clustering problem, by replacing $L_{2}$ norm with $L_{1}$ norm in the objective function of FCA.**
>    In fact, it can be extended to any $K$-clustering with the $L_{p}$ norm, where $p \ge 1$.
>    See Section A.4 for details about its mathematical justification.
>
>    **In Section C.3.8 of Appendix, we present the experimental results comparing FCA and SFC, based on the $K$-median clustering cost.
>    Table 12 shows the results that FCA still outperforms SFC**, i.e., achieves lower costs and higher balances.
>
> - - - - - -
>
> > **Q4**: In Table 2, the authors compare the numerical stability of FCA and VFC when achieving the maximum balance. The experimental results show that FCA performs better in achieving perfect fairness. Whether this is achieved by sacrificing part of the clustering cost, and if so, to what extent?
>
> - *Answer*
>
> Please see our response to W3 above.
>
> - - - - - -
>
> > **Q5**: Why the experiments are conducted on smaller datasets? Is it possible to test the proposed method on larger datasets, e.g., millions or even hundreds of millions? Additionally, testing more k values across different datasets will be beneficial.
>
> - *Answer*
>
> Please see our response to W4 above.

---

> > ### Comment · Reviewer_3f1D · 2024-11-25
> > **Response to the Rebuttal**
> >
> > Thanks for the response. All of my concerns have been addressed. I will keep my initial score.

---

### Official Review · Reviewer_1MfK · 2024-11-04

**Soundness:** 4
**Presentation:** 2
**Contribution:** 3
**Rating:** 8
**Confidence:** 4

**Summary:**

This paper proposed a new algorithm for fair clustering (FC). The proposed method mainly focuses on the case the sensitive attribute is binary. This mean every sample is a pair $(x,s)$ where $x$ is a $d$-vector and $s$ (the sensitive label) is 0 or 1. The goal is to divide the samples into clusters, and ensure that the distribution of samples with $s=0$ or $1$ is consistent with the distribution in the whole data set.

The main idea of the proposed method (called FCA) is to formulate the FC problem as an optimization problem about the clustering center locations and a  joint distribution matrix. The process would alternatively optimize the distribution and the center location. Each step of optimization process is decomposed into two phases: first, one tries to find a one-to-many matching (represented by joint distribution matrix $Q$) from $0$ samples to $1$ samples. This can be formulated as a problem similar to the optimal transport (OT) and can be solved using linear programming. Then in the second phase, one finds tries to find a set of centers for the $k$-clustering, with the assumption that $0$ and $1$ samples are matched/"aligned" as specified by $Q$, and matched samples are to be put into the sample cluster, which enforce the fairness requirement. This can be done using standard clustering algorithms such as the K-means++ algorithm or gradient descent methods on the so called "aligned space" induced from $Q$.

While the above method assumes the strict fairness balance requirement, the paper also present an extension (call FCA-C) that allows control of the balance level. This relaxation is achieved by apply the standard K-means algorithm to some the samples and FCA to the others. This adds another sets of parameters (which specified which samples to apply standard k-means) to the optimization process.

**Strengths:**

This paper present a fairly strong theoretical result. The proposed algorithms and the analysis are quite non-trivial. The idea of aligning data from different sensitive groups looks quite interesting. This optimization process algo reveals an interesting relation between fair clustering and optimal transport.

**Weaknesses:**

The discussion mainly focuses on the case that there are only two sensitive groups. It is not clear whether there is a possibility that the proposed method can be extended to multinary sensitive attributes. The result could be stronger if the authors could provide some hint on how some of the core ideas are not restricted to binary attributes setting.

The writing can be improved. The clustering problem setting is quite different from standard k-means clustering. More detailed explanation in the preliminary section is suggested. For example, it could be not immediately clear that why we need to consider the empirical distribution in a clustering problem.

**Questions:**

Which of the ideas in FCA/FCA-C do you believe are extendable to multinary sensitive attributes cases? Any thoughts on how the extension could possibly be achieved?

---

> ### Author Response · Authors · 2024-11-22
> **Point-by-point responses**
>
> ### Weaknesses
>
> - - - - - -
>
> > **W1**: The discussion mainly focuses on the case that there are only two sensitive groups. It is not clear whether there is a possibility that the proposed method can be extended to multinary sensitive attributes. The result could be stronger if the authors could provide some hint on how some of the core ideas are not restricted to binary attributes setting.
>
>
> - *Answer*
>
> Yes, **we can extend FCA to handle multiple (more than two) protected groups.**
> Note that this paper mainly focuses on the case of two protected groups, for ease of discussion.
> The main idea is similar: matching multiple individuals from different protected groups.
>
> Let $G$ be the number of protected groups, and denote $s\^{\*} \\in \\{0, \ldots, G-1\\}$ as a fixed (reference) sensitive attribute.
> For the case of equal sample sizes that $n\_{0} = \ldots = n\_{G-1},$ we can similarly decompose the objective function in terms of matching map as follows, which we prove in Section A.3 of Appendix:
> $$ C(\mu, \mathcal{A}\_{0}, \ldots, \mathcal{A}\_{G-1})
>                 = \mathbb{E} \sum\_{k=1}\^{K} \mathcal{A}\_{S}(\mathbf{X}) \Vert \mathbf{X} - \mu\_{k} \Vert\^{2}
> $$
> $$
> = \\mathbb{E}\_{s\^{\*}} \\sum\_{k=1}\^{K} \\mathcal{A}\_{s\^{\*}} (\\mathbf{X})\_{k} \\frac{1}{G}
> ( \\sum\_{s=0}\^{G-1} \\Vert \\frac{ \\sum\_{s’ \\neq s} (\\mathbf{T}\_{s}(\\mathbf{X}) - \\mathbf{T}\_{s’}(\\mathbf{X})) }{G} \\Vert\^{2} + \\Vert \\frac{ \\mathbf{X} + \\sum\_{s\^{’} \\neq s\^{\*}} \\mathbf{T}\_{s\^{’}}(\\mathbf{X}) }{G} - \\mu\_{k} \\Vert\^{2}  ),
> $$
>
> where $\mathbf{T}\_{s’}$ is the one-to-one matching map from
> $\mathcal{X}\_{s\^{\*}}$ to $\mathcal{X}\_{s’}$ for $s' \\in \\{0, \ldots, G-1\\}.$
> Note that this result holds for any $s\^{\*} \\in \\{0, \ldots, G-1 \\}.$
>
> For the case of unequal sample sizes, we expect that a similar decomposition (i.e., eq (A) below) would be valid.
> We first define the alignment map as:
>         $
>         \mathbf{T}\^{\textup{A}} (\mathbf{x}\_{0}, \ldots, \mathbf{x}\_{G-1})
>         := \pi\_{0} \mathbf{x}\_{0} + \ldots + \pi\_{G-1} \mathbf{x}\_{G-1}.
>         $
> Then, we find the joint distribution and cluster centers by minimizing the following objective (A):
>         $$
>             \mathbb{E}\_{ (\mathbf{X}\_{0}, \ldots, \mathbf{X}\_{G-1}) \sim \mathbb{Q}}
>             \left(
>             \sum\_{s = 0}\^{G-1} G \pi\^{G} \bigg\Vert \sum_{s' \neq s} \left( \mathbf{X}\_{s} - \mathbf{X}\_{s’} \right) \bigg\Vert\^{2}
>             + \min\_{k} \Vert \mathbf{T}\^{\textup{A}} ( \mathbf{X}\_{0}, \ldots, \mathbf{X}\_{G-1} ) - \mu\_{k} \Vert^{2}
>             \right),
>             $$
> where $\pi\^{G} := \prod\_{s=0}\^{G-1} \pi\_{s}.$
> Furthermore, since finding the joint distribution for multiple protected groups is technically similar to the case of two protected groups, the optimization of the objective in eq. (A) can be solved using a linear program.
> Unfortunately, we have not succeeded in proving (A), which we are trying now.
>
> We newly add this feasible extension of FCA, with details in Section A.3 of Appendix.
>
>
> - - - - - -
>
> > **W2**: The writing can be improved. The clustering problem setting is quite different from standard k-means clustering. More detailed explanation in the preliminary section is suggested. For example, it could be not immediately clear that why we need to consider the empirical distribution in a clustering problem.
>
>
> - *Answer*
>
> Thank you for the constructive comment.
> In Section 2, we newly add the following statement:
> *`We specifically define these distributions to discuss the (probabilistic) matching between two protected groups of different sizes, and subsequent probabilistic assignments for fair clustering.’*
>
>
> - - - - - -
> - - - - - -
>
>
> ### Questions
>
> - - - - - -
>
> > **Q1**: Which of the ideas in FCA/FCA-C do you believe are extendable to multinary sensitive attributes cases? Any thoughts on how the extension could possibly be achieved?
>
>
> - *Answer*
>
> Please see our response to W1 above.

---

> > ### Comment · Reviewer_1MfK · 2024-11-25
> >
> > I thank the authors for providing more discussion on the potential extension to multiple protected groups. I can understand that focusing mainly the two group cases would make the discussion easier, and that some of the facts about the multiple group extension are not yet proved. I am pleased to keep my current score.

---

### Official Review · Reviewer_FzLv · 2024-11-06

**Soundness:** 2
**Presentation:** 3
**Contribution:** 2
**Rating:** 6
**Confidence:** 4

**Summary:**

This paper proposes a problem, called Fair Clustering via Alignments (FCA), which is a new formulation of the well-known Fair Clustering problem that optimizes clustering objective functions subject to fairness constraints, represented by the notion of the "balance" of different groups of data in each cluster. There are two factors concerning the objective cost: the alignment of points that guarantees fairness in clustering, and the cluster centers. The proposed algorithm take turns to fix one and optimize the other, until the solution converges. This method has improved performance over previous solutions while satisfying the fairness constraints.

**Strengths:**

The introduced problem is an interesting re-interpretation of a well-known problem. The method separates aligning points from different groups from the computation of clustering and also the centers, which might be useful intuition in practice. I'm not sure if it is the first of such iterative fair clustering approaches, but it seems novel to me.

En route this paper has also developed a more generalized formulation of the fair clustering problem with fractional cluster assignments and analyzed its properties. It has arguably made the optimization easier to solve.

The algorithm has strong performance in utility, including clustering cost and fairness constraint, in numerical experiments.

**Weaknesses:**

Although this paper proposes an FCA algorithm that works in a Lloyd's k-means algorithm fashion, I think its main contribution is more conceptual rather than algorithmic. Indeed the optimal solution of FCA is guaranteed to be the optimal fair k-means solution subject to some fairness constraints; however this does not guarantee an algorithm that can efficiently find that solution. In that sense, the abstract is also slightly misleading. Judging from the discussion I don't think the given algorithm is guaranteed to find the global optimal solution, instead of converging to local optimum.

I also feel there should be a thorough discussion on run-time and approximation guarantees of this new method. It could be slower than the fairlet-based approache since it could take many iterations to converge despite better performance.

The decomposition cannot easily be extended to other clustering problems, even the k-clustering ones since it heavily depends on the $ell_2^2$ distance properties.

**Questions:**

The proposed algorithm, like Lloyd's, either fixes the centers and optimizes the alignments; or fixes the grouping and recompute the clustering. For Lloyd's we know certain seeding mechanisms for the initialization, such as the k-means++ mentioned in the paper, can improve performance guarantees. Do we know such things for the FCA algorithm?

I'm wondering what role is played by that decomposition of clustering objective function in the analysis and algorithm design. Is it mostly for the sake of separating the two terms

Apart from the perfect fairness case, when is the optimal distribution guaranteed to be deterministic and when will the algorithm find deterministic solutions? If it is not deterministic, how do we round it to a solution that is deterministic?

This paper separates the problem into two parts. It is essentially an iterative version of the fairlet-based method that keeps updating the fairlets themselves, at least when the clusters have the same balance as the original dataset. Although the decomposition might not hold for other k-clustering problems, is it possible to apply the same logic there?

---

> ### Author Response · Authors · 2024-11-22
> **Point-by-point responses #1**
>
> ### Weaknesses
>
> - - - - - -
>
> > **W1**:
>     Although this paper proposes an FCA algorithm that works in a Lloyd's k-means algorithm fashion, I think its main contribution is more conceptual rather than algorithmic. Indeed the optimal solution of FCA is guaranteed to be the optimal fair k-means solution subject to some fairness constraints; however this does not guarantee an algorithm that can efficiently find that solution. In that sense, the abstract is also slightly misleading. Judging from the discussion I don't think the given algorithm is guaranteed to find the global optimal solution, instead of converging to local optimum.
>
> - *Answer*
>
> As Lloyd's $K$-means algorithm has the descent property (i.e., the cost decreases with each iteration; Lloyd (1982); Selim \& Ismail (1984)), FCA inherits this property as well and thus converges to an optimum.
> The solution would be a local optimum, because FCA depends on the initial cluster centers (as well as Lloyd's algorithm) and involves optimizing the joint distribution.
>
> To clarify this point, we modify the abstract and introduction by adding the term `(local) optimum' where appropriate.
>
> Experimentally, we observe that using Lloyd's algorithm with $K$-means++ initialization in FCA results in more effective performance compared to baseline methods (FCA tends to find a good local optimum).
> Moreover, in Table 7 in Section C.3.4 of Appendix, we compare three algorithms: $K$-means (Lloyd's) algorithm with $K$-means++ initialization and random initialization (added newly in the revised paper), and a gradient-based optimization.
> The results show that $K$-means++ initialization provides superior performance.
>
> - - - - - -
>
> > **W2**: I also feel there should be a thorough discussion on run-time and approximation guarantees of this new method. It could be slower than the fairlet-based approaches since it could take many iterations to converge despite better performance.
>
> - *Answer*
>
> Thank you for the constructive feedback.
> In response, we discuss the point by comparing computational time for FCA with the fairlet-based method, SFC, as follows.
>
> Note that FCA involves two levels of iteration:
> an outer iteration (alternately updating the joint distribution and cluster centers),
> and an inner iteration (updating the cluster centers using Lloyd's $K$-means algorithm).
>
> 1. We compare the number of iterations required for the $K$-means algorithm to converge.
> For SFC, we calculate the number of iterations in the $K$-means algorithm used (after finding fairlets).
> For FCA, we calculate the total number of $K$-means algorithm iterations, summed over all outer iterations.
>
>    **The results, presented in Table 10 in Section C.3.7 of Appendix, show that the computation time of FCA is almost linear with respect to the number of $K$-means algorithm iterations until convergence.**
>    However, SFC requires smaller number of total iterations compared to FCA, primarily because FCA involves the outer iterations.
>
> 2. To further analyze whether using early-stopping in FCA can maintain reasonable performance, we conduct an additional analysis:
> we fix the number of $K$-means iterations to 1 per outer iteration of FCA, then perform a total of 10 outer iterations.
> With this setup, the total number of iterations for FCA becomes 10, which is comparable to or smaller than that of SFC (15 for Adult, 10 for Bank, and 32 for Census dataset, as shown in Table 10).
> We observe that the performance of FCA with this **early-stopping is slightly worse than the original FCA (where $K$-means algorithm runs until convergence), but it still outperforms SFC (see Table 11 in Section C.3.7 of Appendix).**
> However, note that this early-stopping approach would be not recommended, at least, for the datasets used in our experiments.
> This is because running $K$-means algorithm takes less than a second or a few seconds, while the computation time of finding joint distribution dominates the time of running $K$-means algorithm until convergence.

---

> ### Author Response · Authors · 2024-11-22
> **Point-by-point responses #2**
>
> > **W3**:
> The decomposition cannot easily be extended to other clustering problems, even the k-clustering ones since it heavily depends on the $L_{2}$ distance properties.
>
> - *Answer*
>
> **We can extend FCA to general $L_{p}$ norms, such as the $L_{1}$ norm for $K$-median clustering,** as follows.
>
> The objective of $K$-clustering for the $L_{p}$ norm ($p \ge 1$) is given by
> $\mathbb{E} \sum\_{k=1}\^{K} \mathcal{A}\_{S}(\mathbf{X})\_{k} \Vert \mathbf{X} - \mu\_{k} \Vert\_{p}.$
> For the simple case where $n\_{0} = n\_{1},$ using the triangle inequality, we can derive an upper bound for the objective:
> $$
> \mathbb{E} \sum\_{k=1}\^{K} \mathcal{A}\_{S}(\mathbf{X})\_{k} \Vert \mathbf{X} - \mu\_{k} \Vert\_{p}
> \le \mathbb{E}\_{s} \sum\_{k=1}\^{K} \mathcal{A}\_{S}(\mathbf{X})\_{k} \left(
> \left\Vert \frac{\mathbf{X} - \mathbf{T}(\mathbf{X})}{2} \right\Vert\_{p}
> +
> \left\Vert \frac{\mathbf{X} + \mathbf{T}(\mathbf{X})}{2} - \mu\_{k} \right\Vert\_{p}
> \right).
> $$
> Hence, we can minimize this upper bound for the fair $K$-clustering objective with the $L_{p}$ norm.
> A more detailed discussion on this extension is newly added in Section A.4 of Appendix.
>
> In specific, we additionally conduct an experiment based on the $L_{1}$ norm for $K$-median clustering.
> See Section C.3.8 of Appendix for experiments comparing FCA (replacing $L_{2}$ norm with $L_{1}$ norm) and SFC (a fairlet-based method), in view of the $K$-median clustering cost.
> The result in Table 12 shows that FCA still outperforms SFC.
>
> - - - - - -
> - - - - - -
>
> ### Questions
>
> - - - - - -
>
> > **Q1**:
> The proposed algorithm, like Lloyd's, either fixes the centers and optimizes the alignments; or fixes the grouping and recompute the clustering. For Lloyd's we know certain seeding mechanisms for the initialization, such as the k-means++ mentioned in the paper, can improve performance guarantees. Do we know such things for the FCA algorithm?
>
> - *Answer*
>
> Thank you for the constructive comment.
> In response, we conduct an additional experiment to examine whether the initialization of cluster centers impacts FCA's performance much.
> Specifically, we run FCA using Lloyd's algorithm with randomly chosen initial centers, and compare with $K$-means++ initialization.
>
> We observe that $K$-means++ initialization performs better than random initialization.
> Detailed results are presented in Table 7 in Section C.3.4 of Appendix.
>
> - - - - - -
>
> > **Q2**:
> I'm wondering what role is played by that decomposition of clustering objective function in the analysis and algorithm design...
>
> - *Answer*
>
> In fact, the decomposition in Theorem 3.3 provides a key technical advantage for FCA algorithm.
> Specifically, it allows us to find a fair clustering without relying on constrained optimization, as the fairness constraint is implicitly assumed when building the aligned space.
>
> That is, once a joint distribution is given, cluster centers are optimized directly on the aligned space using standard clustering algorithms (e.g., $K$-means algorithm).
> This approach also enhances the numerical stability of FCA.
>
> - - - - - -
>
> > **Q3**:
> Apart from the perfect fairness case, when is the optimal distribution guaranteed to be deterministic and when will the algorithm find deterministic solutions? If it is not deterministic, how do we round it to a solution that is deterministic?
>
> - *Answer*
>
>    - The optimal joint distribution is guaranteed to be deterministic when the sizes of the two groups are equal ($n_{0} = n_{1}$), as discussed in Remark 3.4.
>    This is because the joint distribution becomes equivalent to the one-to-one matching.
>
>    - FCA can find optimal solutions in both cases where probabilistic or deterministic assignments are optimal (i.e., for unequal and equal sample sizes, respectively).
>    **FCA algorithm does not differ between the two cases; rather, it automatically finds deterministic assignment when the sample sizes are equal, or finds probabilistic assignment when the sample sizes are unequal.**
>    The discussion of the case of equal sizes is presented before the case of unequal sizes, to introduce our core idea of matching (alignment) more intuitively.
>
>    - Furthermore, probabilistic assignments can be rounded as follows.
>    We first construct a fair clustering achieving a given balance using FCA-C.
>    Then, deterministic assignments are made using the argmax approach (mentioned in lines 384-385), as done in our experiments.
>
>
> - - - - - -
>
> > **Q4**:
> This paper separates the problem into two parts. It is essentially an iterative version of the fairlet-based method that keeps updating the fairlets themselves, at least when the clusters have the same balance as the original dataset. Although the decomposition might not hold for other k-clustering problems, is it possible to apply the same logic there?
>
> - *Answer*
>
>    - Yes, as we mentioned in our response to W3, FCA can be extended to a given $K$-clustering objective with the $L_{p}$ norm ($p \ge 1$).

---

> ### Author Response · Authors · 2024-11-28
>
> *Dear reviewer FzLv*:
>
> Once again, thank you very much for your review, which has been greatly helpful to improve our work.
> We hope this post doesn't cause any inconvenience, but we have about 5-6 hours left to make revisions to the paper's PDF (although discussions can continue for a few more days).
>
> We would like to ask whether our responses have addressed your concerns or if there are any remaining discussion points that require further revisions to the paper (minor points can still be discussed over the next few days!).
>
> Thank you.

---

> > ### Comment · Reviewer_FzLv · 2024-11-28
> > **Thank you!**
> >
> > I've read the authors' response and the other reviewers' comments. I would like to thank the authors for their thorough extended discussion. The new experiments have slightly assuaged my concerns for the practical usefulness of this algorithm. I would keep my initial score.

---

### Author Response · Authors · 2024-11-22
**To all the reviewers: summary of changes (updated 11/26)**

**Dear reviewers:**
    We appreciate your review comments and are grateful for the opportunity to improve our work.
    We hope that our point-by-point responses provided to each of you, along with the revised paper (with changes highlighted in blue color), resolve your concerns.
    Below is a summary of major changes made to the revised paper.
    Note that most of these changes are incorporated in Appendix due to the page limit.

--------

- Main body

   - Several revisions have been made in Abstract, Sections 1, 2, 3.1, 4.2, 5.2, 5.4, and 6, including brief descriptions of the revisions to Appendix (provided below).

- Appendix
   - Section A.3:

        We discuss how FCA can be extended to multiple (more than two) protected groups, and provide a feasible approach.

   - Section A.4:

        We show that FCA, which is originally designed for $K$-means clustering, can also be applied to the $K$-clustering with the $L_{p}$ norm ($p \ge 1$), including the $L_{1}$ norm for $K$-median clustering.

   - Section A.5:

        We introduce other fairness notions in clustering problems, including proportional fairness and individual fairness.

   - Section B.5:

        We introduce an approximation guarantee of FCA-C.

   - Section C.3.4:

        We additionally perform FCA with randomly initialized cluster centers, and compare its performance to $K$-means++ initialization and gradient-based optimization.

   - Section C.3.6:

        We conduct an additional ablation study to examine the impact of $K$ (the number of clusters).
        The results consistently show that FCA outperforms baseline methods, across various values of $K.$

   - Section C.3.7:

        We compare the computation costs of FCA, FCA-C, and SFC (a fairlet-based method).
        The results imply that
        (i) the computation time of FCA-C is comparable to FCA, and
        (ii) FCA with reduced iterations (i.e., a number of iterations comparable to SFC), still outperforms SFC, despite the total number of iterations for original FCA being higher than that of SFC.

   - Section C.3.8:

        We compare FCA and SFC based on the $K$-median clustering cost.
        The results show that FCA still outperforms SFC.

   - Section C.3.9:

        We conduct an additional experiment using a larger dataset.
        The results show that FCA is scalable and outperforms a baseline method, VFC.

---

### Meta-Review · Area_Chair_YGT2 · 2024-12-17

**Metareview:**

This paper introduces a new approach to fair clustering, termed Fair Clustering via Alignments (FCA).  The authors propose a novel formulation that optimizes clustering objectives while satisfying fairness constraints based on the "balance" of different groups within each cluster.

Reviewers appreciate the new perspective offered by FCA and the combination of theoretical analysis and experimental validation. The proposed algorithm shows promising results in achieving a better balance between fairness and clustering utility.

However, reviewers also raise some concerns:

- Connection to Existing Work: The paper's notion of fairness is closely related to concepts presented in previous work by Chierichetti et al. This connection should be made more explicit, clearly differentiating the novel contributions of FCA.
- Approximation Guarantees: The paper could benefit from a clearer explanation of the approximation guarantees provided by the proposed algorithm.
- Writing Quality: The clarity and presentation of the paper could be further improved.
- Late-Stage Theoretical Addition: During the rebuttal phase, the authors introduced a new theoretical guarantee (O(1)-approximation with O(ε) violation). While interesting, this constitutes a significant change to the original submission and would ideally require re-evaluation.

Recommendation:

While the paper presents a promising approach to fair clustering, the reviewers feel that it does not meet the bar for acceptance at ICLR in its current form.

**Additional Comments On Reviewer Discussion:**

The paper improved a lot during discussion phase but major changes have been done and so the paper should be re-submitted to go through a complete review.

---

### Decision · Program_Chairs · 2025-01-22

Reject